# Somatic deficiency of the human E3 ubiquitin ligase CBL in leukocytes impairs B cell but not T cell development and function

The E3 ubiquitin ligase Casitas B-lineage lymphoma (CBL) promotes positive selection and antigen responses in mouse T lymphocytes by ubiquitinating ZAP70. Conversely, mouse CBL and CBL-B mutually redundantly regulate SYK ubiquitination and B cell receptor signaling. Here we studied individuals with somatically homozygous *CBL* loss-of-function variants in leukocytes. Human CBL is largely redundant for the development and function of human T cells. Conversely, B cell development is altered at the immature stage, with a tenfold increase in transitional cells, enhanced survival of autoreactive clones and impaired tolerance manifested by autoantibody production. B cell maturation is intrinsically impaired by reduced apoptosis and dysregulated B cell receptor signaling. CBL deficiency impairs humoral immunity by limiting memory B cell formation and reducing class switching and somatic hypermutation. Consequently, antigen-specific B cell generation and adaptive immune memory are disrupted, predisposing individuals to infection. Human CBL is critical for B cell development and function but redundant for T cell biology.

Casitas B-lineage lymphoma (CBL), discovered in 1989 (refs. 1,2), is an E3 ubiquitin ligase that restrains receptor-proximal kinase signaling[3] by ubiquitinating phosphorylated targets[4,5], such as ZAP70 (refs. 6,7) and SYK[8,9], thereby limiting T cell antigen receptor (TCR) and B cell receptor (BCR) signaling. Through these and other substrates, including FYN, LYN and LCK[10–12], CBL modulates amplitudes and dynamics of TCR and BCR signaling pathways[6–12], thus regulating antigen-induced responses in adaptive immune cells. The CBL paralog CBL-B shares substrates and provides partly redundant control[13,14].

The immunological implications of these regulatory actions have been thoroughly investigated in gene-targeted mice. CBL-deficient mice have hypercellular thymi and lymph nodes[15], whereas CBL-deficient thymocytes exhibit elevated TCR responses, leading to enhanced positive selection[16]. Deletion of both *Cbl* and *Cblb* from the mouse genome is embryonically lethal, while the combination of *Cblb* knockout (KO) and conditional deletion of *Cbl* in hematopoietic tissues causes mice to rapidly succumb to myeloproliferative disease[17]. Thymocyte-specific combined deletion of *Cbl* and *Cblb* leads to dysregulation of T cell development, including major histocompatability complex-independent

generation of CD4⁺ and CD8⁺ T cells[18]. Mice with germline knock-in (KI) of a ubiquitin ligation loss-of-function (LOF; Ub^LOF) mutant CBL develop severe T cell lymphopenia due to elevated TCR-dependent apoptosis during thymic development[19]. Despite these effects on T cells, neither *Cbl* KO nor Ub^LOF KI mice exhibit any detectable B cell phenotypes[15,19]. Conditional deletion of *Cbl* and *Cblb* in B-lineage cells caused substantial B cell dysregulation, as evidenced by elevated numbers of peripheral B cells, elevated serum IgM and systemic lupus erythematosus-like disease with autoantibodies to double-stranded DNA and nuclear antigen[20]. Thus, combined deficiency of both CBL and CBL-B underpin impressive B cell-intrinsic dysregulation in mice. These studies suggest that CBL has a nonredundant role in regulating mouse T cell biology but is largely redundant with CBL-B in regulating B cells.

In humans, inherited and somatic variants in *CBL* drive myeloid neoplasms[21]. Children with heterozygous germline *CBL* Ub^LOF variants undergo a myeloproliferative episode in early childhood when somatic loss-of-heterozygosity (LOH) occurs at the *CBL* locus[22]. Although this neoplasm typically resolves spontaneously[22], most hematopoietic cells remain permanently homozygous for the *CBL* Ub^LOF variant. Recently,

e-mail: s.tangye@garvan.org.au; bohlen@genzentrum.lmu.de

we demonstrated that homozygosity for *CBL* Ub[LOF] variants drives clinical autoinflammation through chronic monocyte activation[23,24]. Patient leukocytes, but not other cell types in the body, are permanently deficient in CBL ubiquitination, mimicking a conditional, hematopoietic *CBL* Ub[LOF] KI in mice. After 30 years of studies on CBL function in mouse lymphocytes, we have now studied the impact of CBL deficiency on human lymphocytes in vivo.

## Results

### Individuals with *CBL*-LOH show a high incidence of infectious disease

Our cohort consists of 11 individuals from 9 families (P1–P11) with germline monoallelic *CBL* Ub[LOF] variants and somatic LOH and 8 individuals (family members and unrelated individuals) with inherited heterozygous *CBL* Ub[LOF] variants[23] (Fig. 1a). All variants could bind substrates but were LOF for substrate ubiquitination[23]. Somatic LOH occurred through segmental uniparental isodisomy (UPD), as evidenced by whole-exome sequencing (WES; Fig. 1b). The breakpoint of this segmental UPD occurred at various positions in the q arm of chromosome 11, encompassing *CBL* among other genes (Fig. 1c). The UPD was detected in hematopoietic, but not nonhematopoietic, tissues (Fig. 1d). Within leukocytes, T and B lymphocytes of the participants (P1–P6) had variant allele frequencies above 90%, like monocytes or polymorphonuclear cells (Fig. 1e). Of note, the lymphocytes of P4 did not exhibit LOH at the time of analysis. It is unclear whether the somatic event originally only affected the participant's myeloid compartment or whether this occurred later in her life. Unusually severe infection occurred in 73% (8/11) of the participants (Extended Data Fig. 1a–d), with two fatalities (P10 and P11), representing a striking incidence of such events. Testing participants at ages 3–26 years indicated intact antibody responses to childhood infections and vaccines, as shown by detection of antibodies to various microbes using clinical serology testing and virome-wide serological profiling, including severe acute respiratory syndrome coronavirus 2 (SARS-CoV-2; Extended Data Fig. 2a,b). Detailed case reports are included in Supplemental Materials.

### Normal T and natural killer cell development in individuals with *CBL*-LOH

As *Cbl*-KO mice and *Cbl* Ub[LOF] KI mice display T cell defects[15,16,19], we studied T cell development in individuals with acquired, hematopoietic CBL deficiency. We assessed thymic output by quantifying signal joint TCR excision circles (sjTRECs) in DNA from peripheral blood by quantitative real-time PCR (qPCR). sjTREC numbers were moderately reduced (Fig. 2a), suggesting mildly impaired thymic output. Consistently, both CD4[+] and CD8[+] recent thymic emigrant T cells were lower in pediatric participants than in age-matched healthy donors (HDs) but were normal in adult participants P4 and P6 (Fig. 2b). Overall, absolute numbers of total CD3[+] T cells were slightly decreased (Extended Data Fig. 3a). However, this reduction in T cell output did not have a strong impact on mature T cell subsets in peripheral blood. Numbers and proportions of naive and memory CD4[+] T cell subsets were very similar to age-matched control individuals, whereas naive CD8[+] T cell numbers were slightly lower. Memory CD8[+] T cell subsets, however, were normal (Fig. 2c). Natural killer (NK) cells and innate-like (mucosal-associated invariant T, γδ T and invariant NK T) lymphocytes were in the range of HDs (Fig. 2c, bottom, and Extended Data Fig. 3a).

### Normal T cell function in individuals with *CBL*-LOH

Next, we tested T cell function. Survival and proliferation of CD4[+] and CD8[+] T cells from patients were indistinguishable from those of parental and nonfamilial HDs (Fig. 2d–f and Extended Data Fig. 3b). Polyclonal T cell lines, established with monoclonal anti-CD2/CD3/CD28 stimulation (T cell blasts), showed normal induction of STAT5 phosphorylation in response to interleukin-2 (IL-2) and IL-27 and normal production of interferon-γ (IFNγ) and tumor necrosis factor

(TNF) (Fig. 2g and Extended Data Fig. 3c). Production of IL-21, which functions as a potent growth and differentiation factor for human B cells[25], by *CBL*-LOH by circulating T follicular helper (cT$_{FH}$) cells was intact (Extended Data Fig. 3d,e), whereas production of IL-10 and IL-13 was moderately reduced and IFNγ production was slightly increased (Extended Data Fig. 3d,e), none of which reached statistical significance.

### CBL deficiency disrupts peripheral B cells

Next, we studied readouts of humoral immunity. Total B cell counts were above or at the upper limit of the healthy range in participants less than 15 years old. By contrast, B cell counts in participants greater than 15 years were below the healthy range (Fig. 3a), demonstrating progressive B cell lymphopenia. All tested individuals with *CBL*-LOH had polyclonal hypergammaglobulinemia, including adults with low numbers of blood B cells (Fig. 3b). The presence of transitional, naive, memory and CD19[hi]CD21[lo] B cells as well as plasmablasts was further investigated by mass cytometry of whole blood. We observed a striking elevation of transitional (10.4-fold) B cells, and milder but still highly significant increases in the numbers of naive (3.3-fold) and memory (2-fold) B cells, in pediatric, but not adult, participants (Fig. 3c). The counts of neither plasmablasts nor CD19[hi]CD21[lo] B cells, which have been implicated in myriad immune dysregulatory conditions[26], were affected by CBL deficiency (Extended Data Fig. 4a). Flow cytometric analyses revealed an approximately threefold increase in proportions of transitional B cells, whereas memory B cells were strongly and significantly underrepresented (reduced fourfold). Naive B cells were mildly but significantly reduced as a proportion of total B cells (Fig. 3d). Notably, the expanded population of transitional B cells was enriched for cells with a CD38[hi]CD5[hi]CD21[lo] phenotype (Fig. 3e), which corresponds to the least mature stage of transitional (T1) B cell development[27]. Although the T1 B cell subset comprises <15% of transitional B cells from HDs, these cells represented >50% of transitional B cells in patients (Fig. 3f). As human B cells develop from a transitional to a naive state, surface IgM (sIgM) is significantly downregulated[28]. Although sIgM expression was reduced on CBL-deficient naive B cells compared with CBL-deficient transitional B cells, overall IgM levels on CBL-deficient B cells remained elevated twofold (Extended Data Fig. 4b). In contrast to IgM, IgD expression was reduced on transitional and naive B cells, being statistically significant for naive B cells (more than twofold difference). Furthermore, while IgD increased as transitional B cells developed into naive B cells in HDs, this upregulation was not observed for CBL-deficient B cells (Extended Data Fig. 4b). Combined, these data indicate that CBL deficiency compromises B cell development at the transitional to naive stage.

### Elevated CD21[lo] B cells are bona fide transitional B cells

It is possible that the expanded population of T1 B cells within the transitional population corresponds to cells that have been termed 'atypical B cells', 'age-associated B cells', 'exhausted/anergic B cells' or 'CD21[lo]CD19[hi] B cells'[26]. To test this, we performed extended phenotypic analysis of transitional (CD10[+]CD27[−]) and CD21[lo]CD19[hi] B cells. Consistent with previous studies, transitional B cells in both HDs and individuals with CBL deficiency were consistently CD38[hi]CD21[lo]CD19[+]CXCR5[+]CXCR3[−]CD95[−]CD11c[−] (refs. 27–29), whereas CD21[lo]CD19[hi] B cells were CD38[lo]CD21[lo]CD19[hi]CXCR5[−]CXCR3[+]CD95[hi]CD11c[+] (refs. 30,31; Extended Data Fig. 4c). These phenotypic differences between B cell subsets in HDs and individuals deficient in CBL establish that the increase in CD21[lo] B cells in CBL deficiency reflects an expansion of transitional B cells, rather than an accumulation of CD21[lo]CD19[hi] anergic/exhausted/atypical-type B cells.

### Accumulation of immature B cells in the bone marrow

The B cell phenotype observed in individuals deficient in CBL is reminiscent of our previous findings for individuals with activated PI3Kδ

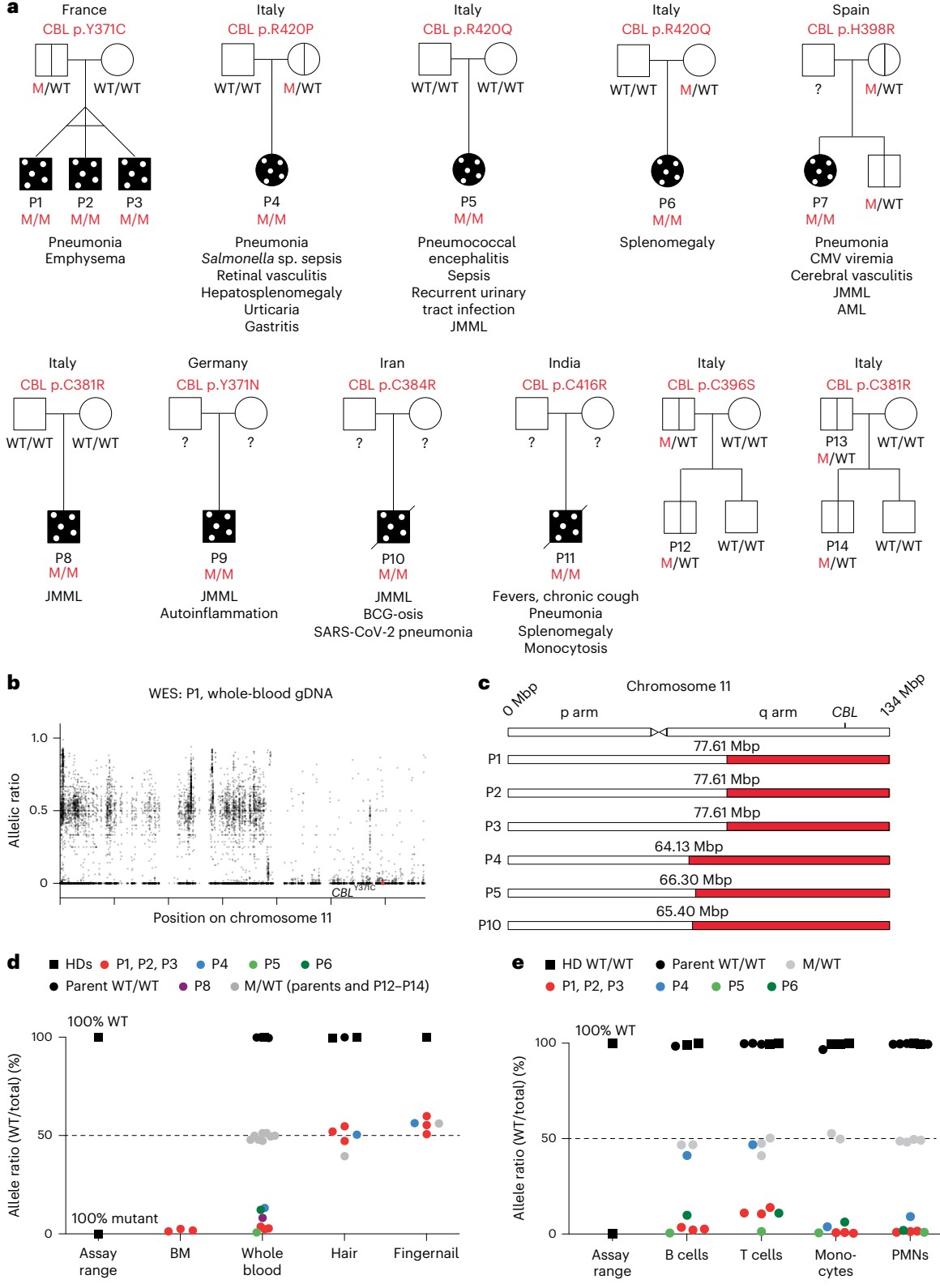

**Fig. 1 | Cohort of 11 individuals with *CBL*-LOH with leukemia, autoinflammation and infections. a**, Family pedigrees of individuals with *CBL*-LOH. The participants are shown in dotted black, indicating somatic mosaicism; black vertical line, asymptomatic heterozygotes; the question mark (?) indicates unknown genotype; M, mutant. **b**, Allelic ratio of variants on chromosome 11 in participant P1 as determined by WES from whole-blood genomic DNA (gDNA). The CBL p.Y371C variant is marked as a red dot. **c**, Schematic illustration

of the position of UPD on chromosome 11 in individuals for whom raw WES data were available. **d,e**, Quantitative genotyping by amplicon sequencing of patient tissues (**d**) and peripheral leukocyte subsets (**e**) targeting the relevant *CBL* variants. JMML, juvenile myelomonocytic leukemia; AML, acute myeloid leukemia; CMV, cytomegalovirus; BCG, Bacillus Calmette–Guerin; Mbp, megabase pairs; mut, mutant; PMNs, polymorphonuclear cells.

syndrome (APDS) due to gain-of-function (GOF) variants in *PIK3CD* who exhibit defects in B cell development in the bone marrow (BM)[29]. We thus explored B cell development in individuals with *CBL*-LOH. While progenitor cells corresponding to all stages of B cell development were detected in BM aspirates from P1, P2 and P3, proportions of cells at each of these stages differed from BM obtained from HDs. We specifically observed a reduction in stage I pre-B cells (pre-BI cells; $CD34^-CD19^+CD10^+CD20^-$) and a corresponding increase in immature ($CD34^-CD19^+CD10^+CD20^{++}$) B cells (Fig. 4a,b). These findings are also similar to individuals with APDS who have fewer pre-BI cells but increased immature B cells in their BM[29]. This suggests that the B cell lymphocytosis observed in individuals with *CBL*-LOH may be due to a block of B cell maturation in the BM.

## Leukocyte-intrinsic mechanism of impaired B cell development

We searched for evidence of a B cell-intrinsic defect that may underlie the aberrant B cell phenotype in participants. We tested the impact of *CBL*-LOH in an in vitro model of B cell development from $CD34^+$ hematopoietic stem/progenitor cells (HSPCs)[32,33]. We edited $CD34^+$ HSPCs isolated from HD cord blood samples with CRISPR–Cas9 to isogenically model CBL deficiency[33–35]. We used guide RNAs (gRNAs) to excise exon 8 of *CBL* (Extended Data Fig. 4d–i), which causes a small 44-amino-acid in-frame deletion and is a recurrent $Ub^{LOF}$ mutation in myeloid neoplasms[36]. We then assessed the ability of edited $CD34^+$ HSPCs to develop into B cells in vitro. Consistent with ex vivo staining of BM samples from individuals with CBL deficiency, we observed a similar accumulation of $CD19^+CD10^+CD20^{++}$ immature-type B cells in cultures seeded with *CBL* $Ub^{LOF}$-edited HSPCs compared with *AAVS1* control-edited HSPCs (Fig. 4c,d). To further investigate the impact of CBL deficiency on early B cell progenitors, we repeated this experiment and recapitulated a validated $PIK3CD^{GOF}$ (p.C416R[37]) allele using adenine base editing (Extended Data Fig. 4i). After sorting $CD19^+CD10^+CD20^{dim}$ B cell progenitors/precursors derived from $CD34^+$ HSPCs, RNA sequencing was conducted. This revealed significant activation of inflammatory pathways in *CBL* $Ub^{LOF}$-edited cells (Extended Data Fig. 4j–m), likely caused by secretion of proinflammatory cytokines by *CBL* $Ub^{LOF}$ myeloid cells present in the coculture system. *CBL* $Ub^{LOF}$ $CD10^+CD20^{dim}$ B cell progenitors/precursors showed increased G2–M checkpoint and mTORC1 signatures, suggestive of altered cell proliferation and enhanced PI3K pathway activation. More than half of the genes affected by *CBL* $Ub^{LOF}$ were shared with $PI3K^{GOF}$-edited B cell precursors, suggesting a substantial common axis of dysregulation between these two genotypes. To confirm this, we generated a set of genes differentially expressed by $PI3K^{GOF}$ compared with *AAVS1*-edited B cell precursors. Gene set enrichment analysis revealed that genes up- or downregulated in $PI3K^{GOF}$ cells were also significantly enriched or depleted, respectively, in *CBL* $Ub^{LOF}$ cells (Fig. 4e).

## Extrinsic B cell regulators are unaffected by CBL deficiency

We aimed to determine whether cell-extrinsic factors may contribute to immature B cell accumulation and memory B cell deficiency in individuals with *CBL*-LOH. Serum levels and production of BAFF and APRIL by cultured peripheral blood mononuclear cells (PBMCs) were intact (Extended Data Fig. 5a,b), while soluble CD40L (sCD40L) production by patient PBMCs and in patient plasma were moderately and significantly increased, respectively. Although elevated serum sCD40L has been reported in systemic lupus erythematosus and other autoimmune diseases[38], there is no known association with impaired B cell development. Consistently, the addition of sCD40L had no effect on B cell differentiation from HD $CD34^+$ HSPCs in vitro (Extended Data Fig. 5c–f). Next, we tested the capacity of patient naive and memory $CD4^+$ T cells to produce IL-21. Similar to $cT_{FH}$ cells, IL-21 production was intact in expanded cultures of naive and memory $CD4^+$ T cells from patients relative to those from HDs (Extended Data Fig. 5g).

## B cell-autonomous defect of maturation

We hypothesized that the block in B cell maturation and differentiation is cell autonomous due to a requisite intrinsic function of CBL. As the patients are mosaic for the UPD that renders cells homozygous for a *CBL* mutation, we aimed to detect differences in maturation between B cells with or without LOH. To this end, we sorted and genotyped B cell subsets and monocytes. Monocytes from individuals who carried relatively high burdens of the variant *CBL* allele (>95%; Fig. 4f). In participants with UPD, we observed consistent and marked differences in allele burden across B cell subsets. Transitional B cells had the highest variant allele frequency, with only ~0.7% of these alleles being wild-type (WT). As each cell harbors two alleles, cells in these patients are either heterozygous or homozygous; therefore, ~1.4% of transitional B cells carry WT *CBL* alleles. However, there was a 5-fold increase in the frequency of naive B cells (7.2%) and a nearly 20-fold increase in memory B cells (26.2%) carrying the WT *CBL* allele. Therefore, although almost all transitional B cells are homozygous for *CBL* $Ub^{LOF}$ alleles, the few transitional B cells harboring the WT *CBL* allele have a marked advantage at becoming naive and subsequently memory B cells. Like a chimeric BM model, this shared environment isolates the causal effect of the *CBL* genotype.

## Impaired apoptosis in CBL-deficient immature B cells

We hypothesized that immature B cells may accumulate during hematopoiesis in CBL deficiency due to impaired apoptosis. One trigger of apoptosis in immature B cells is dimerization of the ectoenzyme CD38 (ref. 39). CD38 directly interacts with CBL[40], and CD38 stimulation activates PI3K and CBL[41]. Transitional and naive B cells from individuals with *CBL*-LOH expressed elevated levels of surface CD38 (Fig. 4g), which is also observed in $PIK3CD^{GOF}$ B cells that have elevated constitutive activation of PI3K[29]. We therefore tested CD38-mediated apoptosis in immature B cells. CD38 expression was comparable on control and CBL-edited HSPC-derived $CD19^+$ B cells (Extended Data Fig. 5h). When

---

**Fig. 2 | Intact T cell development and function in individuals with *CBL*-LOH.** **a**, sjTREC quantification in HDs (black dots), heterozygous donors (gray dots) and individuals with *CBL*-LOH (colored dots), as determined by qPCR of whole-blood DNA; WBCs, white blood cells. **b**, Recent thymic emigrant $CD4^+$ and $CD8^+$ T cells quantified in peripheral fresh blood by mass cytometry and gating of $CD31^+$ cells among naive T cells; data are shown as mean ± s.d. The statistical significance of differences was assessed by multiple two-sided Mann–Whitney tests, with correction for multiple testing; *$P < 0.05$ and **$P < 0.005$. **c**, Quantification of the indicated T cell subsets in the peripheral blood of HDs, heterozygous HDs and individuals with *CBL*-LOH of the indicated ages as determined by mass cytometry; data are shown as mean ± s.d. The statistical significance of differences was assessed by multiple two-sided Mann–Whitney tests, with correction for multiple testing; *$P < 0.05$ and **$P < 0.005$. In **b** and **c**, controls 0–3 years old ($n = 2$), controls 4–15 years old ($n = 9$), controls 16–100 years old ($n = 28$), pediatric participants (LOH) ($n = 5$), adult participants (LOH) ($n = 2$) and heterozygous individuals ($n = 3$). **d**, Percentage of dead cells in cultures of activated fresh PBMCs from HDs, the heterozygous father and participants (P1–P3) after 5 days of TCR stimulation, as determined by dead cell marker staining and flow cytometry; $n = 3$ HDs and $n = 3$ patient; data are shown as mean ± s.d. **e,f**, Cell division index of $CD4^+$ (**e**) and $CD8^+$ (**f**) T cells of HDs, the heterozygous father and participants (P1–P3) after 5 days of the indicated TCR stimulation, as determined by dilution of CFSE; $n = 3$ HDs and $n = 3$ patients; data are shown as mean ± s.d. **g**, Cytokine response by STAT5 phosphorylation (left) and cytokine production (right) by T cell blasts that are homozygous (red), heterozygous (gray) and homozygous WT for *CBL* $Ub^{LOF}$ variants following the indicated stimuli. The bars show the mean of the displayed data points (one for each T blast line); NS, not significant; TEMRA, terminally differentiated effector memory T cells; MAIT, mucosal-associated invariant T cells; MFI, median fluorescence intensity; PHA, phytohemagglutinin; PMA, phorbol 12-myristate 13-acetate.

these cells were stimulated by cross-linking CD38, the frequency of early apoptotic (Annexin V$^+$) cells was substantially increased in control, but not *CBL* Ub$^{LOF}$-edited, cells (Fig. 4h). Similarly, KI of the p.Y371C missense Ub$^{LOF}$ variant into the endogenous *CBL* locus of the REH B leukemia cell line[42], which expresses high levels of CD38 (Extended Data Fig. 5i), revealed that WT REH cells underwent significantly more apoptosis than *CBL*$^{Y371C}$ KI cells (Fig. 4i). *CBL*$^{Y371C}$ KI BJAB cells were not resistant

to apoptosis through BCR cross-linking (Extended Data Fig. 5j), attesting to the specificity of this phenotype. Consistent with altered CD38 signaling, we observed elevated pERK in *CBL*$^{Y371C}$ KI cells following CD38 stimulation at early and late time points compared with the parental WT cell line (Fig. 4j and Extended Data Fig. 6a–c). The AKT pathway, downstream of PI3K, exhibited increased activity only at later time points (Extended Data Fig. 6d–f). Consistently, ERK phosphorylation

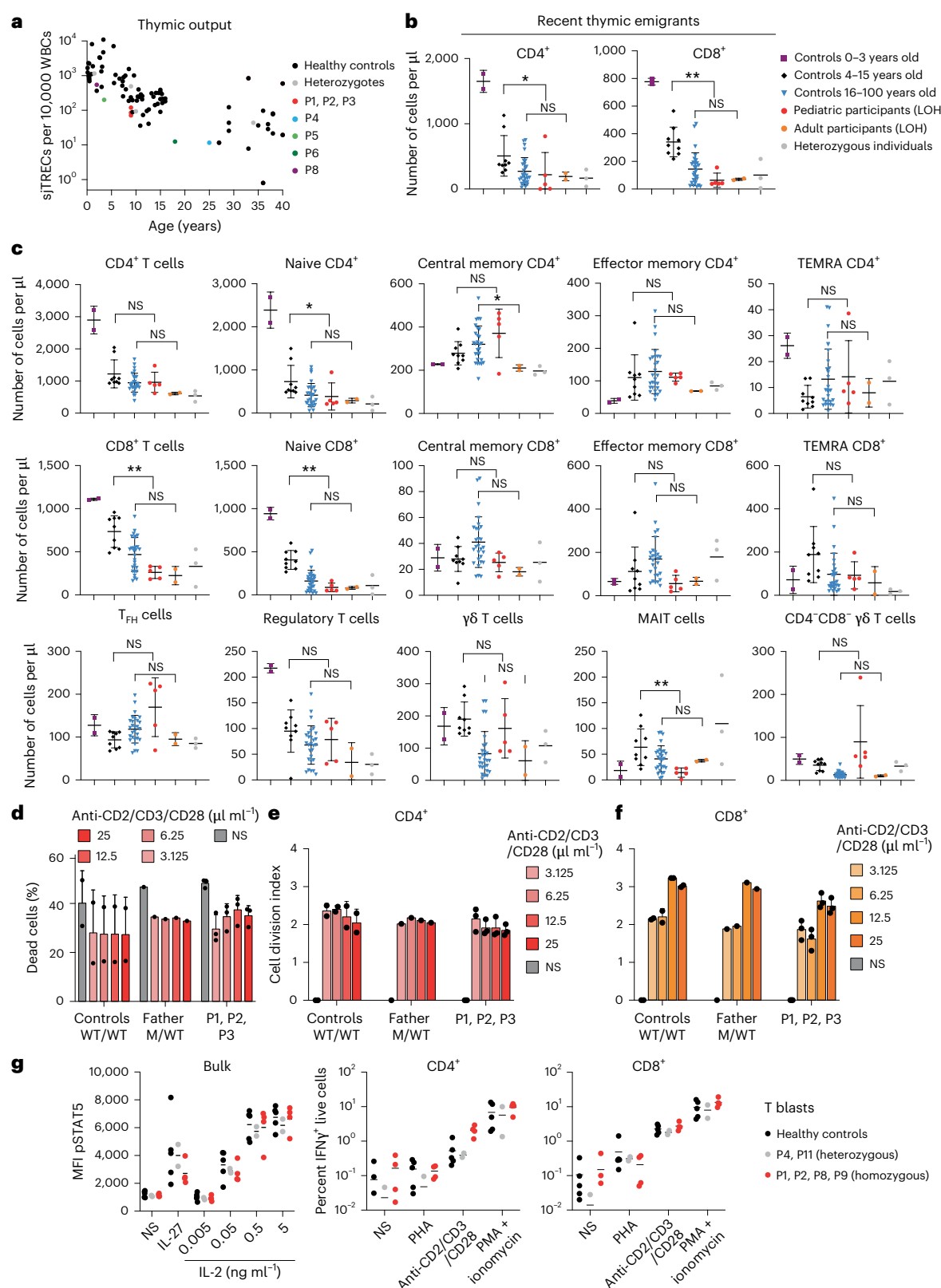

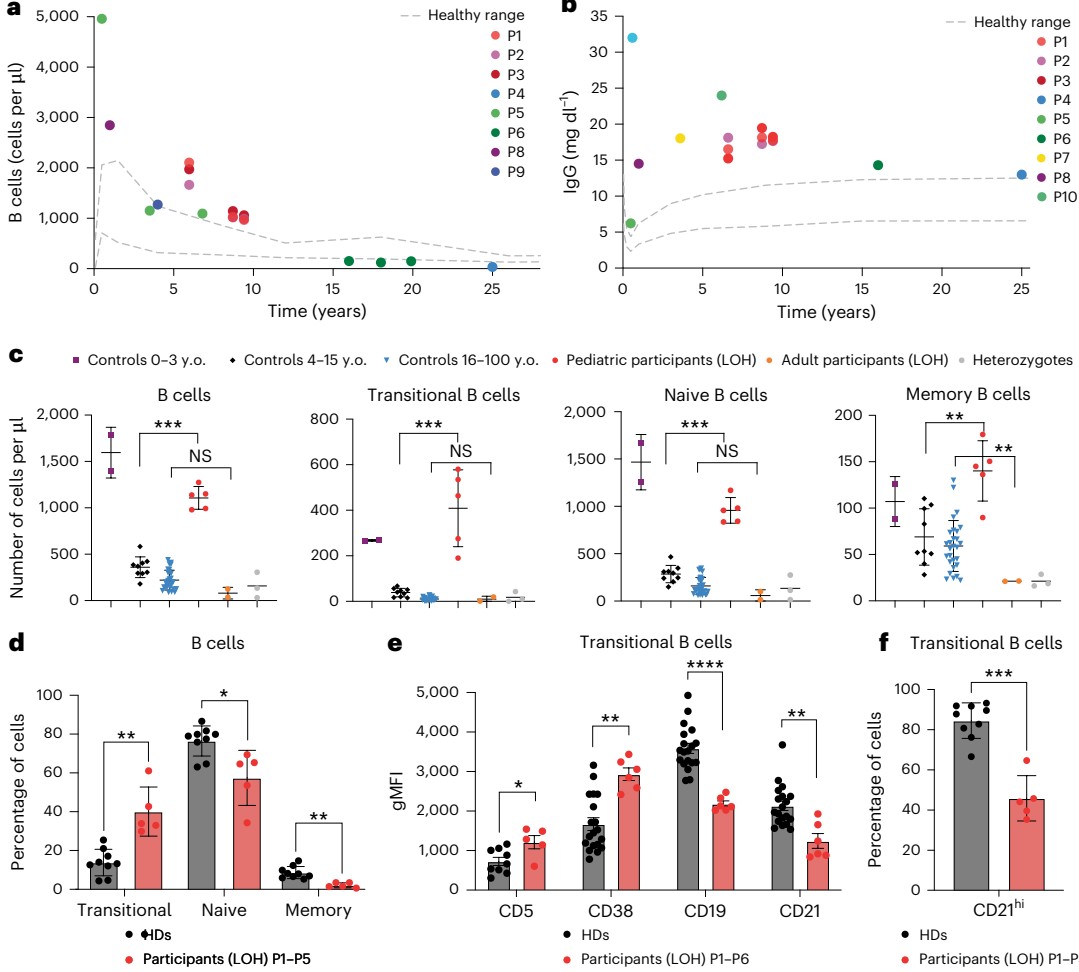

**Fig. 3 | Dysregulated B cell development in individuals with *CBL*-LOH. a**, Counts of peripheral B cells in individuals with *CBL*-LOH over time compared with the healthy range. Data from eight participants are shown. Healthy ranges are from pediatric clinical recommendations. **b**, IgG levels in individuals with *CBL*-LOH (P1–P8) over time compared with the healthy range. Data from eight participants are shown. Healthy ranges are from pediatric clinical recommendations. **c**, Quantification of the indicated B cell subsets in the peripheral cryopreserved mononuclear cell blood of HDs, heterozygous HDs and individuals with *CBL*-LOH of the indicated ages as determined by mass cytometry; controls 0–3 years old (*n* = 2), controls 4–15 years old (*n* = 9), controls 16–100 years old (*n* = 28), pediatric participants (LOH) (*n* = 5), adult participants (LOH) (*n* = 2), heterozygous individuals (*n* = 3). Data are shown as mean ± s.d. The statistical significance of differences was assessed in multiple two-sided Mann–Whitney tests, with correction for multiple testing; **P* < 0.005 and ***P* < 0.0005. y.o., years old.

**d**, Frequency of B cell subsets in cryopreserved PBMCs from HDs and individuals with *CBL*-LOH as determined by flow cytometry; HDs (*n* = 13), individuals with *CBL*-LOH (*n* = 5). Data are shown as mean ± s.d. The statistical significance of differences was assessed using multiple two-sided Mann–Whitney tests, with correction for multiple testing; **P* < 0.05 and ***P* < 0.005. **e**, CD5, CD9, CD21 and CD38 expression on transitional B cells of HDs (black) and individuals with *CBL*-LOH (red) as determined by flow cytometry; HDs (*n* = 19), individuals with *CBL*-LOH (*n* = 6). Data are shown as mean ± s.d. The statistical significance of differences was assessed by multiple two-sided Mann–Whitney tests, with correction for multiple testing; **P* < 0.05, ***P* < 0.005 and *****P* < 0.00005. **f**, Percentage of CD21^hi cells among transitional B cells of HDs (black) and individuals with *CBL*-LOH (red) as determined by flow cytometry; HDs (*n* = 9), individuals with *CBL*-LOH (*n* = 5). Data are shown as mean ± s.d. The statistical significance of differences was assessed by Mann–Whitney test; ****P* < 0.0005; gMFI, geometric mean fluorescence intensity.

was increased in *CBL* Ub^LOF HSPC-derived B cell progenitors following CD38 stimulation (Extended Data Fig. 6g,h).

### Impaired immunoglobulin secretion by CBL-deficient mature B cells ex vivo

We hypothesized that mature, CBL-deficient B cells may also be dysfunctional in the context of responding to diverse antigenic stimuli. Inherited defects of B cells and humoral immunity underlie susceptibility to bacterial infections[43], similar to those observed in participants with *CBL*-LOH. Sorted B cell subsets were cultured for 7 days under cytokine-dependent or cytokine-independent stimulation to assess IgM, IgA and IgG secretion. As previously reported[27], transitional B cells produced lower amounts of IgM than naive B cells (Extended Data Fig. 7a,b), and both produced IgA and IgG, albeit lower amounts, exclusively following CD40L/IL-21 stimulation

(Extended Data Fig. 7b). All CBL-deficient B cell subsets produced substantially less Ig than cells from HDs under one or both in vitro culture conditions (Fig. 5a). *CBL*-LOH B cells showed weaker responses to cytokine-independent stimuli, whereas responses to CD40L/IL-21 were less, although still significantly, affected (Fig. 5a). The defect in memory B cells was less marked than in naive and transitional B cells (Fig. 5a); this may reflect enrichment of memory cells heterozygous for the *CBL* Ub^LOF variants. BM plasma cells (CD38^hiCD27^hiCD20^lo) from P2 and P3 showed reduced IgA and absent IgG secretion (Fig. 5b).

### B cell-autonomous defect in BCR signaling and immunoglobulin secretion

To substantiate these observations of impaired human B cell development and function due to *CBL* LOF, we generated *CBL*-KO BJAB (IgM^+ B lymphoma cell line) clones (Extended Data Fig. 7c). These clones

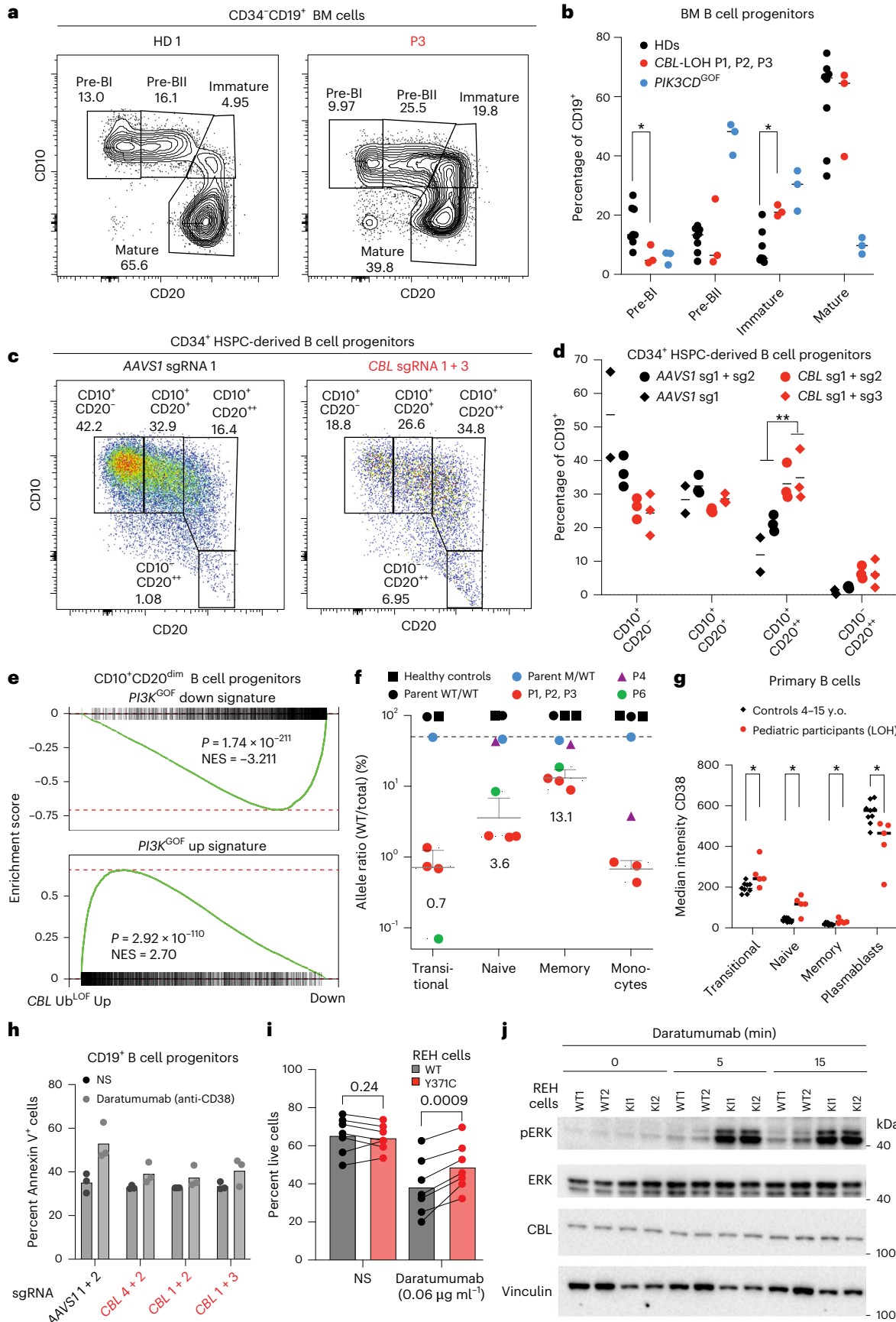

**Fig. 4 | Cell-autonomous defect in B cell maturation in *CBL* Ub^LOF cells.**
**a,b**, Defective B cell maturation in the BM of individuals with *CBL*-LOH. Flow cytometry staining of cryopreserved BM mononuclear cells of HDs and participants P1, P2 and P3 with *CBL*-LOH. **a**, Representative flow staining of CD20 versus CD10 expression levels on CD34⁻CD19⁺ cells in BM samples. **b**, Quantification of these subsets for HDs (*n* = 8), individuals with *CBL*-LOH (*n* = 3) and individuals with *PIK3CD*^GOF (*n* = 3). The line shows the mean of the data points. Statistical significance was assessed using multiple two-sided Mann–Whitney tests corrected for multiple testing; *\**P* < 0.05. **c,d**, In vitro differentiation of control *AAVS1*-edited and *CBL*-edited CD34⁺ HSPCs toward B cell identity. **c**, Flow cytometry staining for CD10 and CD20 among CD19⁺ cells in differentiation cultures after 3 weeks of coculture. **d**, Quantification of B cell 'subsets' based on flow cytometry marker expression (pre-BI, CD10⁺CD20⁻; pre-BII, CD10⁺CD20⁺; immature B, CD10⁺CD20⁺⁺; mature B, CD10⁻CD20⁺⁺) in this culture in control and two *CBL*-edited reactions. The lines show the means of three biological replicates, except for *AAVS1* single guide RNA (sgRNA) 1, where two replicates are shown. Statistical significance was assessed using multiple two-sided Mann–Whitney tests corrected for multiple testing; *\*\**P* < 0.005. **e**, Transcriptional overlap between *CBL*-edited and *PI3K*^GOF HSPC-derived B cell

progenitors. Gene set enrichment analysis for *PI3K*^GOF gene signatures in *CBL* Ub^LOF samples is shown; NES, normalized enrichment score. No correction for multiple testing was performed for the two binomial tests. **f**, Quantitative genotyping by amplicon sequencing of B cell subsets and monocytes in individuals with *CBL*-LOH and HDs, as well as parents of P1–P3; HDs (*n* = 3), individuals with *CBL*-LOH (*n* = 5). Data are shown as mean ± s.d. **g**, CD38 staining intensity of primary B cell subsets of pediatric individuals with *CBL*-LOH compared with age-matched HDs; HDs (*n* = 9), individuals with *CBL*-LOH (*n* = 5). Data are shown as mean ± s.d. The statistical significance of differences was assessed by multiple two-sided Mann–Whitney tests, with correction for multiple testing; *\**P* < 0.05. **h**, Rate of apoptosis following stimulation with daratumumab of CD19⁺ cells from in vitro differentiation cultures of control and *CBL*-edited HSPCs. Data show the mean of three technical replicates. The experiment is representative of three biological replicates. **i**, Rate of apoptosis following stimulation with daratumumab of control and *CBL*^Y371C KI REH cells. Each dot represents one biological replicate; *n* = 7. The statistical significance of differences was assessed using multiple paired, two-sided *t*-tests, with correction for multiple testing. **j**, Western blot of control and *CBL*^Y371C KI REH cells following stimulation with monoclonal anti-CD38 (daratumumab) for the indicated times (min).

produced substantially lower amounts of IgM than WT BJAB cells (Fig. 5c), which was rescued by re-expression of WT *CBL* (Fig. 5c and Extended Data Fig. 7a). Furthermore, KI of the *CBL*^Y371C Ub^LOF variant (Extended Data Fig. 7d) resulted in a comparable reduction in IgM production as *CBL* KO in BJAB clones (Extended Data Fig. 7e). ERK phosphorylation, as a readout of ERK signaling, was reduced in *CBL*^Y371C BJAB cells compared with WT BJAB cells following BCR stimulation (Extended Data Fig. 7f).

## Inflammation underlies hypergammaglobulinemia in individuals with *CBL*-LOH

We aimed to reconcile the observation of hypergammaglobulinemia in all individuals with reduced Ig production by primary B cells and isogenic cell lines (Fig. 5a–e). We hypothesized that chronic inflammation driven by monocytes from individuals with *CBL*-LOH may induce B cell activation and Ig overproduction[23]. Thus, we compared RNA-sequencing data from primary B cells obtained directly from cryopreserved PBMCs to data from B cells that were cultured for 24 h without stimulation. Compared with B cells from healthy pediatric donors, B cells from individuals with *CBL*-LOH showed substantial and broad inflammatory pathway overactivation when sequenced directly; however, these transcriptional differences disappeared after 24 h of culture (Extended Data Fig. 7g). This suggests that *CBL*-LOH B cells are primed by the inflammatory environment they originated from. To investigate the consequence of this priming on Ig production, we isolated primary B cells from fresh blood samples of individuals with *CBL*-LOH and cultured them for 24 h. Analysis of Ig production revealed significant increases in IgG1, IgG3 and IgM by B cells from individuals with *CBL*-LOH compared with those from HDs (Extended Data Fig. 7h).

Finally, we tested whether the hypersecretory phenotype could be conferred by the inflammatory environment created by monocytes from individuals with *CBL*-LOH. Isolated CD14⁺ monocytes from individuals with *CBL*-LOH and HDs were cultured for 24 h without stimulation. Supernatants derived from cultured *CBL*-LOH monocytes induced significantly increased IgG1 and IgM production by HD B cells compared with supernatants derived from HD monocytes (Fig. 5e).

## CBL deficiency disrupts formation of long-lived antigen-specific B cells

Considering these in vitro and ex vivo B cell defects, we further explored memory B cells in CBL deficiency. First, we determined proportions of memory (CD19⁺CD20⁺CD27⁺) B cells that had undergone Ig isotype switching in vivo. In CBL-deficient individuals, the proportions of all memory B cells were decreased (Fig. 3d) as well as the proportion of Ig-class-switched memory B cells relative to that observed in HDs (Fig. 5f). Second, by using tetramers of SARS-CoV-2 spike protein, we quantified frequencies of antigen-specific B cells at different times following vaccination. Spike-binding B cells were detected in peripheral blood of HDs and CBL-deficient individuals 4–10 weeks after receiving 2 doses of a SARS-CoV-2 mRNA vaccine (Fig. 5g and Extended Data Fig. 8a). In HDs, proportions of SARS-CoV-2-specific B cells increased approximately threefold 6–12 months after vaccination and then declined to levels similar to those observed at earlier times. By contrast, proportions of spike-binding B cells detected in the same individuals deficient in CBL assayed 12 and 20–24 months after vaccination were unchanged or reduced compared with the earlier time point (Fig. 5g and Extended Data Fig. 5f). In addition to this proportionate decrease in total spike-binding B cells, significantly

**Fig. 5 | Cell-autonomous defect in B cell function of mature *CBL* Ub^LOF B cells.**
**a**, Ig production by sorted primary B cell subsets from HDs and individuals with *CBL*-LOH from cryopreserved PBMCs after the indicated stimulations. Supernatants were collected after 5–7 days, and Ig levels were measured by enzyme-linked immunosorbent assay (ELISA). The line shows the mean of the displayed data points (one per individual); HDs (*n* = 13), participants with *CBL*-LOH (*n* = 5). The statistical significance of differences was assessed using multiple two-sided unpaired *t*-tests, with correction for multiple testing; *\**P* < 0.05, *\*\**P* < 0.005 and *\*\*\**P* < 0.0005. **b,c**, Ig production by plasma cells sorted from cryopreserved BM mononuclear cells of participants P1 and P2 with *CBL*-LOH and HDs (*n* = 2; **b**) and IgM production of WT, *CBL*-KO and rescue BJAB cell lines within 24 h of culture (**c**); data are shown as mean ± s.d. of *n* = 5 independent biological replicates. **d**, Western blot of WT and *CBL*^Y371C KI BJAB cells following BCR stimulation with monoclonal anti-IgM for the indicated time periods. Data are representative of three biological replicates. **e**, Primary monocytes from

HDs (*n* = 15) and participants with *CBL*-LOH (*n* = 6) were sorted from fresh blood samples. After 24 h of nonstimulated culture, supernatants were collected, and B cells from HDs were stimulated with the supernatants for 24 h. Ig production was assessed by ELISA. Data are shown as mean ± s.d. Statistical significance was assessed using multiple two-sided Mann–Whitney tests adjusted for multiple testing; *\*\**P* < 0.005 and *\*\*\**P* < 0.0005. **f**, Frequency of IgA⁺ and IgG⁺ B cells among memory B cells in patients (*n* = 3) and HDs (*n* = 13), as determined by flow cytometry. Statistical significance was assessed by unpaired, two-sided *t*-tests; *\*\**P* < 0.005 and *\*\*\**P* < 0.0005. **g**, Frequency of spike⁺ B cells in patients (*n* = 4) and HDs (*n* = 10), as determined by flow cytometry with tetramer staining. Data are shown as mean ± s.d. Statistical significance was assessed by unpaired, two-sided *t*-tests; *\**P* < 0.05 and *\*\**P* < 0.005. **h**, Frequency of IgG⁺, IgA⁺ and IgG⁻IgA⁻ B cells among spike⁺ B cells in patients (*n* = 4) and HDs (*n* = 6). Error bars indicate s.e.m. Statistical significance was assessed by unpaired, two-sided *t*-tests; *\**P* < 0.05; EV, empty vector; Vinc, vinculin; wk, weeks; mo, months; yrs, years.

fewer SARS-CoV-2-specific B cells in CBL-deficient individuals (~30–50%) underwent Ig class switching to express IgG relative to HDs (~70–85%; Fig. 5h).

## Defects in immunoglobulin gene usage and somatic hypermutation

To further understand the impact of CBL deficiency on B cell development and differentiation, we analyzed the BCR repertoire of transitional,

naive and memory B cells from five individuals deficient in CBL. This revealed unequivocal differences between CBL-deficient individuals and HDs. There was increased usage of the *IGHV4-34* gene element in transitional, naive and IgM⁺ memory B cells from CBL-deficient individuals (Fig. 6a) and significantly reduced usage of *IGHJ6* and increased usage of *IGHJ4* genes by CBL-deficient transitional and naive B cells (Fig. 6b). CDR3 lengths of Ig expressed by transitional and naive B cells were shorter for CBL-deficient individuals than in HDs (Fig. 6c).

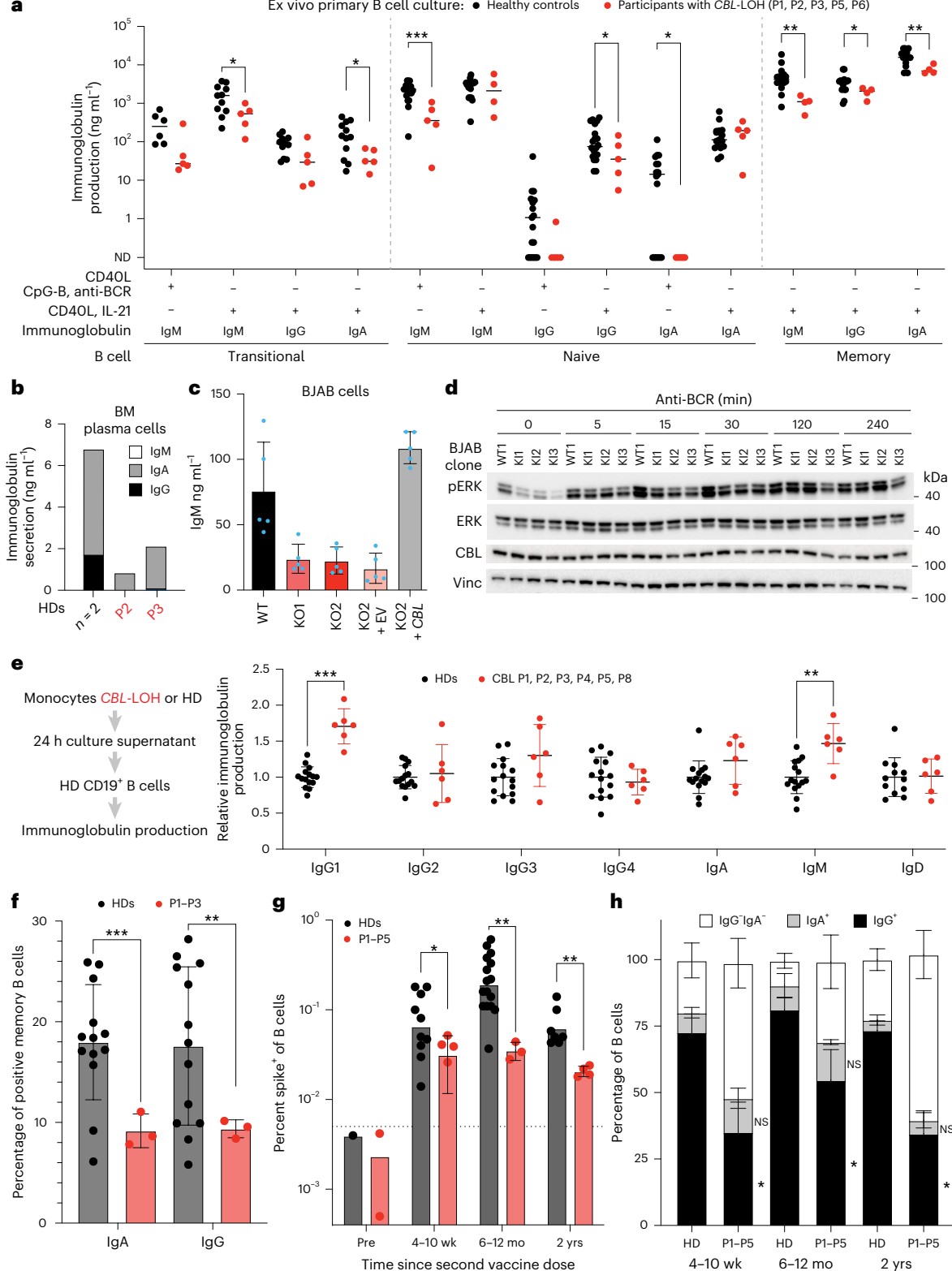

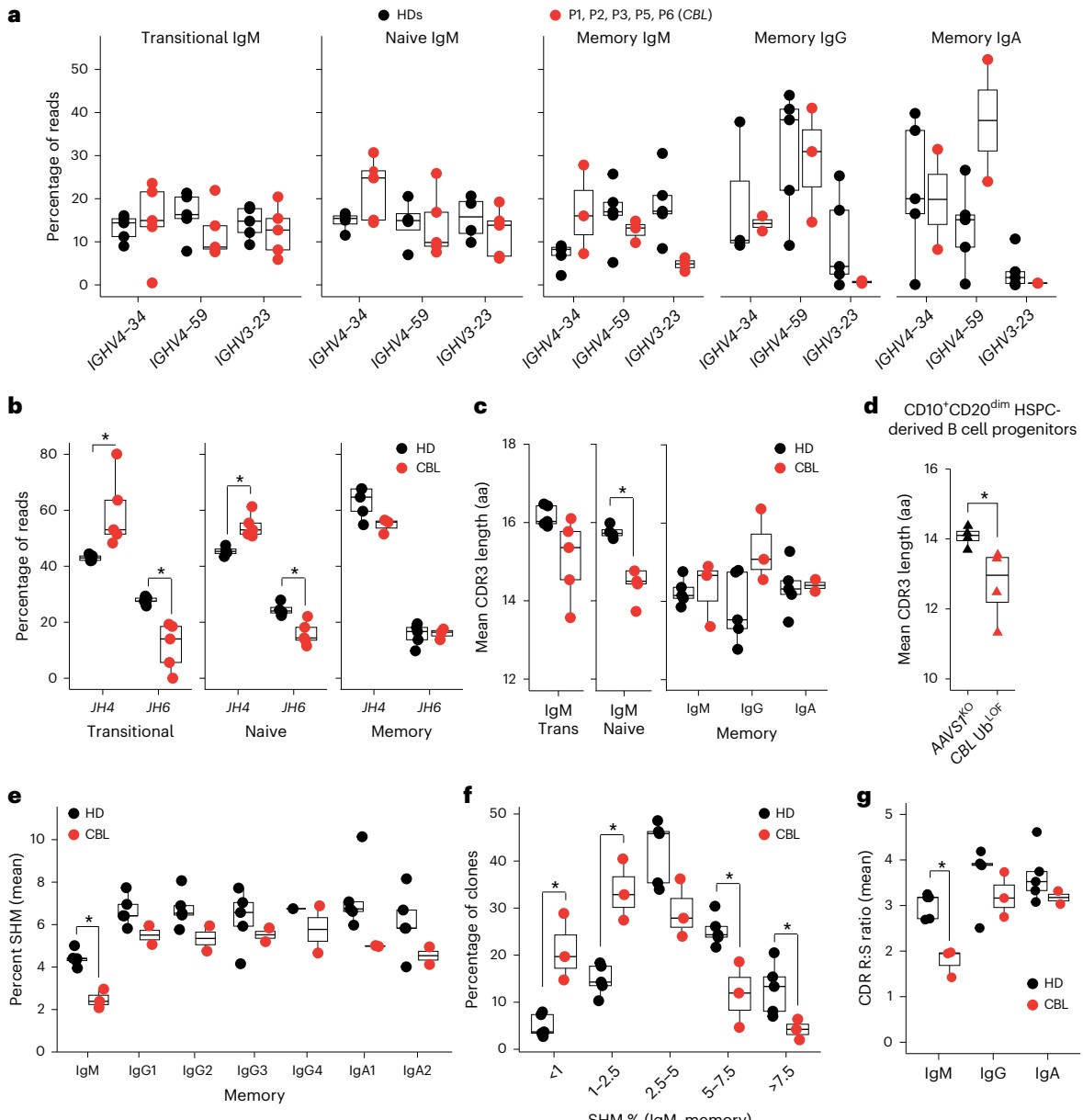

**Fig. 6 | BCR repertoire of individuals deficient in CBL reveals a defect in immunoglobulin V gene usage and somatic hypermutation. a,b,** Usage of the top three IGHV *IGHV4-34*, *IGHV4-59* and *IGHV3-23* gene elements (**a**) and Ig *JH4* and *JH6* elements (**b**) in transitional, naive and memory B cells isolated from HDs and the indicated individuals deficient in CBL. **c,d,** CDR3 lengths in transitional, naive and memory cells isolated from HDs and individuals deficient in CBL (**c**) or CD10+CD20dim HSPC-derived B cell progenitors edited at the *AAVS1* or *CBL* locus (**d**). aa, amino acids; Trans, transitional. **e,** Frequency of Ig somatic hypermutations (SHM) in memory B cells defined by the expression of distinct class-switched Ig isotypes. **f,** Frequency of clones with different levels of somatic hypermutation within IgM+ memory B cells. **g,** CDR replacement:silent (R:S) ratios in IgM, IgG and IgA memory B cells. Statistical significance was assessed with a Wilcoxon test with Bonferroni correction for multiple testing (if needed; **b**–**d** and **g**) or Dunn's test for multiple comparisons (**e** and **f**); *$P < 0.05$. Data shown were generated from BCR sequencing of HDs ($n = 5$) and individuals with *CBL*-LOH ($n = 5$). Boxes and whiskers indicate the median (center line), quartiles (box) and data range within 1.5× interquartile range (whiskers), and dots show data values beyond 1.5× interquartile range. All individual data points are shown.

This likely results from decreased *IGHJ6* usage (Fig. 6b), as this gene element contributes the highest number of amino acids to Ig CDR3 regions[44]. To solidify these findings, we assessed BCR rearrangements from the bulk RNA-sequencing data of HSPC-derived B cell progenitors. This also revealed significantly reduced CDR3 lengths in *CBL* Ub^LOF CD10+CD20dim B cell progenitors (Fig. 6d), strongly suggesting that aberrations to the BCR repertoire of *CBL*-LOH B cells is caused by a cell-intrinsic process during *IGH* rearrangement. Furthermore, our analysis of BCR repertoires revealed significantly reduced levels of somatic hypermutation in CBL-deficient IgM+ memory B cells (Fig. 6e). Indeed, when mutation load was quantified in terms of percentiles, most clones

from IgM+ memory B cells from HDs exhibited a mutation rate of 2.5 to >7.5%, whereas most clones from CBL-deficient IgM+ memory B cells had accumulated mutations at a rate of <2.5% (Fig. 6f). Last, mutational targeting and selection, as determined by replacement:silent ratios, was also significantly reduced in CBL-deficient IgM+ memory B cells (Fig. 6g).

## *CBL*-LOH causes a break in B cell tolerance
The level and dynamics of BCR signaling are tightly regulated to control B cell differentiation and enforce self-tolerance[45]. As individuals with *CBL*-LOH were frequently positive for autoantibodies[23],

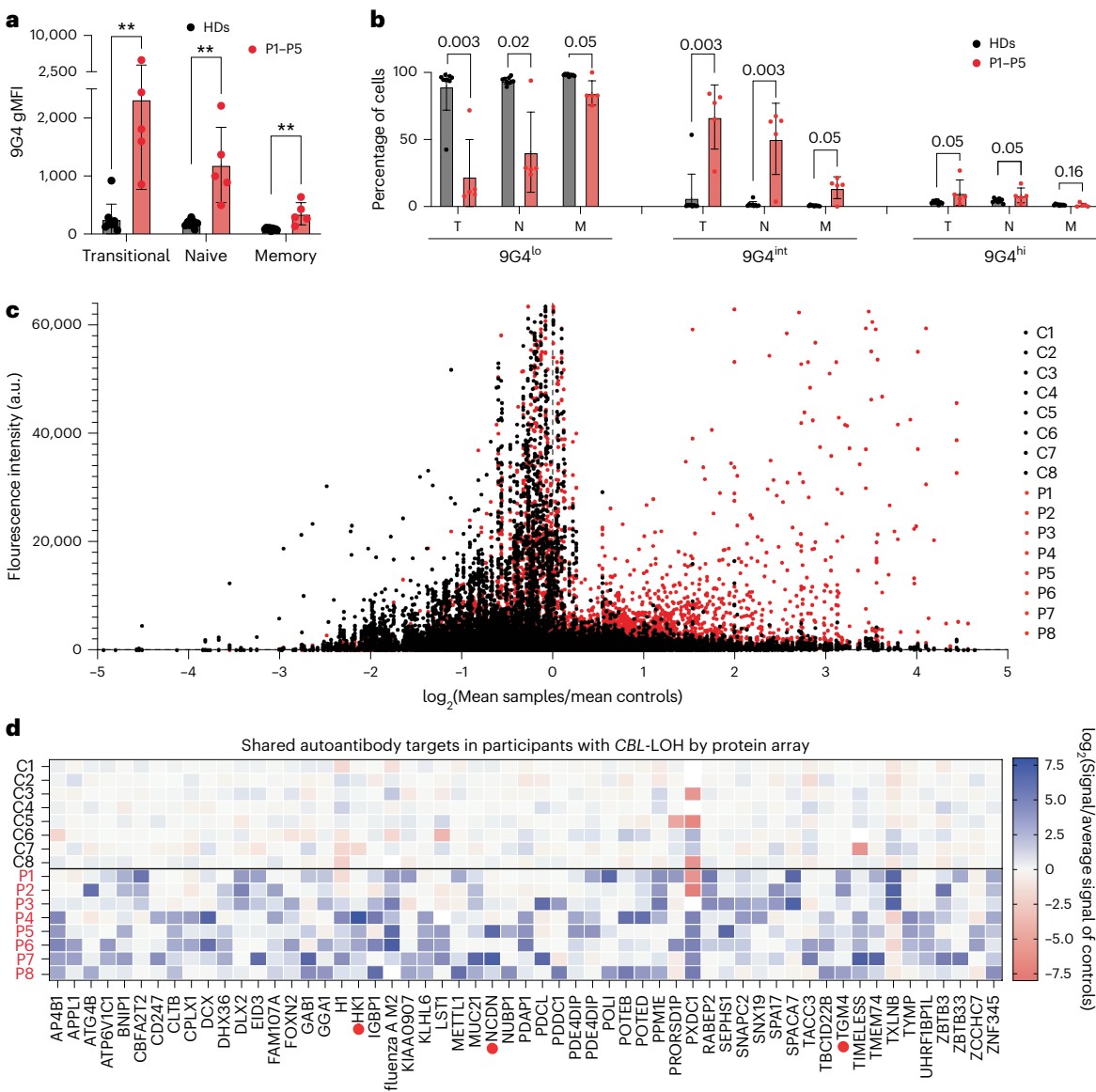

**Fig. 7 | Autoimmunity in individuals with *CBL*-LOH. a,b,** 9G4 staining of primary B cells from cryopreserved PBMCs from HDs and individuals with *CBL*-LOH (P1–P5) by flow cytometry. **a,** Median fluorescence intensity of the indicated B cell subsets in HDs (*n* = 9) and individuals with *CBL*-LOH (*n* = 5). Data are shown as mean ± s.d. Statistical significance was assessed using multiple two-sided Mann–Whitney tests adjusted for multiple testing; **P* < 0.005. **b,** Frequency of 9G4lo, 9G4int and 9G4hi cells among transitional (T), naive (N) and memory (M) B cells in HDs (*n* = 9) and individuals with *CBL*-LOH (*n* = 5). Data are shown as mean ± s.d. Statistical significance was assessed with multiple two-sided Mann–Whitney tests adjusted for multiple testing. **c,d,** Human protein microarray autoantibody

detection. **c,** Protein microarray fluorescence intensity. The ratio of values for plasma from HDs (*n* = 8) to those from samples from individuals with *CBL*-LOH (*n* = 8) is shown. Data are shown as mean values from pairs of duplicates. One protein microarray was used per individual, and the results have been normalized to account for interexperiment variation. Fluorescence intensity is expressed in arbitrary units (a.u.). **d,** Reactivities for the indicated autoantigens common to P1–P8 and absent from eight age-matched control individuals. Serum samples from participants and blood donor controls were screened for IgG reactivity to 20,000 full-length human proteins on microarrays (HuProt). Red dots indicate autoantibodies previously shown to be associated with a clinical condition.

we asked whether dysregulation of *CBL* UbLOF B cells may cause defects in these processes. To test this, and extend our findings of increased *IGHV4-34* usage, we stained B cells from HDs and CBL-deficient individuals with the idiotypic monoclonal antibody 9G4. 9G4 detects Ig molecules encoded by the *IGHV4-34* Ig heavy chain gene[46], which are almost exclusively self-reactive[47,48]. In HDs, 5–10% of naive B cells expressing IGHV4-34 BCRs are stained at high levels by 9G4, but very few of these cells secrete antibodies[48–50]. When 9G4+ Abs are secreted, they bind self-glycans on circulating transitional and naive B cells that do not themselves express IGHV4-34 BCRs, 'painting' the B cells to create a 9G4int population[50–52]. Approximately 65% of transitional, ~50% of naive and ~15% of memory B cells from CBL-deficient

individuals were 9G4int, compared with 6%, 1.6% and 0.7% of these cell subsets, respectively, in HDs (Fig. 7a,b). As *IGHV4-34* only accounted for 15–30% of *IGHM* mRNA in transitional or naive B cells from individuals with *CBL*-LOH, this suggests that autoreactive antibodies to IGHV4-34 in plasma 'paint' epitopes on the surface of these cells. Consistently, we observed this painting effect on B cells from HDs that had been incubated with plasma from individuals deficient in CBL (Extended Data Fig. 8b). Further, we profiled the plasma of eight individuals for the presence of IgG specific for 20,000 human proteins. All participants tested showed a striking increase in reactivity to human antigens compared with eight sex-matched adult controls (Fig. 7c). We found several strong candidate targets that, strikingly,

were detected in sera from unrelated individuals (Fig. 7d). We validated the presence of autoantibodies to TXLNB in plasma from P1, P2 and P3 through multiplex bead assays (Extended Data Fig. 8c).

## Discussion

CBL is an E3 ubiquitin ligase with a well-characterized function in mouse T cell selection and differentiation[15,16,18,19]. CBL is dispensable for mouse B cell development and function. We studied individuals with leukocyte-specific somatic loss of CBL activity to unravel non-redundant roles of CBL in human lymphocytes. In stark contrast to mice, we demonstrated that CBL is largely redundant for human T cell development and function but has fundamental, nonredundant roles at several critical stages during human B cell development, selection, maturation and differentiation.

By combining patient-derived and engineered cell models, we found that loss of CBL impairs B cell maturation and tolerance, causing immature BM B cell accumulation and excess peripheral transitional (T1) cells. Previous studies demonstrated that early transitional B cells are enriched with autoreactive BCRs[27,50]. Binding to self-antigens downregulates sIgM on mouse transitional and mature B cells[45,53,54] and on human mature B cells expressing sIgM comprising *IGHV4-34* (ref. [49]) or *JH6* or have long CDR3$_H$[55] to attenuate chronic signaling and preserve B cells in an anergic state where they do not secrete self-binding antibodies. Exaggerated sIgM signaling due to defective SHP-1 function or binding self-antigens that increase sIgM cross-linking triggers premature sIgM downregulation and developmental arrest of autoreactive B cells at the T1 stage[45]. Downregulation of sIgM is defective and IgM signaling exaggerated in mouse B cells lacking both CBL and CBL-B[20]. Consistent with failure of these tolerance checkpoints, CBL-deficient B cells exhibited increased sIgM, depletion of JH6 and long CDR3$_H$, increased usage of the *IGHV4-34* gene element and secretion of autoantibodies containing IGHV4-34, manifesting as elevated levels of IgG against a range of self-antigens in individuals deficient in CBL.

CBL-deficient B cells were resistant to apoptosis, likely due to impaired BCR signaling and elevated RAS activity that may promote survival downstream of CD38 engagement. This is consistent with augmented survival of CBL-mutated leukemic cells[56]. Notably, CBL-deficient B cells retain aberrantly high expression of CD38, which would enable sustained RAS signaling and survival. Thus, the accumulation of immature B cells leads to impaired tolerance and the onset of autoimmunity. Additionally, altered BCR signaling may impair censoring of autoreactive clones, allowing the differentiation into autoantibody-secreting cells. Thus, by assessing autoantibodies at the molecular, cellular and protein levels, our data explain (1) increased binding of idiotypic monoclonal 9G4 antibody to CBL-deficient B cells, (2) increased reactivity of serum from CBL-deficient individuals to B cells from HDs and (3) autoantibodies against a wide range of self-antigens in sera from CBL-deficient individuals. These differences in molecular architecture of immunoglobulin expressed by CBL-deficient B cells likely contribute to autoreactivity exhibited by these individuals.

Circulating CBL-deficient B cells exhibited defects in differentiation. The findings of impaired Ig secretion by purified B cells, and that the proportion of memory B cells expressing the WT *CBL* allele is significantly increased compared with transitional and naive B cells, established that these impairments in differentiation were B cell intrinsic. These deficits in naive and memory B cell differentiation and function, and impaired affinity maturation of memory B cells, would contribute to increased recurrent and severe bacterial infections in early life in individuals with *CBL*-LOH.

Individuals with APDS due to *PIK3CD*$^{GOF}$ variants also exhibit an accumulation of immature B cells in the BM, an increased proportion of early transitional B cells and a reduction in memory B cells, impaired Ig secretion in vitro and increased frequencies of 9G4$^+$

(IGHV4-34) B cells and serum autoantibodies[29,50]. In mice, CBL negatively regulates the magnitude of T cell PI3K signaling[57]. Interestingly, CBL, SYK and BTK are phosphorylated following BCR cross-linking, and CBL physically associates with PI3K p85α[9]. Notably, following BCR engagement, CBL-deficient B cells exhibit heightened and sustained phosphorylation of SYK and Igα[14], increased binding of PI3K p85α to pCD19 (ref. [10]), augmented expression of survival proteins BCL-2 and BCL-XL[10] and greater BCR signaling[10,14]. We propose that *CBL*$^{LOF}$ would manifest as heightened PI3K signaling, akin to APDS. Thus, constitutive PI3K signaling due to *PIK3CD*$^{GOF}$ variants or CBL deficiency likely underpins the shared cellular phenotypes and functional defects in these genetically distinct inborn errors of immunity (IEIs). Clinical trials of leniolisib, a specific PI3K p110δ inhibitor, as a treatment for APDS have demonstrated efficacy in alleviating lymphoproliferation, attenuating the frequency of infection, reducing the need for Ig replacement therapy and restoring proportions of B cells subsets[58]. Serum levels of proinflammatory mediators (IFNγ, TNF, CXCL10 and CXCL13) were also reduced in leniolisib-treated individuals with APDS[58]. Given the comparable defects in B cell development and function in individuals with APDS[29] and those with CBL deficiency, together with increased production of inflammatory cytokines (IL-6, IL-1β, TNF and IL-10) and chemokines (CCL2) by CBL-deficient PBMCs[23], leniolisib may also be a candidate pharmacological treatment for CBL deficiency.

The stark contrast between mice and humans regarding lymphocyte-specific functions of CBL reminds us that findings in mice cannot a priori be generalized to humans[43]. There are multiple examples for which genetic deficiencies affect different lineages in mice and humans. For instance, human B cell development is abolished by mutations in *BTK*, causing B cell deficiency and agammaglobulinemia, whereas BTK deficiency in mice only modestly reduces B cell numbers in BM and the periphery[43]. By contrast, human (but not mouse) B cell development is largely independent of IL-7R/γc or BAFF-R signaling, as evidenced by intact B cell development in individuals with mutations in *IL2RG*, *IL7RA* or *TNFRSF13C*[43]. Thus, B cell development in mice is more dependent on signaling via cytokines, whereas human B cell development requires BCR signaling. Thus, CBL deficiency marks another example of the divergence in functional redundancies in lineage development in humans versus mice.

## Limitations of this study

REH cells are a leukemic cell line that lacks BCR expression; they therefore potentially have limited applicability to model transitional B cells, which typically express surface IgM.

## Online content

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

Taja Vatovec[1,2,3,4,62], Anna-Lena Neehus [5,6,7,8,62], Katherine J. L. Jackson [9], Danielle T. Avery [9], Ivan Bagarić[1,2,3,4], Lucia Erazo[2,3,10], Carlos A. Arango-Franco [2,3,10,11], Masato Ogishi[12], Syed F. Ahmed [13], Axel Cederholm [14], Amanda J. Russell [9], Erika Della Mina [9], Dena Al-Rifai[9], Rowena Bull [9], Lori Buetow[13], Steicy Sobrino[3,15,16], Allison Zhang[5,6,7,8], Lara Wahlster[5,6,7,8], Marine Michelet[17], Nima Parvaneh[18,19], Jessica Peel[12], Federica Barzaghi[20], Davide Leardini [21], Quentin Philippot[2,3], Francesco Saettini [22,23], Jacques Dutrieux[24], Benedicte de Muylder[24], Francesca Vendemini[23], Francesco Baccelli[21], Albert Catala[25], Eleonora Gambineri [26,27], Marinella Veltroni[27], Vignesh Pandiarajan[28], Yurena Aguilar [29], Filomeen Haerynck[30,31], Michael Elliott[32,33], Stuart Turville [9], Fabienne Brillot [9], Taushif Khan[34,35,36], Filippo Consonni [27,37], Laureline Berteloot [38], William A. Sewell [9,39], Geetha Rao[9], Laetitia Largeaud[40], Francesca Conti[41,42], Cecile Roullion[3,43], Cécile Masson[3,44], Francesco Pegoraro[27,37], Tianyi Ye[5,6,7,8], Samantha Joubran[5,6,7,8], Emily Villalpando[5,6,7,8], Boris Bessot[3,45], Yoann Seeleuthner[2,3], Tom Le Voyer [2,3,46], Jérémie Rosain[2,3,47], Hailun Li[2,3], Zarah Janda [2,4], Edoardo Muratore[21], Camille Soudée[2,3], Eric Delabesse [48], Claire Goulvestre[49], Mohammad Shahrooei[50], Anne Puel [2,3,13], Isabelle André [3], Christine Bole-Feysot[3,43], Laurent Abel[2,3,12], Miriam Erlacher[51,52], Vivien Béziat [2,3,11], Chantal Lagresle-Peyrou [3,44], Remi Cheynier [24], Emmanuelle Six [3,16], Nico Marr [34,35], Marlène Pasquet[53], Laia Alsina [54], Christopher C. Goodnow [9,55], Nils Landegren [14,56], Alessandro Aiuti [20,57], Peng Zhang [3,12], Riccardo Masetti[21], Danny T. Huang[13,58], Cindy S. Ma [9,39], Jean-Laurent Casanova[2,3,6,12,59], Vijay G. Sankaran [5,6,7,8], Jacinta Bustamante[2,3,12,47], Stuart G. Tangye [9,39,63] ✉ & Jonathan Bohlen [1,2,3,60,61,63] ✉

[1]Gene Center and Department of Biochemistry, Ludwig-Maximilians-Universität, Munich, Germany. [2]Laboratory of Human Genetics of Infectious Diseases, Necker Hospital for Sick Children, Necker Branch, Inserm U1163, Paris, France. [3]Paris Cité University, Imagine Institute, Paris, France. [4]Heidelberg University, Heidelberg, Germany. [5]Division of Hematology/Oncology, Boston Children's Hospital, Harvard Medical School, Boston, MA, USA. [6]Howard Hughes Medical Institute, Boston, MA, USA. [7]Broad Institute of MIT and Harvard, Cambridge, MA, USA. [8]Department of Pediatric Oncology, Dana-Farber Cancer Institute, Harvard Medical School, Boston, MA, USA. [9]Garvan Institute of Medical Research, Sydney, New South Wales, Australia. [10]Primary Immunodeficiencies Group, Department of Microbiology and Parasitology, School of Medicine, University of Antioquia, Medellín, Colombia. [11]Laboratory of Neurogenetics and Neuroinflammation, Imagine Institute, INSERM UMR1163, Université Paris Cité, Paris, France. [12]St. Giles Laboratory of Human Genetics of Infectious Diseases, The Rockefeller University, New York, NY, USA. [13]Cancer Research UK Scotland Institute, Garscube Estate,

Glasgow, UK. [14]Science for Life Laboratory, Department of Medical Biochemistry and Microbiology, Uppsala University, Uppsala, Sweden. [15]Laboratory of Chromatin and Gene Regulation During Development, Paris Cité University, INSERM U1163, Imagine Institute, Paris, France. [16]Laboratory of Human Lymphohematopoiesis, INSERM U1163, Imagine Institute, Paris, France. [17]Unit of Allergy and Pneumology, Childrens Hospital, Toulouse, France. [18]Division of Allergy and Clinical Immunology, Department of Pediatrics, Tehran University of Medical Sciences, Tehran, Iran. [19]Children's Medical Center, Tehran, Iran. [20]San Raffaele Telethon Institute for Gene Therapy (SR-Tiget) and Pediatric Immunohematology and Bone Marrow Transplantation Unit, IRCCS San Raffaele Scientific Institute, Milan, Italy. [21]Pediatric Hematology and Oncology, IRCCS Azienda Ospedaliero-Universitaria di Bologna, Bologna, Italy. [22]Centro Tettamanti, Fondazione IRCCS San Gerardo dei Tintori, Monza, Italy. [23]Pediatrics, Fondazione IRCCS San Gerardo dei Tintori, Monza, Italy. [24]Université Paris Cité, CNRS, Inserm, Institut Cochin, Paris, France. [25]Pediatric Hematology and Oncology Department, Hospital Sant Joan de Déu, University of Barcelona, Barcelona, Spain. [26]Department of Neurosciences, Psychology, Drug Research and Child Health (NEUROFARBA), University of Florence, Florence, Italy. [27]Division of Pediatric Oncology/Hematology, Meyer Children's Hospital IRCCS, Florence, Italy. [28]Allergy Immunology Unit, Department of Pediatrics, Advanced Pediatrics Centre, Postgraduate Institute of Medical Education and Research, Chandigarh, India. [29]Pediatric Oncology and Hematology Department, Miguel Servet Hospital, Zaragoza, Spain. [30]Department of Pediatric Pulmonology, Infectious Diseases and Immunology, Ghent University Hospital, Ghent, Belgium. [31]Primary Immunodeficiency Research Lab, Centre for Primary Immunodeficiency Ghent, Ghent University Hospital, Ghent, Belgium. [32]Sydney Medical School, University of Sydney, Sydney, New South Wales, Australia. [33]Chris O'Brien Lifehouse Cancer Centre, Royal Prince Alfred Hospital, Sydney, New South Wales, Australia. [34]College of Health and Life Sciences, Hamad Bin Khalifa University, Doha, Qatar. [35]Department of Immunology, Sidra Medicine, Doha, Qatar. [36]The Jackson Laboratory, Farmington, CT, USA. [37]Department of Experimental and Clinical Biomedical Sciences 'Mario Serio', University of Florence, Florence, Italy. [38]Department of Pediatric Imaging, Necker Hospital for Sick Children, Paris, France. [39]School of Clinical Medicine, Faculty of Medicine and Health, UNSW Sydney, Sydney, New South Wales, Australia. [40]Laboratory of Hematology, Hospital Center of the University of Toulouse, Toulouse, France. [41]Pediatric Unit, IRCCS Azienda Ospedaliero Universitaria di Bologna, Bologna, Italy. [42]Department of Medical and Surgical Sciences, Alma Mater Studiorum, University of Bologna, Bologna, Italy. [43]Genomics Core Facility, Institut Imagine-Structure Fédérative de Recherche Necker, INSERM U1163 and INSERM US24/CNRS UAR3633, Paris Cite University, Paris, France. [44]Bioinformatic Plateform, Institut Imagine-Structure Fédérative de Recherche Necker, INSERM U1163 et INSERM US24/CNRS UAR3633, Paris Cite University, Paris, France. [45]Biotherapy Clinical Investigation Center, Groupe Hospitalier Universitaire Ouest, APHP, INSERM, Paris, France. [46]Clinical Immunology Department, Assistance Publique Hôpitaux de Paris (AP-HP), Saint-Louis Hospital, Paris, France. [47]Study Center for Primary Immunodeficiencies, Necker Hospital for Sick Children AP-HP, Paris, France. [48]Department of Hematology, CHU and Centre de Recherche de Cancérologie de Toulouse, University Paul-Sabatier Toulouse, Toulouse, France. [49]Laboratory of Immunology, Cochin Hospital, Paris, France. [50]Dr. Shahrooei Lab, Tehran, Iran. [51]Division of Pediatric Hematology and Oncology, Department of Pediatrics and Adolescent Medicine, Medical Center, Faculty of Medicine, University of Freiburg, Freiburg, Germany. [52]Department of Pediatrics and Adolescent Medicine, University Medical Center Ulm, Ulm, Germany. [53]Department of Pediatric Hematology and Oncology, Centre Hospitalo-Universitaire de Toulouse, Toulouse, France. [54]Clinical Immunology and Primary Immunodeficiences Unit, Pediatric Allergy and Clinical Immunology Department, Hospital Sant Joan de Déu Barcelona, Institut de Recerca Sant Joan de Déu, Universitat de Barcelona, Barcelona, Spain. [55]Cellular Genomics Futures Institute & School of Biomedical Sciences, UNSW Sydney, Sydney, New South Wales, Australia. [56]Centre for Molecular Medicine, Department of Medicine (Solna), Karolinska Institute, Stockholm, Sweden. [57]Università Vita-Salute San Raffaele, Milan, Italy. [58]School of Cancer Sciences, University of Glasgow, Glasgow, UK. [59]Department of Pediatrics, Necker Hospital for Sick Children, Paris, France. [60]Department of Pediatrics, Dr. von Hauner Childrens Hospital, LMU Klinikum, Munich, Germany. [61]German Center for Child and Adolescent Health (DZKJ), Munich Site, Munich, Germany. [62]These authors contributed equally: Taja Vatovec, Anna-Lena Neehus. [63]These authors jointly supervised this work: Stuart G. Tangye and Jonathan Bohlen. ✉e-mail: s.tangye@garvan.org.au; bohlen@genzentrum.lmu.de

## Methods

### Inclusion and ethics

This study was conducted in compliance with all relevant ethical regulations for research involving human participants and animals. Ethical approval was obtained from the relevant regulatory bodies, and written informed consent was received from all participants. The research team is committed to fostering inclusivity and diversity in research practices, ensuring equitable access and representation across all aspects of the study.

### Study design and approval

Informed consent was obtained in accordance with local regulations and institutional review board (IRB) approvals in Iran, Italy, France, Spain and Germany. Treating physicians recorded demographic, clinical and microbiological data; gender and socioeconomic information were not collected. Experimental work was conducted in Australia, France, Germany, Sweden and the United States under IRB approvals from Rockefeller University (JCA-0699), INSERM (C10-07, C10-16) and the Sydney Local Health District (X16-0210/LNR/16/RPAH/257). HDs were recruited from France, Spain, Italy, the United States and Australia. Use of discarded cord blood samples was approved by the Boston Children's Hospital IRB. This cohort includes both female and male participants, with no distinct phenotype segregation based on sex. Healthy control individuals of both sexes were also recruited, and no significant differences related to sex were observed.

### WES and Sanger sequencing

gDNA was extracted from whole blood using a iPrep PureLink gDNA Blood kit (Thermo Fisher). Exome capture was performed with a SureSelect Human All Exon 50 Mb kit (Agilent) from 3 µg of gDNA, followed by single-end sequencing on an Illumina Genome Analyzer IIx. Variant calls and familial segregation were confirmed by PCR amplification, agarose gel analysis and Sanger sequencing using BigDye Terminator v3.1 on an ABI Prism 3700.

### Quantitative genotyping by amplicon sequencing

gDNA was extracted from whole blood using a Qiagen Blood and Tissue kit. DNA from leukocyte subsets was obtained by fluorescence-activated cell sorting (FACS) and bulk extraction, except granulocyte DNA, isolated from Lymphoprep pellets. *CBL* variants (H398/C381/C396, R420, Y371) were amplified by PCR using variant-specific primers. DNA templates were quantified by qPCR, and amplicons were generated with minimal cycles (DreamTaq, Thermo). Up to 100 ng of purified amplicon was processed with a TruSeq DNA PCR-Free kit (Illumina), reamplified (eight cycles, KAPA HiFi), bead purified (AMPure XP) and quantified by Qubit and Fragment Analyzer. Equimolar libraries were pooled and sequenced on an Illumina NovaSeq 6000 (paired-end, 100 bp).

The following primers were used for each variant: *CBL*[H398,C381,C396] 5'-TGAGATGGGCTCCACATTCC-3' (forward) and 5'-CAGGCCACCCCT TGTATCAG-3' (reverse); *CBL*[R420] 5'-TCTTTTGCTTCTTCTGCAGGAATC-3' (forward) and 5'-TCTGCTCCTTGCCTCAACAG-3' (reverse); *CBL*[Y371] 5'-GGAAACAAGTCTTCACTTTTTCTGT-3' (forward) and 5'-GTGTCC ACAGGGCTCAATCT-3' (reverse).

### TREC levels

sjTRECs were quantified by nested qPCR, with the primers and standard curve plasmid described by Dion et al.[59]. The qPCR protocol was adapted as previously described[60] using ~500 ng of purified gDNA for each quantification.

### Mass cytometry-based immunophenotyping

Whole-blood mass cytometry was conducted on 200 µl of fresh heparinized blood from participants and healthy controls using a customized antibody panel, as detailed in Bohlen et al.[23] and in accordance with Fluidigm's recommendations. The labeled cells were stained for dead cells overnight, frozen and stored at −80 °C until analysis. Acquisition was performed on a Helios machine (Fluidigm), and the data were analyzed using OMIQ software. Antibodies used in the panel are listed in Extended Data Table 1.

### T cell proliferation assay and cell death

PBMCs from participants and HDs were labeled with CFSE (CellTrace, Thermo Fisher) at 1:10,000 for 20 min at 37 °C, quenched with RPMI, washed and resuspended at $5 \times 10^6$ cells per ml. Cells ($0.5 \times 10^6$ per well) were cultured with graded concentrations of ImmunoCult Human CD3/CD28/CD2 activator (25–3.1 µl ml$^{-1}$) for 5 days. After incubation, viability was assessed by LIVE/DEAD Aqua staining (Thermo Fisher) before surface labeling with anti-CD3, anti-CD4 and anti-CD8. Data were acquired on a NovoCyte Quanteon and analyzed in FlowJo.

### cT$_{FH}$ cell cultures

cT$_{FH}$ cells were isolated by sorting CD4$^+$CD45RA$^-$CXCR5$^+$ T cells from the peripheral blood of HDs or individuals with *CBL*-LOH (FACSAria III, Becton Dickinson). Purified cT$_{FH}$ cells were then cultured with T cell activation and expansion beads (anti-CD2/CD3/CD28; Miltenyi Biotech) in 96-well, round-bottom plates. After 5 days, supernatants were collected, and production of IL-4, IL-5, IL-10, IL-13, IL-17A, IL-17F, IFNγ and TNF was determined by using cytometric bead arrays (Becton Dickinson); secretion of IL-22 (eBioscience) was determined by ELISA. For cytokine expression, activated cT$_{FH}$ cells were re-stimulated with phorbol 12-myristate 13-acetate (100 ng ml$^{-1}$)/ionomycin (750 ng ml$^{-1}$) for 6 h, with Brefeldin A (10 mg ml$^{-1}$) added after 2 h. Cells were then fixed, and expression of intracellular cytokines was detected and quantified by flow cytometric analysis[61].

### Human lymphocyte phenotyping

Buffy coats from HDs were purchased from the Australian Red Cross Blood Service. PBMCs from HDs and individuals deficient in CBL were incubated with monoclonal antibodies to CD20, CD27 and CD10 with monoclonal antibodies specific for CD5, CD11c, CD19, CD21, CD38, CD23, CD44, CD95, CXCR3, CXCR5, IgM, IgD, IgG and IgA. The proportions of transitional (CD20$^+$CD27$^-$CD10$^+$), naive (CD20$^+$CD27$^-$CD10$^-$), memory (CD20$^+$CD27$^+$CD10$^-$) and CD21$^{lo}$CD19$^{hi}$ B cells, as well as levels of expression or proportions of B cells within each of these subsets expressing these molecules, were determined by flow cytometry (LSRII SORP, Becton Dickinson) and analyzed using FlowJo software (Tree Star)[27–29]. BM was obtained from individuals undergoing lymphoma staging and was subsequently found to be uninvolved. BM aspirates were incubated with monoclonal antibodies to CD34, CD19, CD20, CD10, IgM, IgD and CD27. Populations of B-lineage cells (CD19$^+$) as well as pro-B (CD19$^+$CD34$^+$CD10$^+$CD20$^-$IgM$^-$), pre-BI (CD19$^+$CD34$^-$CD10$^+$CD20$^-$IgM$^-$), pre-BII (CD19$^+$CD34$^-$CD10$^+$CD20$^{dim}$IgM$^-$), immature (CD19$^+$CD34$^-$CD10$^+$CD20$^+$IgM$^+$) and recirculating mature (CD19$^+$CD34$^-$CD10$^-$CD20$^+$) B cells and plasma cells (CD19$^+$CD20$^{lo}$CD38$^{hi}$CD27$^{hi}$) were then quantified[29,62,63].

### In vitro HSPC gene editing and B cell differentiation

Human CD34$^+$ HSPCs were isolated from cord blood (EasySep, StemCell 17856) and cultured in StemSpan II medium with CC100 cytokines, 50 ng ml$^{-1}$ thrombopoietin and supplements. After 48 h, cells were electroporated (Lonza 4D, program DZ-100) with 100 pmol of Cas9 (IDT) complexed to 100 pmol of sgRNA targeting *AAVS1* (5'-GGGGCCACTAGGGACAGGAT-3'; 5'-ccggccctgggaatata agg-3') or *CBL* (5'-GGGTCCTATTTTAAGCTCCA-3'; 5'-ATAGCCTTTA CTGATACAAG-3'; 5'-GCCACCCCTTGTATCAGTAA-3'; 5'-AACCAGAAAG CATCTAGTCT-3').

For *PIK3CD* base editing, 2 µg of ABE8e mRNA and 100 pmol of sgRNA (5'-GGGCAGTCCTGCAGAAGGAC-3') were used. Editing efficiency was assessed after 72 h by PCR (Platinum II HotStart,

ThermoFisher) with the following primers: 5′-CCGTTTTTCTGGACAACC CC-3′ (*AAVS1* forward), 5′-CCAGGATCAGTGAAACGCAC-3′ (*AAVS1* reverse), 5′-AAGCACTGGCAAATTGGCTT-3′ (*CBL* forward), 5′-CTCT GCTCCTTGCCTCAACA-3′ (*CBL* reverse), 5′-GAGTAGGGGTGAGGT GGGAA-3′ (*PI3K* forward) and 5′-CAGGCAGATGAGCAGGGCAG-3′ (*PI3K* reverse).

PCR products were analyzed by Nanopore sequencing and CRISPResso2. B cell differentiation was induced by coculture on MS-5 stroma in IMDM with 5% fetal bovine serum (FBS), 20 ng ml⁻¹ IL-7 and supplements for 21–28 days. Differentiation was assessed by flow cytometry using anti-CD34−Alexa488, anti-CD10−PE, anti-CD19− BV421, anti-CD20−PE-Cy7 and anti-CD45−APC.

## Generation of base editor mRNA from in vitro transcription

For base editing, ABE8e (Addgene, 138489) was subcloned into the PEmax-mRNA (Addgene, 204472) backbone. Base editor mRNA was generated from purified PCR product of the template. In brief, 1 μg of PCR product was transcribed using a HiScribe T7 High-Yield RNA Synthesis kit (New England Biolabs, E2040S) according to the manufacturer's protocol. Uridine was substituted with $N^1$-methylpseudouridine-5′-triphosphate (TriLink Biotechnologies, N-1081), and co-translational capping was performed with CleanCap Reagent M6 (TriLink Biotechnologies, N-7453). Residual input DNA was digested using DNase I (New England Biolabs, M0303S), and mRNA was purified using a Monarch Spin RNA Cleanup kit (New England Biolabs, T2050S).

## RNA sequencing of HSPC-derived B cells

For bulk RNA sequencing of *AAVS1*- (n = 2), *CBL*- (n = 2) and *PI3K*^GOF^-edited (n = 1) CD34⁺-derived B cell progenitors, CD19⁺CD10⁺CD20^dim^ cells were sorted on day 21 of coculture and rested for 12 h in RPMI + 1% FBS. RNA was isolated (Norgen Total RNA Micro kit) with on-column DNase treatment. Ultra-low-input RNA sequencing yielded ~30 million reads per sample. FASTQ files were quality control checked with FastQC and aligned to GRCh38 (Ensembl v104) using STAR; quantification used RSEM. Counts were merged, quantile normalized and converted to transcripts per million, and technical replicates were summed. Differential expression was performed in edgeR. Gene set enrichment used fGSEA with MSigDB Hallmark 2024 sets (false discovery rate < 0.05). Genes were preranked by signed $\log_2$(fold change) × $-\log_{10}$(adjusted *P* value). The *PI3K*^GOF^ signature was defined as | $\log_2$(fold change) | ≥ 0.5 and false discovery rate < 0.05 and was used for enrichment analysis.

## CD38 surface staining

For each condition, 5 × 10⁵ HEK293T (ATCC, CRL-3216), BJAB (DSMZ, ACC 757) or REH (ATCC, CRL-8286) cells were washed in FACS buffer (PBS, 2% FBS and 2 mM EDTA) and stained with a fixable viability dye (1:500) for 10 min at 4 °C. Half remained unstained; the others were labeled with APC/Cy7 anti-CD38 (clone HIT2, BioLegend 303533, 1:100, 1 h, 4 °C). After washing, cells were analyzed on a BD LSRFortessa, and median fluorescence intensity was quantified in FlowJo.

## Generation of *CBL*^Y371C^ KI cell lines

Cas9 (7.5 pmol) and sgRNA (5′-GTCACCATGAGTAGTAGTTT-3′) were complexed for 10 min at 25 °C, followed by the addition of 15 pmol of single-stranded DNA donor (5′-ATTCAATTACTGGAAAATAAAAG GAGTTCATGTAGTTTTTGTCCAccCTTGAGTCACAATGGGTAGT AGTCTAGGAAAAGAAAAAGACTCTAAAGAAAAAGATCCCAAAGTACC ATCAGCCAAGGAAAGAGAAAAGG-3′). In total, 2 × 10⁵ cells were electroporated with the ribonucleoprotein and single-stranded DNA mix using the Neon NxT system (1,750 V, 20 ms, one pulse) and cultured in RPMI + 20% FBS, penicillin/streptomycin and 1 μM HDR Enhancer (IDT). After 1 week, edited pools were genotyped by Sanger sequencing; single-cell clones were derived by limiting dilution and confirmed by sequencing.

## Anti-CD38 B cell apoptosis assay

CD34⁺-derived B cells were purified with CD19 MicroBeads (Miltenyi) and plated at 2 × 10⁵ per well. Cells were treated with 0.6 μg ml⁻¹ daratumumab for 30 min, cross-linked with 25 μg ml⁻¹ goat anti-human IgG + IgM for 16 h, stained with Annexin V and 7-AAD (BioLegend) and analyzed on a Fortessa X cytometer. REH cells (5 × 10⁵) were treated identically but incubated for 24 h and analyzed on a NovoCyte or Fortessa X-20.

## BJAB cell apoptosis assay

BJAB cells were seeded at 2 × 10⁵ cells per well in V-bottom, 96-well plates in RPMI 1640 medium supplemented with 20% fetal calf serum (FCS). Cells were stimulated with AffiniPure goat anti-human IgG/IgM (25 μg ml⁻¹; Jackson ImmunoResearch) and incubated for 16 h at 37 °C. After incubation, cells were washed and stained with Annexin V and 7-AAD (BioLegend) according to the manufacturer's instructions. Apoptosis was assessed by flow cytometry using a BD Fortessa cytometer, and data were analyzed with FlowJo.

## Intracellular staining for pERK

CD19⁺ or CD19⁺CD20^hi^ HSPC-derived B cell progenitors were sorted after 21–28 days of coculture and starved overnight in RPMI + 2%FBS. Cells were then stimulated with daratumumab (60 μg μl⁻¹) and cross-linked with 25 μg ml⁻¹ AffiniPure Goat Anti-Human IgG + IgM (H + L) for 15 min at 37 °C. Cells were subsequently fixed with Phosflow Fix Buffer I (BD Biosciences, 557870) for 10 min at 37 °C, permeabilized with Phosflow Perm Buffer III (BD Biosciences, 558050) for 20 min at 4 °C and stained with pERK−AF647 (1:25; BD Biosciences, 612593) for a minimum of 4 h. Samples were analyzed on an LSR Fortessa X-20.

## Stimulation of BJAB cells for western blotting

In total, 1 × 10⁶–1.5 × 10⁶ REH or BJAB cells were seeded in 96-well, V-bottom plates per condition and starved in 0% FCS RPMI 1640 medium for 2 h. BJAB cells were stimulated with AffiniPure Goat Anti-Human IgG + IgM (H + L) at a final concentration of 25 μg ml⁻¹ at different time points (5, 15, 30, 120 and 240 min). Next, cells were washed two times with 1× PBS and lysed with lysis buffer. Protein lysates were analyzed in line with the immunoblotting protocol.

## REH cell stimulation and lysis for immunoblotting

REH cells were seeded at a density of 1.5 × 10⁶ cells per well in RPMI 1640 medium supplemented with 5% FCS and maintained at 37 °C with 5% CO₂. Cells were stimulated with 12 μg ml⁻¹ daratumumab (administered every 8 h) for a total of 24 h. Following stimulation, cells were collected by centrifugation at 450*g* for 5 min, washed with cold PBS and lysed in RIPA buffer containing protease and phosphatase inhibitors. Lysates were clarified by centrifugation at 10,000*g* for 15 min at 4 °C, and protein concentration was determined using a BCA assay. Samples were then processed for SDS−PAGE and immunoblotting as described below.

## Immunoblotting

Immunoblotting was performed as previously described by Bohlen et al.[23]. Briefly, cells were washed in FCS-free DMEM or PBS and lysed in RIPA buffer supplemented with protease inhibitors (Roche Mini EDTA-free, one tablet per 10 ml), phosphatase inhibitors (2 mM sodium orthovanadate; Roche PhosSTOP, one tablet per 10 ml; 0.1 M sodium fluoride; 0.1 M β-glycerophosphate) and Benzonase (50 IU ml⁻¹). Lysates were clarified, protein concentrations were measured by BCA or Bradford assay, and equal amounts were subjected to SDS−PAGE. Proteins were transferred to 0.2-μm nitrocellulose membranes, Ponceau stained, blocked in 5% skim milk in PBS + 0.05% Tween-20 (PBST) (1 h), rinsed and incubated overnight at 4 °C with primary antibodies in 5% bovine serum albumin (BSA) in PBST or 5% skim milk in PBST. Membranes were washed three times for 15 min in PBST, incubated with secondary antibodies (1:5,000 in 5% skim milk in PBST, 1 h, room temperature) and washed again three times for 15 min. Chemiluminescence

was detected using ECL reagents and a Bio-Rad Chemidoc. Reagents used included human EGF (Sigma-Aldrich), ATP (Fisher Bioreagents) and chloroquine (Cell Signaling Technology).

## In vitro B cell differentiation

Buffy coats from HDs were purchased from the Australian Red Cross Blood Service. PBMCs were isolated and then labeled with monoclonal antibodies to CD20, CD27 and CD10, and transitional (CD20⁺CD10⁺CD27⁻), naive (CD20⁺CD10⁻CD27⁻) or memory (CD20⁺CD10⁻CD27⁺) B cells were then sort purified using a FACSAria III (Becton Dickinson). Purity of the recovered populations was typically >98%. Transitional, naive and memory B cells were then cultured in 96-well, U-bottom plates (Falcon; $5 \times 10^3$ cells per 200-µl well) for 5–7 days to determine secretion of IgM, IgG and IgA[29,64]. B cells were stimulated with 200 ng ml⁻¹ CD40L cross-linked to 50 ng ml⁻¹ HA Peptide monoclonal antibody (R&D Systems) alone or together with 50 ng ml⁻¹ IL-21 (PeproTech), 2.5 µg ml⁻¹ F(ab′)₂ fragment of goat anti-IgA/IgG/IgM (H + L; Jackson ImmunoResearch) or CpG. Following in vitro stimulation with CD40L/IL-21, the proportion of plasmablasts generated from cultured transitional/naive B cells isolated from HDs or individuals with *CBL*-LOH was determined by flow cytometry (as CD20ˡᵒCD38ʰⁱCD27ʰⁱ cells) as previously described[29]. BM mononuclear cells were labeled with monoclonal antibodies to CD19, CD20, CD38 and CD27. BM plasma cells were identified as CD19⁺CD20ˡᵒCD38ʰⁱCD27ʰⁱ, sorted from the BM of HDs and P2 and P3 and cultured for 5 days in complete medium[63]. After this time, supernatants were collected, and levels of secreted Ig were determined.

## Detection of SARS-CoV-2-specific B cells

Biotinylated full-length SARS-CoV-2 spike protein (Acro Biosystems) was labeled with streptavidin (SA)–BUV395 or SA–PE (BD Biosciences) at a 20:1 ratio for 1 h at 4 °C; SA–FITC served as a decoy probe. Cryopreserved PBMCs from vaccinated individuals deficient in CBL and HDs were stained with 200 ng of spike and 20 ng of decoy probe in Brilliant Buffer for 1 h at 4 °C. Prepandemic PBMCs were used as negative controls to confirm assay specificity.

## BJAB *CBL*-KO cell line generation and stable overexpression

BJAB cells were transduced with pLENTI-V2 encoding Cas9 and sgRNA (5′-AAGCTCATGGACAAGGTGAA-3′) targeting *CBL*, followed by 2 weeks of puromycin selection. KO efficiency was verified by western blotting, and single clones were obtained by limiting dilution. For stable overexpression, lentiviral particles were produced in HEK293T cells transfected with pPAX2, pHBX2, pVSV-G and either pTRIP-EV or pTRIP-CBL using X-tremeGENE 9 (Roche). Supernatants collected after 20 h were filtered and supplemented with 8 µg ml⁻¹ protamine sulfate and used to transduce $1.5 \times 10^5$ BJAB cells by spinoculation (2 h, 1,200$g$). After 3–4 days, transgene expression was assessed by DNGFR staining and enriched by magnetic-activated cell sorting.

## BJAB immunoglobulin production and quantification

Two hundred thousand BJAB cells were seeded in 96-well, V-bottom plates. Cells were stimulated for 24 h in 100 µl of 20% FCS RPMI 1640 medium and were either left unstimulated or were stimulated with IL-4 (R&D Systems, BT-004). The supernatant was collected 24 h later. Cytokine secretion by BJAB cells was quantified with a LEGENDplex Human Immunoglobulin Isotyping Panel (eight-plex). Cytokine determinations were performed according to the manufacturer's protocol, except that the beads, antibody and SA–PE were diluted fourfold with the assay buffer supplied before use.

## Bulk RNA-sequencing of HD and *CBL*-LOH naive B cells

RNA was extracted from $2 \times 10^4$ sorted naive B cells from individuals with *CBL*-LOH and age-matched control individuals using an RNeasy Plus Micro kit (Qiagen). Full-length cDNA was generated from 1 ng of RNA with SMART-Seq v4 (Clontech) and used for library prep with Nextera XT (Illumina). Barcoded libraries were pooled equimolarly and sequenced on an Illumina NovaSeq 6000 (100 bp, paired-end). FASTQ quality was assessed with FastQC, and reads were aligned to GRCh37. p13 (human) or GRCm38.p6 (mouse) using STAR v2.6 and quantified with featureCounts v1.6.0. Gene set enrichment analysis (fgsea) used Hallmark gene sets from MSigDB. Healthy pediatric datasets were previously published (PRJNA1141130).

## Fresh B cell isolation and immunoglobulin production

Peripheral blood was collected under institutional ethical approval. PBMCs were isolated by Lymphoprep density centrifugation (450$g$, 20 min, 25 °C) and washed with PBS. CD19⁺ B cells were purified using CD19 MicroBeads and LS columns (Miltenyi). Cells ($2 \times 10^5$ per well) were cultured in RPMI + 10% FCS for 24 h at 37 °C with 5% CO₂ without stimulation. Supernatants were collected and analyzed for Ig production using the Human Immunoglobulin Isotyping Panel (LEGENDplex, BioLegend).

## Fresh B cell stimulation with monocyte-derived supernatants

Peripheral blood from individuals with *CBL*-LOH and HDs was processed by Lymphoprep density centrifugation. CD14⁺ monocytes were isolated using CD14 MicroBeads (Miltenyi) and cultured in RPMI + 10% FCS at 37 °C with 5% CO₂ for 24 h. Supernatants were collected and stored at −80 °C. CD19⁺ B cells from HDs were purified with CD19 MicroBeads, plated at $2 \times 10^5$ cells per well and stimulated with 100 µl of monocyte supernatant for 24 h. Cell-free supernatants were then analyzed for immunoglobulin isotypes using the Human Immunoglobulin Isotyping Panel (LEGENDplex, BioLegend).

## *IGH* repertoire sequencing

PBMCs were isolated from HDs and individuals deficient in CBL, stained with monoclonal antibodies to CD20, CD27 and CD10 and sorted by flow cytometry into subsets of transitional (CD20⁺CD10⁺CD27⁻), naive (CD20⁺CD10⁻CD27⁻) and memory (CD20⁺CD10⁻CD27⁺) B cells. *IGH* libraries for IgM (all cell types), IgG (memory) and IgA (memory) were prepared from sorted populations as previously reported[31]. Briefly, RNA was prepared from sorted populations (RNAeasy kit, Qiagen), and cDNA was synthesized using oligo(dT) and random hexamer primers. *IGH* transcripts were then amplified in separate PCRs for each immunoglobulin isotype using an isotype-specific reverse primer and a pool of *IGHV* forward primers. Samples were barcoded with a Nextera Index kit and equimolar pooled for 2 × 300 bp paired-end sequencing on an Illumina MiSeq instrument. Samples were demultiplexed during FASTQ generation. Paired-end reads were merged using flash (version 1.2.11)[65]. Sequences were then processed with pRESTO (version 0.7.2)[66]. Sequences were quality filtered with FilterSeq (-q 20), and MaskPrimers was used to identify and trim forward and reverse primers, requiring exact matches (−maxerror 0), and to tag the constant region exon sequence to determine the isotype subclass (−maxerror 0.05). Datasets were then dereplicated with CollapseSeq to keep a single representative of each unique nucleotide sequence before alignment to the human *IGH* reference with IgBLAST (version 1.14.0)[67]. IgBLAST outputs were filtered to remove sequences that had truncated *IGHV* genes (length < 200), lacked *IGHJ* calls, lacked a defined *CDR3* sequence or contained ambiguous nucleotides. Clones were assigned by first subsetting the pooled data for each person by V gene (no allele), J gene (no allele) and *CDR3* length and then clustering the *CDR3* nucleotide sequences using cd-hit[68] using a 90% identity threshold. To explore *IGH* in CRISPR–Cas9-edited cells, *IGH* contigs were built from bulk RNA sequencing using TRUST4 (version 1.1.6)[69]. TRUST4 contigs were postprocessed with IgBLAST. *IGH* data were analyzed in R (version 4.4.1) using RStudio and the following packages: tidyverse (version 2.0.0; https://doi.org/10.21105/joss.01686) and rstatix (version 0.7.2; https://CRAN.R-project.org/package=rstatix).

### IGHV9G4 staining and cell painting

PBMCs from HDs and individuals with CBL deficiency were stained with anti-CD20 FITC (L27), anti-CD27–PE-Cy7 (M-T271) and anti-CD10–PE (HI10a), all from BD Biosciences, as well as the monoclonal anti-9G4 (IGM Bioscience), which recognizes unmutated antibodies to IGHV4-34 (ref. [70]). Binding of anti-9G4 to transitional (CD20⁺CD10⁺CD27⁻), naive (CD20⁺CD10⁻CD27⁻) or memory (CD20⁺CD10⁻CD27⁺) B cells was then determined by gating on these B cell populations[50]. To detect the levels of antibodies to IGHV4-34 in the serum, serum from HDs or individuals with CBL deficiency was incubated with PBMCs from HDs for 30 min on ice. Cells were then stained with antibodies to CD20, CD27 and CD10, as well as the monoclonal anti-9G4, and the mean fluorescence intensity of anti-9G4 staining on transitional, naive and memory B cells was determined[50].

### VirScan phage immunoprecipitation sequencing

Phage immunoprecipitation sequencing was performed on plasma from control individuals and individuals with CBL deficiency using an expanded VirScan library and analyzed as previously described[71–74], with minor modifications. Species-specific seropositivity cutoffs were derived from an in-house dataset using a generalized linear model. Virus-specific scores were calculated as the ratio of enriched, non-homologous peptide counts to cutoff values and visualized as heat maps. Nineteen age-matched individuals of Arab ancestry from an 800-person reference cohort, pooled IVIg (Privigen, CSL Behring) and IgG-depleted serum (Molecular Innovations) served as controls.

### Protein microarray analysis

Protein microarray analyses were performed as described by Le Voyer et al.[75]. Briefly, HuProt protein microarrays (CDI Laboratories) were incubated for 90 min in 5 ml of blocking buffer (2% BSA and PBST) and immersed overnight in 5 ml of the same buffer containing serum from HDs or individuals with CBL deficiency (1:2,000). Arrays were washed five times for 5 min each with 5 ml of PBST. Alexa Fluor 647 goat anti-human IgG (Thermo Fisher, A-21445, RRID:AB_2535862) and Dylight 550 goat anti-GST (Columbia Biosciences, D9-1310) were diluted in blocking buffer (1:2,000 and 1:10,000, respectively), and arrays were incubated in 5 ml of this mixture for 90 min. Arrays were washed again five times, as described above. All incubations and washes were performed on an orbital shaker, and arrays were protected from light after fluorescent antibody addition. Arrays were rinsed three times in deionized water and centrifuged for ~30 s to dry. The same day, arrays were scanned on an Innoscan 1100AL fluorescence scanner (Innopsys) using Mapix v9.1.0. Images were analyzed with GenePix Pro v5.1.0.19 or GenePix Pro 7. Signal intensities were normalized to correct for interexperiment variation. Data from additional HD arrays generated in independent experiments were included. Signal intensities were extracted with GenePix Pro v5.1.0.19 and GenePix Pro 7 with subtraction of local background.

### Multiplex bead assay

The method used to detect human IgG in serum using magnetic beads was as outlined in Voyer et al.[75] except for choice of analytes. Briefly, magnetic bead coupling was performed with an AnteoTech Activation kit for Multiplex Microspheres (A-LMPAKMM-10) following the manufacturer's protocol, including optional blocking, to couple MagPlex beads (Luminex) to the analyte panels. Samples were diluted 1:25 in PBS and then 1:10 in assay buffer (PBST, 3% BSA and 5% milk). Bead stocks were sonicated for 1 min and mixed with kit storage buffer, and samples were centrifuged for 1 min at 3,000 rpm after the first dilution. For binding, 45 µl of sample was incubated with 5 µl of bead stock for 2 h at room temperature in the dark with shaking at 650 rpm. Beads were washed three times with PBST and centrifuged at 2,000 rpm between washes. Beads were resuspended in 50 µl of 0.2% paraformaldehyde per well, vortexed, incubated for 10 min at room temperature and

centrifuged again at 2,000 rpm. After another three PBST washes, beads were incubated with secondary antibody (Invitrogen H10104, 2384336) for 30 min at room temperature. A final three-cycle PBST wash was performed before resuspension in PBST for Luminex FlexMap 3D acquisition. This workflow was applied to both antigen panels. Panel 1 (68 analytes) included anti-human IgG, EBNA1, GM-CSF, IFNA1, IFNA7, IFNB1, IFNG, IFNL4, IFNW1, IL-12, IL-17A, IL-17F, IL-22, IL-23, IL-28a, IL-28b, IL-29, IL-6, IL1RN, SARS-CoV-2 spike, TNF, Trove2 and TXLNB. Panel 2 (28 analytes) included ACAN, angiotensin II, ANKS4B, anti-human IgA, anti-human IgG, ARNT, EBNA1, ENTPD1, FOXP3, GAD2, HNF4A, IFNA1, IFNA10, IFNA14, IFNA16, IFNA17, IFNA2, IFNA21, IFNA4, IFNA5, IFNA6, IFNA7, IFNA8, IFNB1, IFNG, IFNL4, IFNW1, IL-12, IL-17A, IL-17F, IL-1F6, IL1RN, IL-22, IL-23, IL-28a, IL-28b, IL-29, IL-6, PF4, PPARG, PROS, RBD, RORC, RXRA, SARS-CoV-2 nucleocapsid, SARS-CoV-2 S protein RBD, SARS-CoV-2 S protein spike, TNF, TNNC2, TROVE2/Ro60, TXLNB, USH1C and VIL1.

### Statistics

To assess statistical significance, we used a $P$ value cutoff of 0.05 after correction for multiple testing. Typically, two groups were compared: healthy controls versus individuals with $CBL$-LOH.

### Reporting summary

Further information on research design is available in the Nature Portfolio Reporting Summary linked to this article.

## Data availability

All raw data for bulk RNA sequencing of primary naive B cells (GSE307942) and HSPC-derived B cell precursors (GSE307131) have been deposited in the Gene Expression Omnibus repository and will be made available as of the publication date. BCR-sequencing data have been deposited on SRA (PRJNA1328925) and are available as of publication. All raw data and resources will be made available upon request to the corresponding authors. There is no original code generated in this study. Source data are provided with this paper.

## Code availability

No new code was written for this paper.

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

## Acknowledgements

We thank the participants and their families for participating in this study. We thank L. Lorenzo-Diaz, M. Chrabieh, A. Begu, M. Woollet, D. Liu and Y. Nemirovskaya for assistance. We also thank the National Facility for Autoimmunity and Serology Profiling at SciLifeLab for instrument support. The laboratory of J. Bohlen is supported by the Ludwig Maximillian University of Munich, DFG TRR237, the Else-Kröner Fresenius Foundation, the Fritz Thyssen Foundation and an Emmy Noether Program grant from the DFG. The Human Genetics of Infectious Disease (HGID) Laboratory is supported by grants from Institut National de la Santé et de la Recherche Médicale (INSERM), the Imagine Institute, Paris Cité University, St. Giles Foundation, National Center for Advancing Translational Sciences, National Institutes of Health (NIH) Clinical and Translational Science Award program (UL1TR001866), National Institute of Allergy and Infectious Diseases (NIAID), NIH (R01AI095983), French National Research Agency (ANR) under the 'Investments for the future' program (ANR-10-IAHU-01), MAFMACRO (ANR-22-CE92-0008), Integrative Biology of Emerging Infectious Diseases Laboratory of Excellence (ANR-10-LABX-62-IBEID), French Foundation for Medical Research (EQU201903007798), Square Foundation, Grandir 'Fonds de solidarité pour l'enfance' and W. E. Ford, General Atlantic's Chairman and Chief Executive Officer, G. Caillaux, General Atlantic's Co-President, Managing Director and Head of Business in EMEA, and the General Atlantic Foundation. J. Bohlen was supported by fellowships from an EMBO postdoctoral fellowship and Marie Sklodowska-CurieAction (101065761). C.S.M. and S.G.T. are supported by Investigator Grants awarded by the National Health and Medical Research Council of Australia (C.S.M.: 2017463 (level 1); S.G.T.: 1176665 (level 3)). A.-L.N. is supported by the EMBO Postdoctoral Fellowship (ALTF 209-2024). A.Z. was supported by an MSTP grant (5T32GM144273-03). N.L. was supported by Swedish Research Council and Göran Gustafsson Foundation. M.O. was supported by David Rockefeller Graduate Program, Funai Foundation for Information Technology, Honjo International Scholarship Foundation and New York Hideyo Noguchi Memorial Society. A. Catala was supported by a La Marató de TV3 grant (202001-32). S.F.A., L. Buetow and D.T.H. were supported by a Cancer Research UK core grant (A29256). F. Barzaghi and A.A. thank the ERN-RITA association for their help and support. S.J. is supported by an NIAID NIH F31 grant (F31AI186590-01). E.V. is supported by the ASH Graduate Hematology Award. Studies from the laboratory of V.G.S. were supported by the Howard Hughes Medical Institute, Alex's Lemonade Stand Foundation, the Gates Foundation, the Edward P. Evans Foundation and the NIH (R01DK103794, R01CA265726, R01CA292941, R33CA278393, R01HL146500). V.G.S. is an investigator of the Howard Hughes Medical Institute. The project described was supported by award number T32GM144273 from the National Institute of General Medical Sciences. The content is solely the responsibility of the authors and does not necessarily represent the official views of the National Institute of General Medical Sciences or the NIH. The funders had no role in study design, data collection and analysis, decision to publish or preparation of the manuscript.

## Author contributions

T.V., A.-L.N., K.J.L.J., D.T.A., I.B., L.E., C.A.A.-F., M.O., S.F.A., A. Cederholm, A.J.R., E.D.M., D.A.-R., R.B., L. Buetow, S.S., A.Z., L.W., J.P., Q.P., J.D., B.d.M., M. Elliott, S.T., F. Brillot, T.K., W.A.S., G.R., C.R., C.M., F.P., T.Y., S.J., E.V., B.B., Y.S., T.L.V., J.R., H.L., Z.J., C.S. and J. Bohlen conducted experiments, and acquired and analyzed data. M.M., N.P., F. Barzaghi, D.L., F.S., F.V., F. Baccelli, A. Catala, E.G., M.V., V.P., Y.A., F.H., F. Consonni, L. Berteloot, L.L., F. Conti, E.M., E.D., C.G. and M.S. collected clinical materials and information. A.P., I.A., C.B.-F., L. Abel, M. Erlacher, V.B., C.L.-P., R.C., E.S., N.M., M.P., L. Alsina, C.C.G., N.L., A.A., P.Z., R.M., D.T.H., C.S.M., J.-L.C., V.G.S., J. Bustamante, S.G.T. and J. Bohlen designed and supervised the research studies. J-.L.C., J. Bustamante, S.G.T. and J. Bohlen wrote the paper, and all authors were involved in editing the paper.

## Funding

## Competing interests

V.G.S. is an advisor to Ensoma, Cellarity and Beam Therapeutics, unrelated to this work. The other authors declare no competing interests.

## Additional information

**Extended data** is available for this paper at https://doi.org/10.1038/s41590-025-02381-7.

**Correspondence and requests for materials** should be addressed to Stuart G. Tangye or Jonathan Bohlen.

**Extended Data Table 1 | Antibodies used in this study for mass cytometry**

| Tag | Antibody Panel | Clone | Catalog # | Manufacturer |
|---|---|---|---|---|
| 089Y | CD45 | HI30 | 3089003B | Fluidigm |
| 116Cd | CD66b | QA17A51 | 396902 | Biolegend |
| 141Pr | CCR6 | G034E3 | 3141003A | Fluidigm |
| 142Nd | CD19 | HIB19 | 3142001B | Fluidigm |
| 143Nd | CD127 | A019D5 | 3143012B | Fluidigm |
| 144Nd | CD38 | HIT2 | 3144014B | Fluidigm |
| 145Nd | CD31 | WM59 | 3145004B | Fluidigm |
| 146Nd | IgD | IA6-2 | 3146005B | Fluidigm |
| 147Sm | CD11c | Bu15 | 3147008B | Fluidigm |
| 148Nd | CD20 | 2H7 | 302302 | Biolegend |
| 149Sm | CD25 | 2A3 | 3149010B | Fluidigm |
| 150Nd | NKVFS1 | NKVFS1 | MCA2243GA | Bio Rad |
| 150Nd | KIR3DL1L2 | REA970 | 130-126-489 | Miltenyi Biotec |
| 151Eu | CD123 | 6H6 | 3151001B | Fluidigm |
| 152Sm | TCR-γδ | 11F2 | 3152008B | Fluidigm |
| 153Eu | Va7.2 | 3C10 | 3153024B | Fluidigm |
| 154Sm | CD3 | UCHT1 | 3154003B | Fluidigm |
| 155Gd | CD45RA | HI100 | 3155011B | Fluidigm |
| 156Gd | CCR10 | REA326 | 130-122-317 | Miltenyi Biotec |
| 158Gd | CD27 | L128 | 3158010B | Fluidigm |
| 159Tb | CD1c | L161 | 331502 | Biolegend |
| 160Gd | CD14 | M5E2 | 3160001B | Fluidigm |
| 161Dy | CLEC9A | 8F9 | 3161018B | Fluidigm |
| 162Dy | CD21 | REA940 | 130-124-315 | Miltenyi Biotec |
| 163Dy | CXCR3 | G025H7 | 3163004B | Fluidigm |
| 164Dy | CD161 | HP-3G10 | 3164009B | Fluidigm |
| 165Ho | NKG2C | REA205 | 130-122-278 | Miltenyi Biotec |
| 166Er | CD24 | ML5 | 3166007B | Fluidigm |
| 167Er | CCR7 | G043H7 | 3167009A | Fluidigm |
| 168Er | CD8 | SK1 | 3168002B | Fluidigm |
| 169Tm | NKG2A | Z199 | 3169013B | Fluidigm |
| 170Er | iNKT | 6B11 | 3170015B | Fluidigm |
| 171Yb | CXCR5 | RF8B2 | 3171014B | Fluidigm |
| 172Yb | CD57 | HNK-1 | 359602 | Biolegend |
| 173Yb | HLA-DR | L243 | 3173005B | Fluidigm |
| 174Yb | CD4 | RPA-T4 | 300502 | Biolegend |
| 175Lu | CCR4 | L291H4 | 3175035A | Fluidigm |
| 176Yb | CD56 | NCAM16.2 | 3176008B | Fluidigm |
| 209Bi | CD16 | 3G8 | 3209002B | Fluidigm |

a

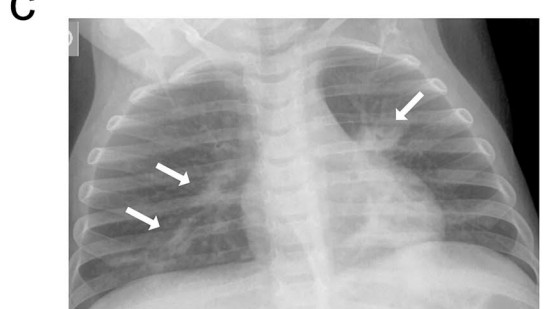

Pneumonia, P1, 5 y.o.

b

Lesions, P1, 9 y.o.

Lesions, P2, 9 y.o.          Lesions, P3, 9 y.o.

c 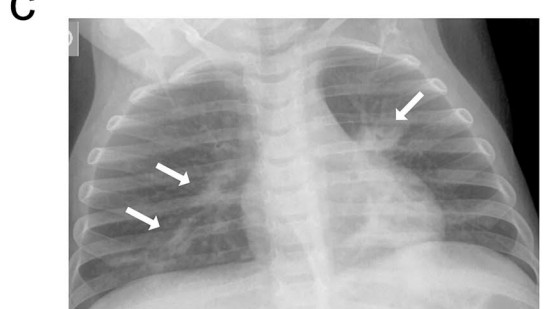 d 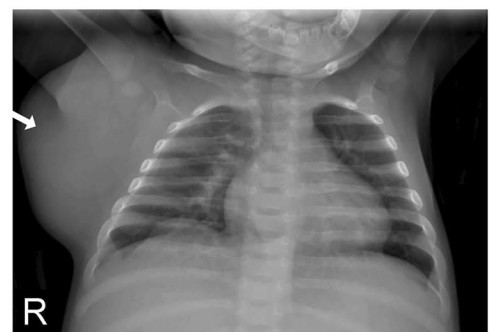

Pneumonia, P7, 15 months old

Lymphadenopathy AFB+, P10, 7 months old,
right axillary lymph node

**Extended Data Fig. 1 | Clinical Imaging of participants' torsos.** (**a**) CT scan during pneumonia of P1 at 5 years of age. (**b**) CT scans of parenchymal lesions of P1, P2 and P3 at 9 years of age. (**c**) Pneumonia in X-ray imagine of P7 at 15 months old. (**d**) X-ray image of P10 at 7 months old with lymphadenopathy affecting the right axillary lymph node.

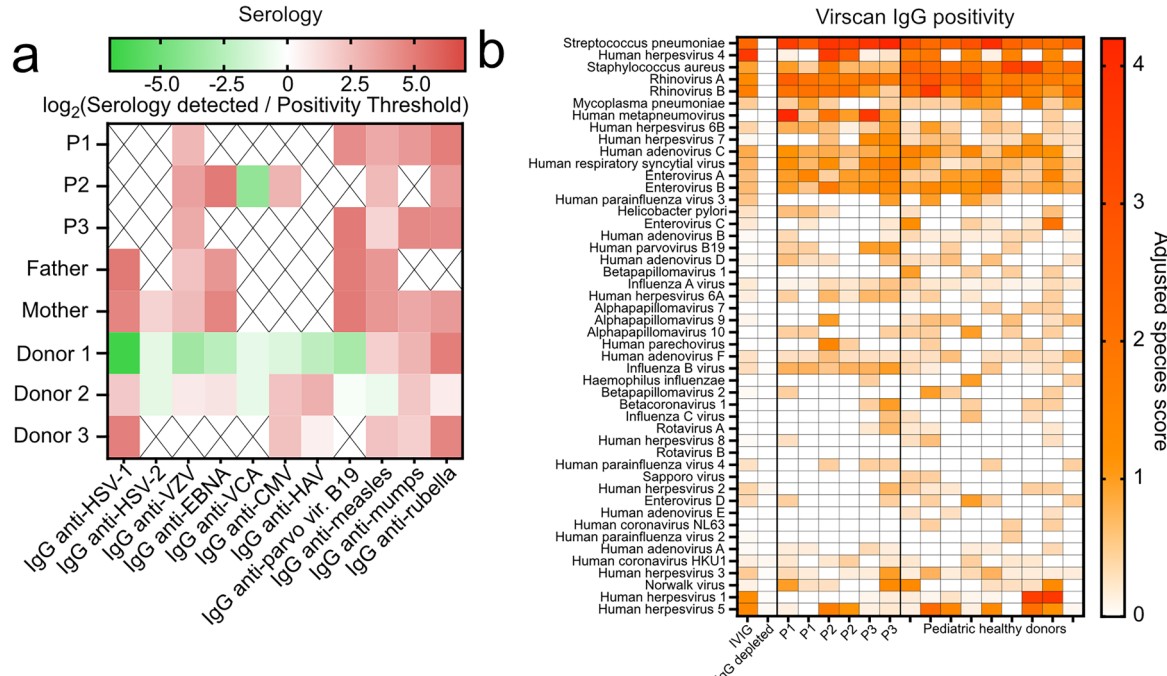

**Extended Data Fig. 2 | Humoral immunity of CBL deficient individuals and healthy donors. (a)** Serology for the indicated microbes of CBL-LOH patients, their relatives and healthy donors. The serology is indicated as the log₂(fold-change) over the positivity limit. **(b)** Virome-wide serological profiling (virscan) of CBL-LOH patients and healthy donors. Signals indicate the adjusted species score calculated from all peptide counts of the indicated species.

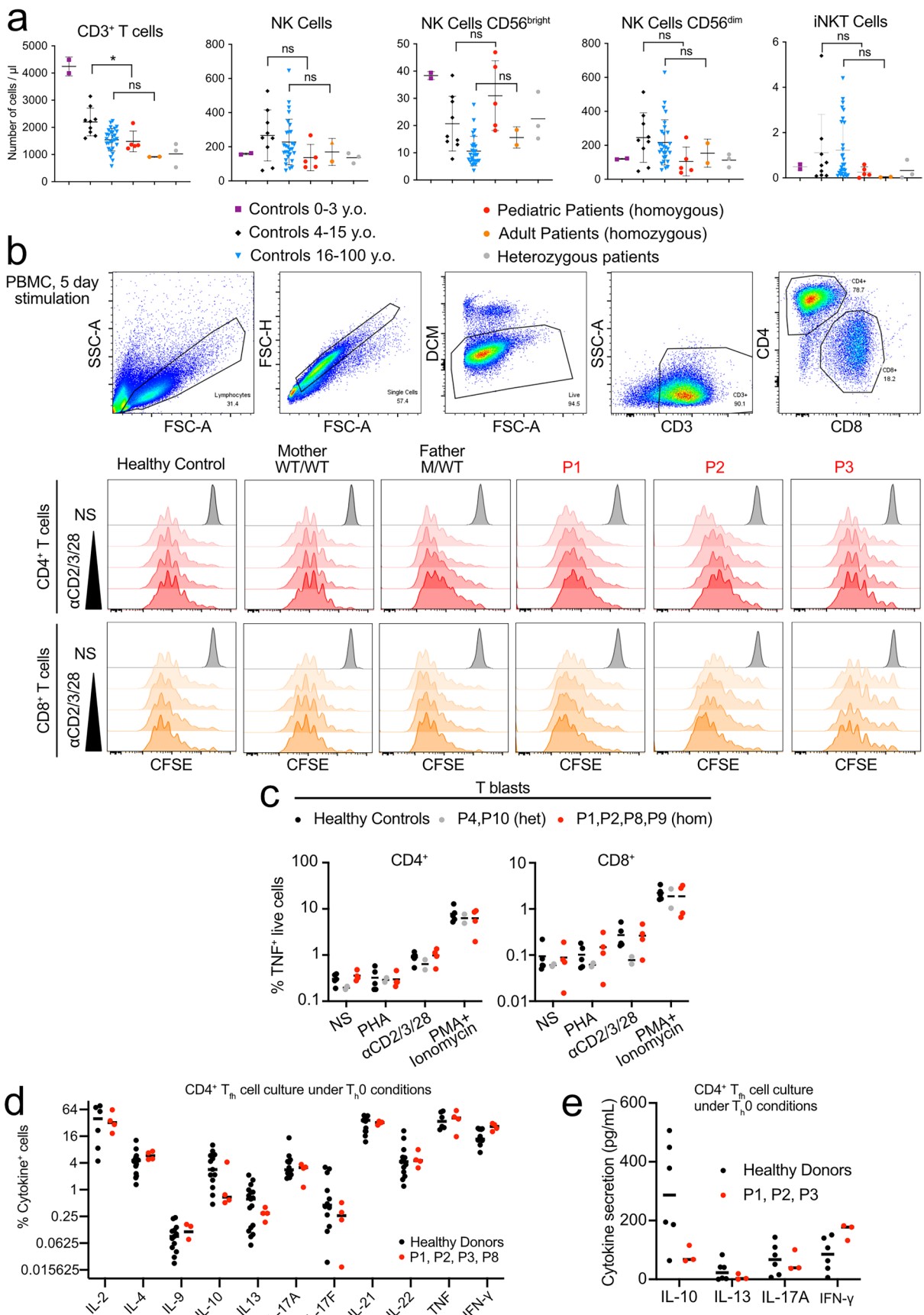

Extended Data Fig. 3 | See next page for caption.

**Extended Data Fig. 3 | Characterization and functional study of patients NK and T cells.** (**a**) CytoF immunophenotyping of patients total CD3[+] T cells, NK cells and indicated NK cell subsets. Controls 0–3 y.o. (n = 2), Controls 4–15 y.o. (n = 9), Controls 16–100 y.o. (n = 28), pediatric patients (LOH) (n = 5), adult patients (LOH) (n = 2), heterozygous individuals (n = 3), mean ± s.d. Statistical significance was assessed with multiple Mann Whitney tests adjusted for multiple testing. *p<0.05. (**b**) Assessment of T cell proliferation of patients P1, P2 and P3. (top) gating strategy, (bottom) CFSE dilution plots. (**c**) Intracellular TNF production of CD4[+] and CD8[+] T cell blasts in homozygous (n = 4), heterozygous (n = 2) and wildtype (n = 5) state of *CBL* Ub[LOF] variants. Line indicates the mean of the displayed datapoints. (**d,e**) Intracellular and extracellular cytokine production by patients T$_{fh}$ cells stimulated with CD2/CD3/CD28 under T$_h$0 conditions for 5 days. (**d**) Intracellular staining of the indicated cytokines. Healthy donors (n = 17), CBL-LOH patients (n = 4). Mann-Whitney testing with correction for multiple testing did not reveal significant (p< 0.05) differences between healthy donors and CBL-LOH patients. Line indicates the mean of the displayed datapoints. (**e**) Extracellular detection of cytokines by ELISA. Healthy donors (n = 6), CBL-LOH patients (n = 3). Mann-Whitney testing with correction for multiple testing did not reveal significant (p< 0.05) differences between healthy donors and CBL-LOH patients. Line indicates the mean of the displayed datapoints.

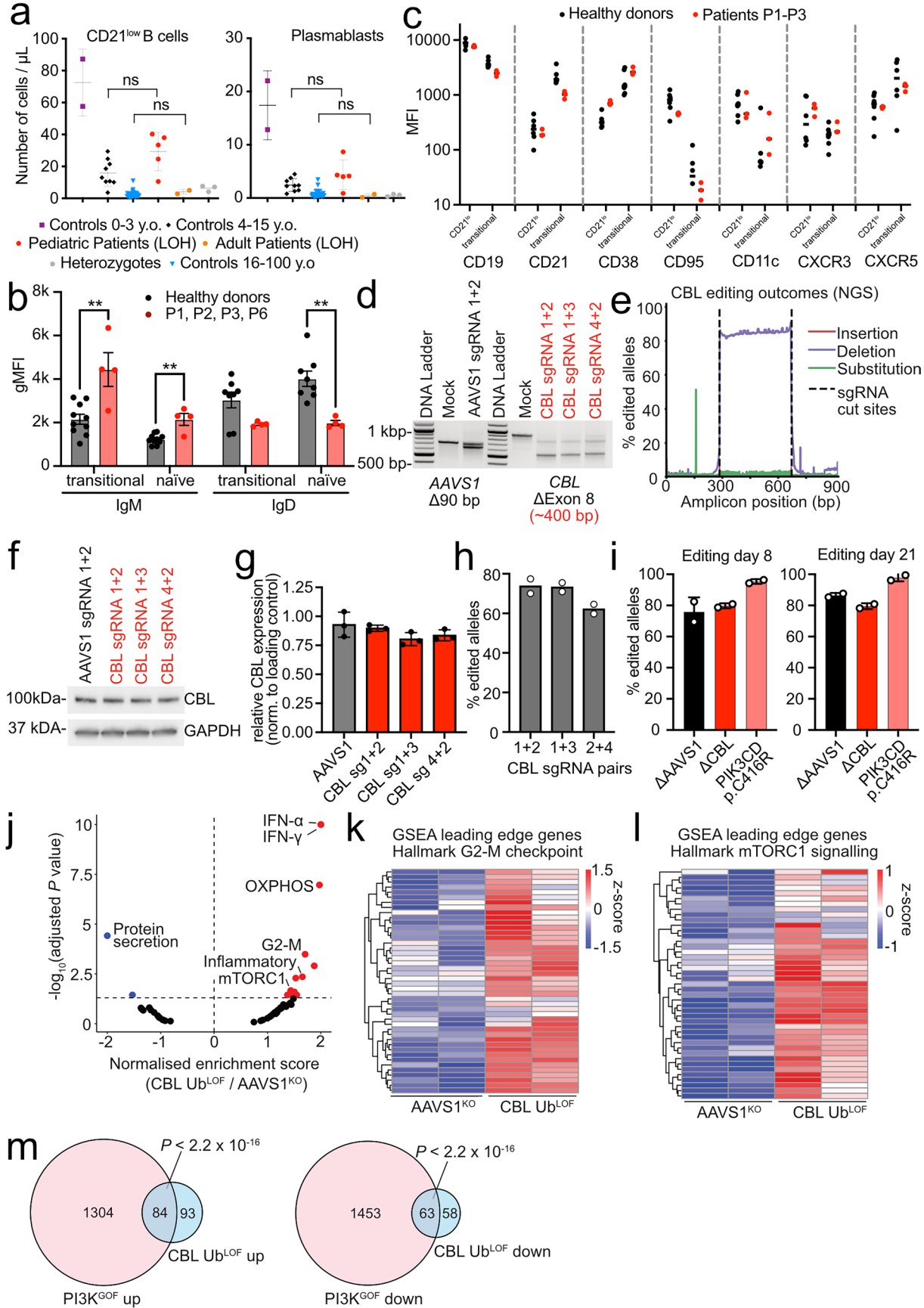

**Extended Data Fig. 4 | See next page for caption.**

**Extended Data Fig. 4 | Characterization of B cell biology in CBL deficient participants and cells.** (**a**) Quantification of B-cell subsets in cryopreserved PBMCs from healthy donors, heterozygous carriers, and CBL-LOH patients of the indicated ages by mass cytometry. Controls 0–3 y.o. (n = 2); controls 4–15 y.o. (n = 9); controls 16–100 y.o. (n = 28); pediatric CBL-LOH (n = 5); adult CBL-LOH (n = 2); heterozygous individuals (n = 3). Mean ± s.d. Statistical significance was assessed using two-sided Mann–Whitney tests with correction for multiple testing. (**b**) Frequency of B-cell subsets in healthy donors (n = 10) and CBL-LOH patients (n = 4) by flow cytometry. Mean ± s.d. Significance was evaluated with two-sided Mann–Whitney tests adjusted for multiple comparisons. **p < 0.005. (**c**) MFI of B-cell markers on CD21$^{lo}$ and transitional B cells from healthy donors (n = 6) and patients P1–3 (n = 3). Lines show means. (**d**–**f**) Modeling the CBL ΔExon8 variant in primary human CD34$^+$ HSPCs. (**d**) Agarose gel electrophoresis of PCR products from AAVS1 or CBL loci 72 h after nucleofection; representative of >5 biological replicates. (**e**) Editing efficiency using sgRNA pair 1+2 by NGS, showing ~80% of the ~400 bp exon-8 deletion. (**f**) Western blot of CBL protein levels after CBL or AAVS1 editing. (**g**) Quantification of three biological replicates from (f). Mean ± s.d. (**h**) NGS quantification of exon-8 deletions for all three guide pairs (n = 2 biological replicates). Bars show means. (**i**) Editing efficiencies at days 8 and 21 in differentiation cultures by NGS. Mean ± s.d. from three biological replicates. (**j**–**m**) Bulk RNA-seq of AAVS1- and CBL-edited CD19$^+$CD10$^+$CD20$^{low}$ HSPC-derived B-cell progenitors. (**j**) Gene-set enrichment analysis showing significantly enriched (red) or depleted (blue) pathways (NES: normalized enrichment score). (**k**,**l**) Differential expression of leading-edge genes in the (**k**) Hallmark G2–M checkpoint and (**l**) Hallmark mTORC1 signaling gene sets (Z-transformed normalized counts). (**m**) Transcriptional overlap between CBL-edited and PI3K$^{GOF}$ progenitors; Venn diagrams show shared significantly up- or downregulated genes, with overlap significance by binomial test.

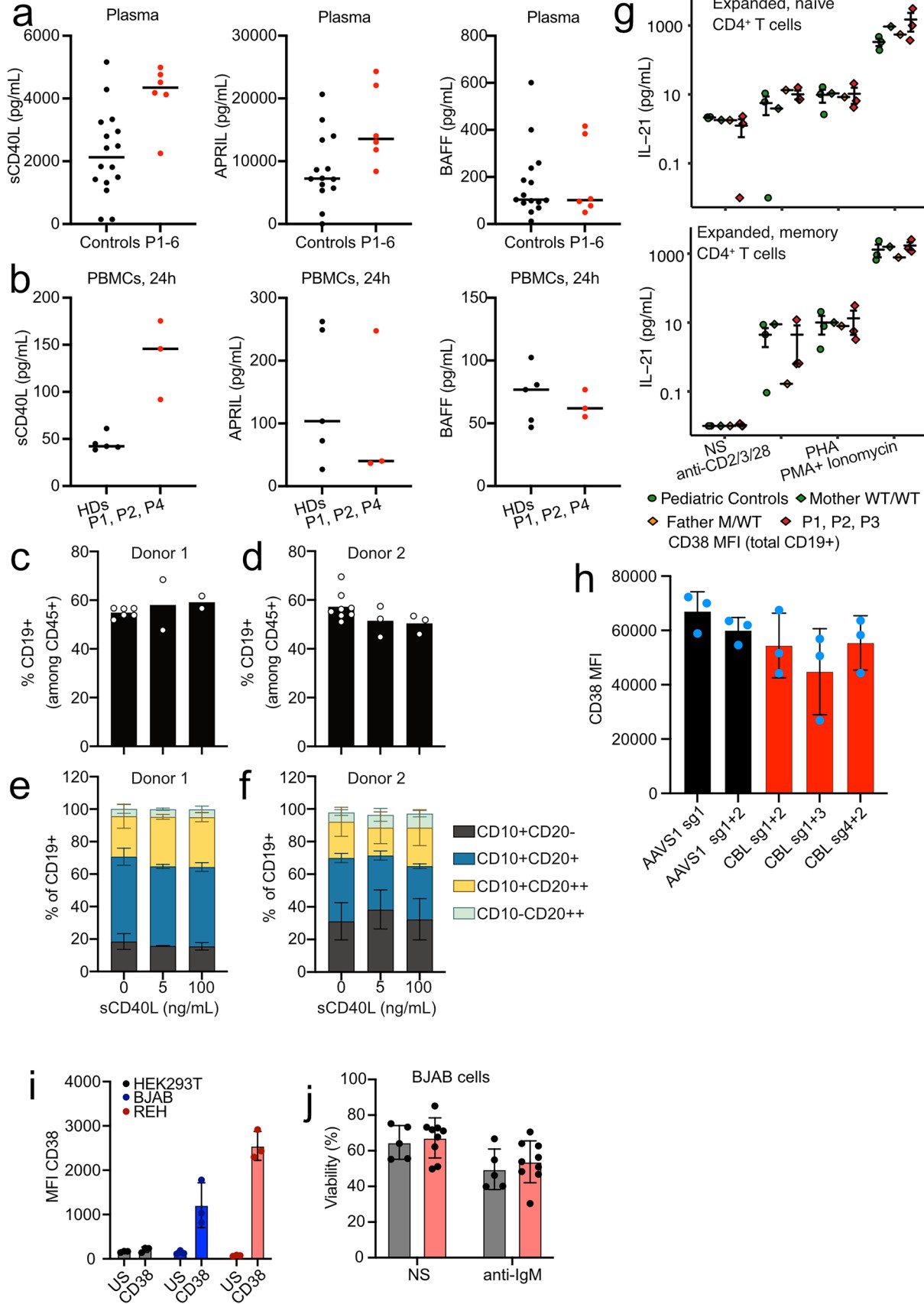

**Extended Data Fig. 5 | See next page for caption.**

**Extended Data Fig. 5 | Intact cell extrinsic determinants of B cell development and function.** (**a**) Levels of sCD40L, APRIL and BAFF in the plasma of CBL-LOH patients (n = 6) and healthy donors (n = 14). Line shows the mean of the displayed datapoints (one point per individual assessed). (**b**) Quantification of the production of sCD40L, APRIL and BAFF by PBMCs of CBL-LOH patients (n = 3) and healthy donors (n = 5). Line shows the mean of the displayed datapoints (one point per individual assessed). (**c-f**) Impact of sCD40L levels on B cell differentiation in vitro using CD34+ HSPCs. (**c,d**) B cell output at day 21 of co-culture of CD34+ HSPCs from two healthy donors. The MS5 co-culture was supplement with IL-7 (20 ng/mL) and the indicated doses of sCD40L. Mean of dots that represent technical replicates. (**e,f**) Quantification of B cell subsets based on CD10 and CD20 marker expression. Cells were treated as in (**c,d**). Mean ± s.d. (**g**) IL-21 production by sorted CD4+ naïve and memory T cells upon the indicated stimuli. Mean ± s.d. (**h**) Surface staining of CD38 expressed on CD19+ HSPC-derived B cell progenitors edited with the indicated sgRNAs. Mean ± s.d. of three biological replicates. (**i**) Surface staining of CD38 expressed on HEK293T, REH and BJAB cells as determined by flow cytometry. Mean ± s.d. of three biological replicates. (**j**) CBL Y371C KI BJAB (n = 9) cells are not more resistant than WT (n = 5) cells to BCR-induced apoptosis. Mean ± s.d. of clones over three independent experiments.

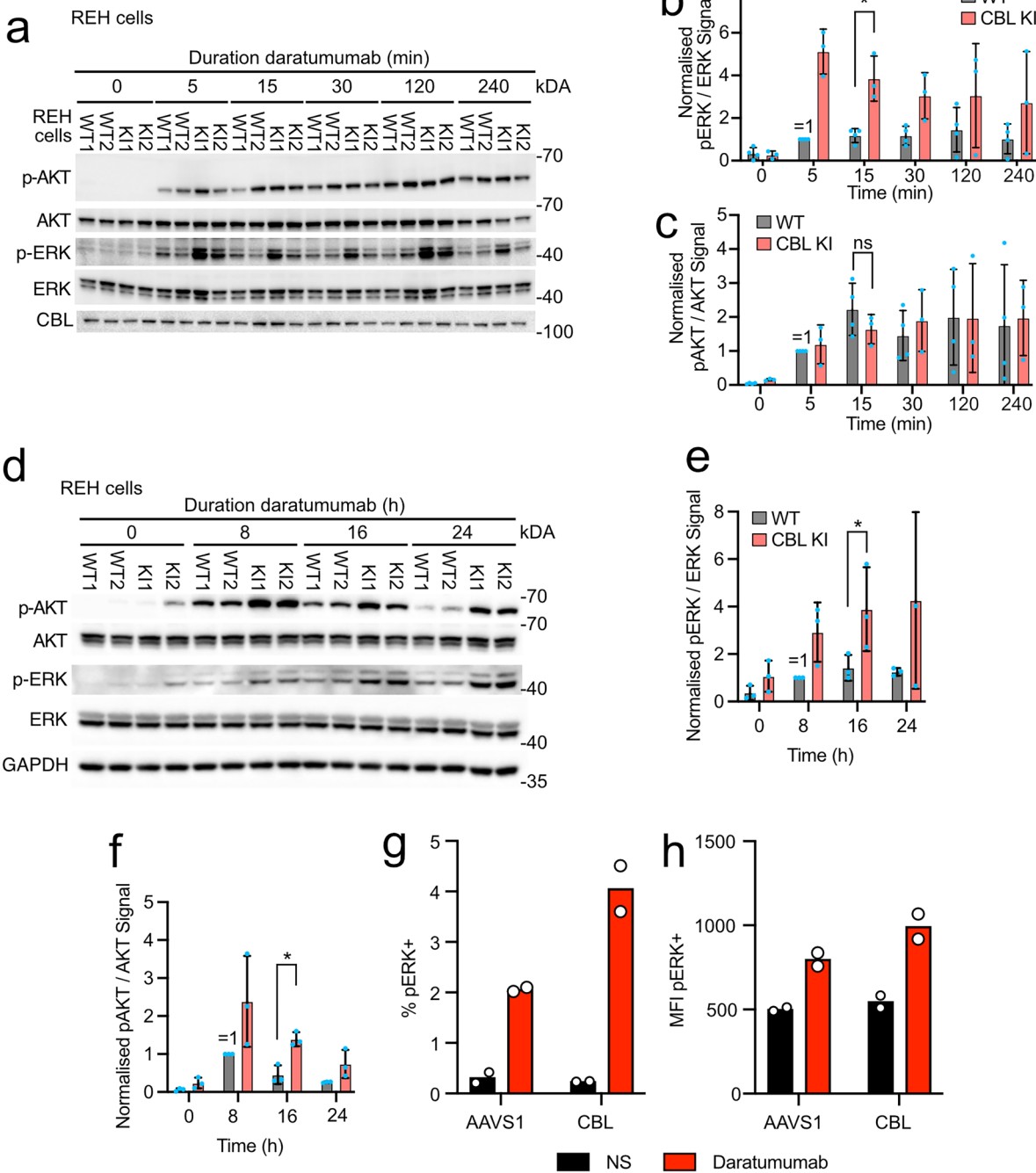

**Extended Data Fig. 6 | Assessment of signalling pathway activation in CBL deficient cells. (a-f)** Western blots of wildtype and CBL Y371C KI REH cells upon CD38 crosslinking with daratumumab for the indicated time periods (min/h). **(b,c, e,f)** Shows quantifications of three biological replicates. Mean ± s.d. Statistical significance was assessed with Mann Whitney tests. *p<0.05. **(g,h)** ERK phosphorylation upon CD38 crosslinking in gene-edited HSPC-derived CD19+ B progenitors. Total CD19+ **(g)** or CD19+CD20high **(h)** B progenitors were sorted from co-cultures edited at the AAVS1 or CBL locus. Cells were stimulated for 15 min with daratumumab, followed by fixation, permeabilization and intracellular staining for phosphorylated ERK (pERK). n = 2 biological replicates. Bars show the mean.

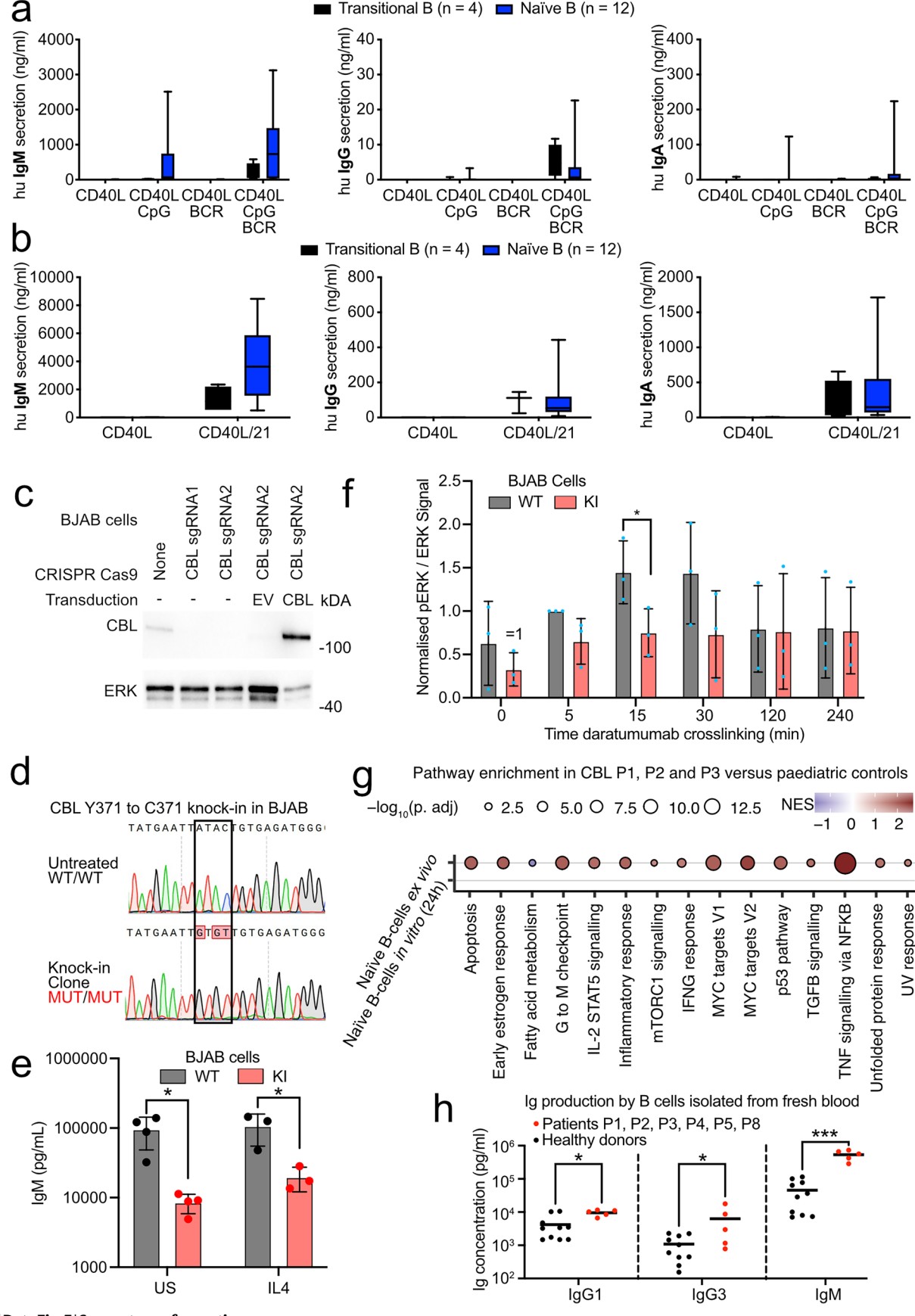

**Extended Data Fig. 7 | See next page for caption.**

**Extended Data Fig. 7 | Functional phenotyping of mature CBL deficient B cells.**
(**a,b**) Transitional B cells (n = 4 independent experiments) and naïve B cells (n = 12 independent experiments) were sort-purified from PBMCs of healthy donors and then cultured in vitro. Levels of IgM, IgG and IgA were measured in supernatants from transitional and naïve B cells after 7 days following stimulation with: (**a**) CD40L alone or in combination with CpG and/or BCR crosslinking, or (**b**) CD40L and IL-21. Box and whiskers indicate median (central line), quartiles (box) and deciles (whiskers). (**c**) Western blot of CBL KO and stable lentiviral overexpression on BJAB cells. (**d**) Sanger genotyping of BJAB CBL Y371C KI and control wildtype cells. (**e**) Ig production of control wildtype and CBL Y371C KI BJAB cell lines within 24 h of culture unstimulated (n = 4 biological replicates) and upon IL-4 stimulation (n = 3 biological replicates). Mean ± s.d. The statistical significance of differences was assessed in multiple two-sided Mann-Whitney

tests, with correction for multiple testing. *p<0.05. (**f**) Quantification of three biological replicates of the western blot shown in Fig. 5e. Mean ± s.d. Statistical significance was assessed with Mann Whitney tests. *p<0.05. (**g**) Pathway enrichment analysis of bulk RNA-sequencing of healthy donor and CBL LOH patients' primary naïve B cells directly after sorting from cryopreserved PBMCs or after 24 h of non-stimulated culture. NES: normalized enrichment ratio. p values were adjusted for multiple testing. (**h**) Immunoglobulin production by sorted primary B cell subsets of healthy donors (n = 10) and *CBL*-LOH patients (n = 5) from fresh blood samples without stimulation. Supernatants were collected after 24 h and Igs were measured by ELISA. Line indicates the mean of the displayed datapoints (one point per individual tested). Statistical significance was assessed using Mann Whitney tests and correction for multiple testing. *p < 0.05, ***p< 0.0005.

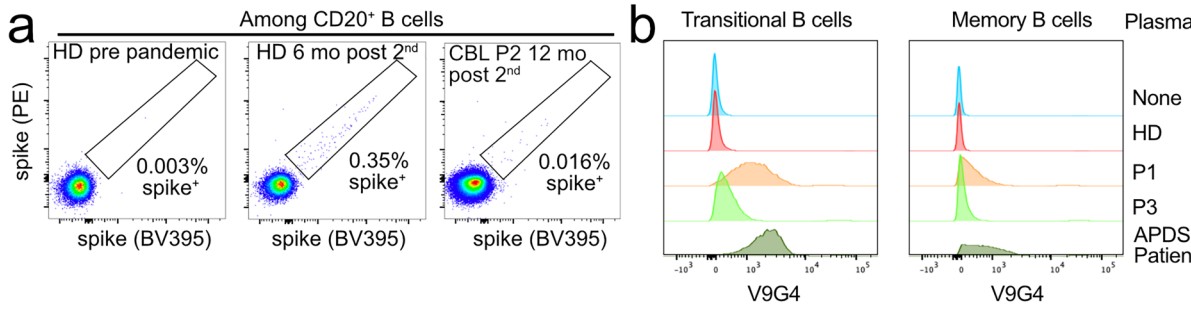

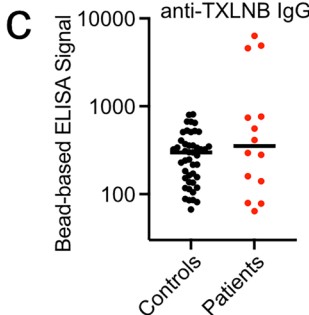

**Extended Data Fig. 8 | Profiling of humoral immunity and autoimmunity.** (**a**) Flow cytometry for Spike tetramers on primary B cells of healthy donors and patient P2. (**b**) Cell painting of healthy donor transitional and memory B cells by the indicated plasma samples. (**c**) Validation of presence of anti-TXLNB autoantibodies in three CBL-deficient patients by multiplex bead assay. Line indicates the mean of the displayed datapoints (one point per individual tested).

# Reporting Summary

## Statistics

For all statistical analyses, confirm that the following items are present in the figure legend, table legend, main text, or Methods section.

| n/a | Confirmed | |
|---|---|---|
| ☐ | ☒ | The exact sample size (*n*) for each experimental group/condition, given as a discrete number and unit of measurement |
| ☐ | ☒ | A statement on whether measurements were taken from distinct samples or whether the same sample was measured repeatedly |
| ☐ | ☒ | The statistical test(s) used AND whether they are one- or two-sided<br>*Only common tests should be described solely by name; describe more complex techniques in the Methods section.* |
| ☒ | ☐ | A description of all covariates tested |
| ☒ | ☐ | A description of any assumptions or corrections, such as tests of normality and adjustment for multiple comparisons |
| ☐ | ☒ | A full description of the statistical parameters including central tendency (e.g. means) or other basic estimates (e.g. regression coefficient) AND variation (e.g. standard deviation) or associated estimates of uncertainty (e.g. confidence intervals) |
| ☐ | ☒ | For null hypothesis testing, the test statistic (e.g. *F*, *t*, *r*) with confidence intervals, effect sizes, degrees of freedom and *P* value noted<br>*Give P values as exact values whenever suitable.* |
| ☒ | ☐ | For Bayesian analysis, information on the choice of priors and Markov chain Monte Carlo settings |
| ☒ | ☐ | For hierarchical and complex designs, identification of the appropriate level for tests and full reporting of outcomes |
| ☒ | ☐ | Estimates of effect sizes (e.g. Cohen's *d*, Pearson's *r*), indicating how they were calculated |

*Our web collection on statistics for biologists contains articles on many of the points above.*

## Software and code

Policy information about availability of computer code

| Data collection | No applicable. |
|---|---|
| Data analysis | Graphpad Prism, Flow Jo, Microsoft Excel |

For manuscripts utilizing custom algorithms or software that are central to the research but not yet described in published literature, software must be made available to editors and reviewers. We strongly encourage code deposition in a community repository (e.g. GitHub). See the Nature Portfolio guidelines for submitting code & software for further information.

## Data

Policy information about availability of data

All manuscripts must include a data availability statement. This statement should provide the following information, where applicable:
- Accession codes, unique identifiers, or web links for publicly available datasets
- A description of any restrictions on data availability
- For clinical datasets or third party data, please ensure that the statement adheres to our policy

All raw data for bulk RNA-seq on primary naïve B cells (GSE307942) and HSPC-derived B cell precursors (GSE307131) have been deposited in the GEO repository and will be made available as of publication date. BCR-sequencing data have been deposited on SRA (PRJNA1328925) and are available as of publication. All raw

data and resources will be made available upon request to the corresponding authors.
There is no original code generated in this study.

# Research involving human participants, their data, or biological material

Policy information about studies with human participants or human data. See also policy information about sex, gender (identity/presentation), and sexual orientation and race, ethnicity and racism.

| Reporting on sex and gender | This cohort includes both female and male patients, with no distinct phenotype segregation based on sex. Healthy controls of both sexes were also recruited, and no significant differences related to sex were observed. |
|---|---|
| Reporting on race, ethnicity, or other socially relevant groupings | No information regarding race or ethnicity was collected or analyzed. |
| Population characteristics | Not applicable. This studies recruitment is based on the presence of rare genetic variants and the size of the cohort is limited. |
| Recruitment | Patients were recruited due to presence of deleterious variants in the CBL gene by clincans involved in the CBL consortium. |
| Ethics oversight | Informed consent was obtained from patients in their respective countries of residence (Iran, Italy, France, Spain, and Germany) in accordance with local regulations and with approval from the relevant institutional review boards (IRB). Physicians responsible for patient care completed a comprehensive questionnaire that recorded demographic data, clinical features, and biological and microbiological findings. No information regarding patients' gender or socioeconomic status was collected. Experiments were carried out in Australia, France, Germany, Sweden, and the United States of America, adhering to local regulations and with IRB approval from Rockefeller University (protocol no. JCA-0699) and INSERM (protocol no. C10-07 and C10-16) for the United States and France, respectively. Healthy donors were recruited from France, Spain, Italy, the United States of America, and Australia. This study was approved by the Sydney Local Health District RPAH Zone Human Research Ethics Committee and Research Governance Office, Royal Prince Alfred Hospital, Camperdown, NSW, Australia (Protocol X16-0210/LNR/16/RPAH/257). |

Note that full information on the approval of the study protocol must also be provided in the manuscript.

# Field-specific reporting

Please select the one below that is the best fit for your research. If you are not sure, read the appropriate sections before making your selection.

☒ Life sciences ☐ Behavioural & social sciences ☐ Ecological, evolutionary & environmental sciences

For a reference copy of the document with all sections, see nature.com/documents/nr-reporting-summary-flat.pdf

# Life sciences study design

All studies must disclose on these points even when the disclosure is negative.

| Sample size | The cohort comprises 11 patients, which is a relatively large cohort for such a rare genetic disease. In each experiment and when possible, at least three,but usually more patients were included. |
|---|---|
| Data exclusions | None |
| Replication | Experiments were reproduced in typically three biological replicates. Assessment of different patients with the same genotype was interpreted as biological replicates. |
| Randomization | Randomization was not feasible in a study of rare genetic disease like this. |
| Blinding | Blinding was carried out when possible (for example when collaborators analyzed BCR-seq data. Often, blinding was not feasible as researchers were working with primary material from patients, blinding of these samples in a clinical research context was not feasible. |

# Reporting for specific materials, systems and methods

We require information from authors about some types of materials, experimental systems and methods used in many studies. Here, indicate whether each material, system or method listed is relevant to your study. If you are not sure if a list item applies to your research, read the appropriate section before selecting a response.

## Materials & experimental systems

| n/a | Involved in the study |
|---|---|
| ☐ | ☒ Antibodies |
| ☐ | ☒ Eukaryotic cell lines |
| ☒ | ☐ Palaeontology and archaeology |
| ☒ | ☐ Animals and other organisms |
| ☐ | ☒ Clinical data |
| ☒ | ☐ Dual use research of concern |
| ☒ | ☐ Plants |

## Methods

| n/a | Involved in the study |
|---|---|
| ☒ | ☐ ChIP-seq |
| ☐ | ☒ Flow cytometry |
| ☒ | ☐ MRI-based neuroimaging |

## Antibodies

| | |
|---|---|
| Antibodies used | Antibodies used in the study are listed in the methods section of the manuscript, including their references, dilutions etc. |
| Validation | Antibodies used were typically well validated antibodies in the field. CBL antibody was validated by western blotting on a CBL knockout cell line. |

## Eukaryotic cell lines

Policy information about cell lines and Sex and Gender in Research

| | |
|---|---|
| Cell line source(s) | Cell lines were obtained from National collections like the ATCC HEK293T and REH) and DSMZ (BJAB). |
| Authentication | Authenticated cell lines were obtained directly form cell banks. The different laboratories that participated in this research use different commercial services for cell line authentication (e.g. Eurofinsgenomics), the cell lines used here were obtained recently from the cell banks and therefore not specifically re-authenticated when this research was performed. |
| Mycoplasma contamination | Presence of mycoplasma was regularly assessed by enzymatic and qPCR assays in house. All cell lines used in the study tested negative for mycoplasma at the time of the experiments. |
| Commonly misidentified lines (See ICLAC register) | HEK293T cells for lentivirus production and CD38 surface staining. |

## Clinical data

Policy information about clinical studies
All manuscripts should comply with the ICMJE guidelines for publication of clinical research and a completed CONSORT checklist must be included with all submissions.

| | |
|---|---|
| Clinical trial registration | Not applicable |
| Study protocol | JCA-0699, C10-07 and C10-16 |
| Data collection | See above. |
| Outcomes | Not applicable. |

## Plants

| | |
|---|---|
| Seed stocks | None |
| Novel plant genotypes | None |
| Authentication | None |

# Flow Cytometry

## Plots

Confirm that:

☒ The axis labels state the marker and fluorochrome used (e.g. CD4-FITC).

☒ The axis scales are clearly visible. Include numbers along axes only for bottom left plot of group (a 'group' is an analysis of identical markers).

☒ All plots are contour plots with outliers or pseudocolor plots.

☒ A numerical value for number of cells or percentage (with statistics) is provided.

## Methodology

| | |
|---|---|
| Sample preparation | All described in the methods section of the manuscript. |
| Instrument | BD Fortessa, Aria II, Novocyte |
| Software | Flow Jo |
| Cell population abundance | Indicated in the figures and methods section |
| Gating strategy | Described in the methods section and figure legends. |

☒ Tick this box to confirm that a figure exemplifying the gating strategy is provided in the Supplementary Information.

