## [Peer Review File · Nature Immunology]

Somatic deficiency of human E3 ubiquitin ligase CBL in leukocytes impairs B cell but not T cell development and function

Corresponding Author: Dr Jonathan Bohlen

Version 0:

Reviewer comments:

Reviewer #1

(Remarks to the Author)

In their paper, Vatovec and colleagues provide a characterization of 10 patients homozygous for loss of function variants of CBL. They conclude that, unlike mice, CBL plays an important role in the development, tolerance and activation of B cells. In contrast, the defects in T cells are mild. This paper both provides important insights into CBL function and illustrates the limitations of mechanistic studies in mice.

The results presented in this paper represent a monumental, international effort. Indeed, there are many outstanding aspects to the primary human studies including the genetic studies and the extensive characterization of both bone marrow and peripheral blood samples. Indeed, it would be difficult to ask for additional experiments on primary human samples.

The primary human characterization is complemented by in vitro models of both B cell development and peripheral activation. While these studies provide important mechanistic insights, they could be improved. In particular, the effect of CBL LOF on relevant signaling pathways is under-explored. While they only assay ERK activation, they make frequent reference to a previous study of patients expressing a GOF p110delta mutant manifesting enhanced PI3K activation. However, they do not assay for PI3K activation in their systems. For example, ERK and PI3K play complementary but very distinct roles across the pre-BI and pre-BII cell stages (at least in mice). In their in vitro system of B cell development, they could sort these cell populations and do RNA-seq with paired-end sequencing. They could then examine ERK and PI3K signaling pathways relative to proliferative pathways and light chain recombination. Ideally, they would have a GOF p110delta mutant control. However, I leave it to the authors on how they would like to assay for PI3K activation.

Specific comments:

1. on page 11, they should show proliferation results from long-term cultures or just delete this statement
2. Did the authors stain for IgM and IgD, especially in the transitional stages? Was there an increase in a putative anergic B cell pool? This was a bit unclear in the manuscript.
3. They showed representative ERK blots but they need to provide quantitation across at least three experiments. This needs to be done for both 4I and 5E. It would be particularly useful for interpreting late time points in 5E.
4. The referencing is incomplete, XXXs in the methods, PMIDs in the discussion. The references are in a different font.

Reviewer #2

(Remarks to the Author)

In this paper, Vatovec and Bohlen, et al. reported their findings of defects in B cell development and function in a cohort of CBL LOH variants patients. Their data has revealed that the hematopoietic CBL mutation does not influence T cells development and function. Instead, it impairs the transitional, immature B cell development into mature and memory B cells. The differentiation of B cells to antibody secreting cells is also impaired in the absence of CBL. Additionally, mechanistic studies demonstrate that the mutation leads to an upregulation of CD38 expression on the mutant immature B cells and impaired CD38 induced apoptosis. Moreover, they have shown that these patients possess high titers of autoantibodies and

autoreactive BCR-expressing B cells, indicating that the immune tolerance of B cells is broken.

This paper has reported an interesting and novel finding from a cohort of very interesting, rare immune disorder patients. Their data not only reveals an important, differential role of CBL in human B and T cell development and function but also presents another example that the same gene may play distinct roles in human and mice. However, I also feel that some hypotheses, experimental designs, and interpretations of the results are not very compelling and sometimes contradictory. In addition, there is no in depth mechanistical studies showing how the CBL deficiency affects B cell maturation and tolerance. Moreover, statistics in some figures are missing; however, this is perhaps due to the limited access to human specimens. The following major concerns need to be addressed, which may help to improve the overall quality of the paper:

1) Authors described that CBL LOH patients had histories of repeated severe infections. However, Fig. 2A-B shows similar levels of serum Abs against various microbes in homozygosity CBLLOF patients and age-matched pediatric donors. Does this suggest that patients' long term antibody responses are normal and the unusual severe infections are not caused by the alteration in patient's B cell compartments? If so, this observation seems to be contradictory to the claim that B cell function is impaired in these patients.

2) Authors concluded that CBL LOH variant T cells retained normal functions, based on T cell proliferation upon TCR/CD28 stimulation, normal STAT5 phosphorylation after IL2 and IL27 stimulation, and INF-gamma and TNF production (Fig. 2 and related sFigs). However, antibody responses to microbe infections depend on Tfh cells. To claim that the LOH mutation does not affect T cell function in the context of pathogen infections, they should at least show that the differentiation and function of CBL LOH Tfh cells were not altered, given that CBL and CBLB proteins have been shown recently to negatively regulate the development and function of Tfh cells (DOI: 10.1016/j.immuni.2024.04.023)

3) Authors showed that transitional, immature, and memory B cells in young LOH patients were expanded and patients were hypergammaglobulinemia (Fig. 3A-C). However, plasmablasts were not increased relative to healthy controls. If the total number of plasmablasts, perhaps also PCs, are not increased, how can the excess amounts of antibodies be produced in these patients? Fig. 3D only shows the proportions of each cell subsets. How about absolute numbers?

4) The HSPC culture experiments (Fig. 4A-C) showed only a proportional increase of immature B cells in CBL mutant cell cultures relative to controls, similar to that found in the BM of CBL LOH patients. The authors should show the absolute numbers of each cell subsets because the difference could be caused by a reduction in the total number of PreB I cells.

5) As having been mentioned by authors that CD38 signaling is one of the triggers to induce apoptosis of immature B cells. Since CD38 is upregulated in CBL LOH immature B cells, they hypothesize that the CBL LOH mutation attenuates CD38 induced apoptosis of immature B cells, consequently impairing the immune tolerance of B cells. To test this hypothesis, they first generated CBL LOH HSPCs-derived B cells and showed that mutant B cells were less susceptible to CD38 induced apoptosis as compared to WT controls. This finding indicates that the CBL LOH mutation attenuates CD38 signaling in B cells. However, when they examined CD38 signaling in Y371C mutant REH cell lines, it was found that ERK signaling was elevated in WT cells and significantly enhanced in the mutant cells after CD38 stimulation (Fig. 4I). This observation seems to raise a paradoxical role of CD38-ERK axis in B cell tolerance, because in WT cells activation of CD38-ERK signaling is interpreted as apoptotic whereas in CBL mutant cells enhanced CD38-ERK signaling is anti-apoptotic. In this regard, they should look at right signaling pathways, rather than ERK, for the signaling insights that are responsible for CBL mutant B cell apoptosis and immune tolerance. Does CD38 activate PIK3CD? Since PIK3CD is involved in B cell apoptosis and PIK3CD GOF and CBL LOH mutations exert a similar effect on B cell development, it is surprising to me why they did not directly compare PIK3CD activities? Additionally, since BCR-induced apoptosis of immature B cells is the main mechanism to delete autoreactive B cells, it will make more sense to look at BCR-induced apoptosis and signaling of immature B cells in this model.

6) To understand the influence of extrinsic factors that might influence B cell development, authors examined soluble factors in patient's serum samples and HSPC cultures. They found that sCD40L, but not APRIL and BAFF, was significantly upregulated as compared to controls (sFig 5A-B) (no statistics was provided). Is the increased sCD40L secreted by activated T cells? If so, it will suggest that the function of CBL LOH T cells is altered. Additionally, is it possible that the increased CD40L causes the expansion of CBL LOH immature B cells, since CD40L may stimulate B cell proliferation? It is also noted that authors have examined Ig secretion by WT and mutant HSPC-derived B cells after CD40L stimulation (Fig. 5A), however, B cell proliferation is not mentioned.

7) Authors have shown that mutant B cells are defective in antibody secretion in vitro after CD40L stimulation (Fig. 5A). However, there is no experiment showing the development of plasma cells. This is an important question to clarify because if the development of CBL LOH B cells to plasma cell is deficient, why patients are hypergammaglobulinemia (Fig. 3B)?

8) CBL LOH patients exhibited a breakdown of B cell immune tolerance, as they produced high titers of autoantibodies against multiple autoantigens. Increased autoreactive B cells were verified by flowcytometry using V9G4 anti-ideotype antibodies, as V9G4hi B cells were increased in LOH patients relative to healthy controls. Since the mutant B cells could have acquired an increased capability to bind non-specifically to the V9G4 antibody, it would be better to provide VH gene DNA sequence data to support their claim.

(Remarks to the Author)

This manuscript by Vatovec et al. investigates the role of the Casitas B lineage lymphoma proto-oncogene (CBL) in human lymphocyte development, maturation, and function, focusing on its distinct roles in T and B cells. The study examines patients with somatic homozygosity for loss-of-function (LOF) variants resulting in defective ubiquitination activity (UbLOF). The findings reveal differences between human and murine lymphocyte biology. In contrast to mouse models, where CBL has a significant role in T cell development and function, human CBL appears largely redundant in T cells. Patients with CBL UbLOF variants exhibited no major defects in T cell maturation or antigen responsiveness, highlighting species-specific variations in CBL function.

Conversely, human CBL is essential for B cell development and function. Patients with CBL UbLOF exhibited an increase in transitional B cell numbers during childhood and greater susceptibility to bacterial infections. The study demonstrates that CBL deficiency impairs B cell maturation through a cell-intrinsic mechanism involving reduced apoptosis and dysregulated B cell receptor (BCR) signaling. This dysregulation results in fewer memory B cells and disrupts adaptive immune memory. Additionally, CBL UbLOF alters bone marrow B cell development at the immature stage, promoting the survival and differentiation of autoreactive B cells. This impairment in B cell tolerance is associated with autoantibody production and an increased risk of autoimmune manifestations.

Overall, the study highlights the distinct role of CBL in human B cell biology and its redundancy in T cells. However, there are critical aspects that need to be addressed.

Specific comments:

1- The first two sections describing the patients' history and infectious disease events are overly descriptive and repetitive, making them difficult to follow. The focus on individual patient details, such as specific infectious episodes and their progression, detracts from the broader scientific implications. Additionally, much of this information, such as the occurrence of severe bacterial infections or associated conditions like JMML, has already been published in other reports.

2- Figure 3 presents data from mass cytometry and flow cytometry, but the differing approaches to patient and control comparisons make the figure difficult to interpret. In mass cytometry, the ages of both patients and controls should be clearly indicated in both datasets to facilitate understanding and to highlight any age-related trends more effectively.

In contrast to mass cytometry, the flow cytometry data are shown for patients (P1–P5) without accounting for age, despite these patients being from different age groups. This inconsistency is confusing and obscures potential patterns, particularly as the pediatric group appears to show the highest elevation in B cell subsets in the mass cytometry analysis. When the flow cytometry data are presented without similar age-group stratification, it becomes unclear how these findings align with the trends observed in the mass cytometry data.

3- In Figure 4, the authors investigate the developmental block in B cell differentiation in patients and conclude that the transition from immature to mature B cells is impaired. To validate this finding, they perform a gene editing experiment in CD34+ hematopoietic progenitor cells, aiming to delete exon 8 of CBL. This deletion is intended to create a ubiquitination-deficient CBL protein while preserving the expression of the remaining portions of the protein. However, the data shown in Figure 3C cast doubt on the authors' interpretation. While the authors suggest that CBL exon 8 deficiency leads to a differentiation block, the results do not clearly support this conclusion. In fact, the differentiation of CBL-edited cells appears better than with the AAVS1 control. On the other hand, the data strongly suggest that CBL gene editing affects cell survival. This discrepancy makes it difficult to follow their interpretation.

Further issues are evident in Supplementary Figure 4, where the authors attempt to demonstrate the efficiency of the gene editing. The data suggest that sgRNA 1+3 results in complete CBL downregulation rather than exon editing, while sgRNA 1+2 is more effective in generating the exon 8 deletion. However, the absence of a loading control makes it impossible to confirm whether the rest of the CBL protein is expressed as expected or to evaluate the efficiency and specificity of the editing. Another limitation of the study is the reliance on data from monozygotic siblings (P1, P2, and P3) as key patient representatives. Since these patients share identical genetic and environmental backgrounds, their data may not fully capture the broader variability among individuals with CBL deficiency. Including data from unrelated patients would provide a more comprehensive and generalizable assessment of the effects of CBL deficiency on B cell differentiation and survival.

4- The authors proceed to investigate apoptosis in CBL Ub-LOF patients and conclude that defective apoptosis is the primary reason for the accumulation of immature B cells. However, they rely on gene-edited CD34+ hematopoietic progenitor cells from in vitro differentiation experiments to support this conclusion, despite the issues identified with the gene editing experiment. To further support their hypothesis, the authors use the REH cell line as a model to investigate apoptosis. However, the rationale for selecting this cell line is unclear. REH cells represent progenitor B cells lacking cytoplasmic heavy chain expression, making them fundamentally distinct from the immature B cells that accumulate in CBL Ub-LOF patients. This discrepancy raises questions about the relevance of the REH cell model to the patient phenotype. The authors do not provide sufficient justification for its use or explain how results from REH cells are applicable to the context of CBL Ub-LOF patients. As a result, the connection between defective apoptosis and the observed accumulation of immature B cells remains speculative.

5- The authors investigate antibody secretion in CBL Ub-LOF patients and conclude that B cells from these patients are impaired in their ability to produce immunoglobulins (Ig). However, this interpretation raises several critical issues. First, the observation of hypergammaglobulinemia in CBL Ub-LOF patients (Figure 3B) contradicts the claim of impaired Ig secretion. Hypergammaglobulinemia suggests an overall increase in Ig levels, which is difficult to reconcile with reduced Ig secretion by patient B cells. Second, the authors report reduced IgG/A secretion by transitional B cells, which is biologically problematic. Transitional B cells are immature and have not undergone class switch recombination, making it unlikely that they would secrete IgG or IgA. This raises concerns about the purity of the sorted subsets and whether contaminating mature or memory B cells could be contributing to the observed results.

6- The data using 9G4 and the IgG reactivity against a wide range of human antigens point to increased autoreactivity, several aspects require further clarification. The authors associate the increase in 9G4 staining with autoreactive potential but do not address whether the elevated fluorescence intensity reflects increased BCR expression. Assessing IgM and IgD levels on B cells from patients compared to controls would help determine whether the increased staining reflects changes in BCR density or is solely due to the painting effect of circulating autoantibodies. While the shared autoreactive targets identified through protein microarray analysis suggest common mechanisms underlying autoreactivity, functional validation of these targets is needed to confirm their biological relevance. These additional investigations would help clarify the mechanisms by which CBL Ub-LOF leads to disrupted tolerance and autoreactivity, strengthening the conclusions drawn from this study.

7- The figure legends remain suboptimal and fail to provide sufficient detail to aid interpretation.

Decision Letter:

24th Jan 2025

Dear Jonathan,

Thank you for providing a point-by-point response to the referees' comments on your manuscript entitled, "Somatic deficiency of human CBL in leukocytes impairs B cell but not T cell development and function". As noted previously, while they find your work of considerable potential interest, they have raised quite substantial concerns that must be addressed. In light of these comments, we cannot accept the current manuscript for publication, but would be very interested in considering a revised version that addresses these serious concerns along the lines proposed in your point-by-point rebuttal.

We invite you to submit a substantially revised manuscript, however please bear in mind that we will be reluctant to approach the referees again in the absence of major revisions.

Specifically, the revision should include new experiments as noted in your response to address:

- (1) BCR repertoire analysis
- (2) Short-term analysis of freshly cultured CBL-deficient B cells
- (3) Analysis of cTfh cells in CBL-deficiency

Please include the additional textual clarifications as indicated in your response letter.

Please note that we are overruling the requests for a phosphoproteomics/RNA-seq analysis of the REH/BJAB cell lines; mutagenesis of the MN60 cell line, and functional validation of autoantibodies observed in the patient samples.

When you revise your manuscript, please take into account all reviewer and editor comments, please highlight all changes in the manuscript text file in Microsoft Word format.

* If you have not done so already please begin to revise your manuscript so that it conforms to our Article format instructions at <http://www.nature.com/ni/authors/index.html>. Refer also to any guidelines provided in this letter.

The Reporting Summary can be found here:

When submitting the revised version of your manuscript, please pay close attention to our <https://www.nature.com/nature-portfolio/editorial-policies/image-integrity> Digital Image Integrity Guidelines and to the following points below:

- that unprocessed scans are clearly labelled and match the gels and western blots presented in figures.
- that control panels for gels and western blots are appropriately described as loading on sample processing controls

-- all images in the paper are checked for duplication of panels and for splicing of gel lanes.

Extended Data figures and tables are online-only (appearing in the online PDF and full-text HTML version of the paper), peer-reviewed display items that provide essential background to the Article but are not included in the printed version of the paper due to space constraints or being of interest only to a few specialists. A maximum of ten Extended Data display items (figures and tables) is typically permitted. When re-submitting your manuscript, please ensure that any supplementary figures and tables that are more critical to the manuscript's conclusions are converted to Extended data to increase these data's visibility.

Link Redacted

If you wish to submit a suitably revised manuscript we would hope to receive it within 6 months. If you cannot send it within this time, please let us know. We will be happy to consider your revision so long as nothing similar has been accepted for publication at Nature Immunology or published elsewhere.

Nature Immunology is committed to improving transparency in authorship. As part of our efforts in this direction, we are now requesting that all authors identified as 'corresponding author' on published papers create and link their Open Researcher and Contributor Identifier (ORCID) with their account on the Manuscript Tracking System (MTS), prior to acceptance. ORCID helps the scientific community achieve unambiguous attribution of all scholarly contributions. You can create and link your ORCID from the home page of the MTS by clicking on 'Modify my Springer Nature account'. For more information please visit please visit www.springernature.com/orcid.

Thank you for the opportunity to review your work.

Kind regards,

Laurie

Laurie A. Dempsey, Ph.D.
Senior Editor
Nature Immunology
l.dempsey@us.nature.com
ORCID: 0000-0002-3304-796X

Reviewers' Comments:

Reviewer #1 (Remarks to the Author):

In their paper, Vatovec and colleagues provide a characterization of 10 patients homozygous for loss of function variants of CBL. They conclude that, unlike mice, CBL plays an important role in the development, tolerance and activation of B cells. In contrast, the defects in T cells are mild. This paper both provides important insights into CBL function and illustrates the limitations of mechanistic studies in mice.

The results presented in this paper represent a monumental, international effort. Indeed, there are many outstanding aspects to the primary human studies including the genetic studies and the extensive characterization of both bone marrow and peripheral blood samples. Indeed, it would be difficult to ask for additional experiments on primary human samples.

The primary human characterization is complemented by in vitro models of both B cell development and peripheral activation. While these studies provide important mechanistic insights, they could be improved. In particular, the effect of CBL LOF on relevant signaling pathways is under-explored. While they only assay ERK activation, they make frequent reference to a previous study of patients expressing a GOF p110delta mutant manifesting enhanced PI3K activation. However, they do not assay for PI3K activation in their systems. For example, ERK and PI3K play complementary but very distinct roles across the pre-BI and pre-BII cell stages (at least in mice). In their in vitro system of B cell development, they

could sort these cell populations and do RNA-seq with paired-end sequencing. They could then examine ERK and PI3K signaling pathways relative to proliferative pathways and light chain recombination. Ideally, they would have a GOF p110delta mutant control. However, I leave it to the authors on how they would like to assay for PI3K activation.

Specific comments:

1. on page 11, they should show proliferation results from long-term cultures or just delete this statement
2. Did the authors stain for IgM and IgD, especially in the transitional stages? Was there an increase in a putative anergic B cell pool? This was a bit unclear in the manuscript.
3. They showed representative ERK blots but they need to provide quantitation across at least three experiments. This needs to be done for both 4I and 5E. It would be particularly useful for interpreting late time points in 5E.
4. The referencing is incomplete, XXXs in the methods, PMIDs in the discussion. The references are in a different font.

Reviewer #2 (Remarks to the Author):

In this paper, Vatovec and Bohlen, et al. reported their findings of defects in B cell development and function in a cohort of CBL LOH variants patients. Their data has revealed that the hematopoietic CBL mutation does not influence T cells development and function. Instead, it impairs the transitional, immature B cell development into mature and memory B cells. The differentiation of B cells to antibody secreting cells is also impaired in the absence of CBL. Additionally, mechanistic studies demonstrate that the mutation leads to an upregulation of CD38 expression on the mutant immature B cells and impaired CD38 induced apoptosis. Moreover, they have shown that these patients possess high titers of autoantibodies and autoreactive BCR-expressing B cells, indicating that the immune tolerance of B cells is broken.

This paper has reported an interesting and novel finding from a cohort of very interesting, rare immune disorder patients. Their data not only reveals an important, differential role of CBL in human B and T cell development and function but also presents another example that the same gene may play distinct roles in human and mice. However, I also feel that some hypotheses, experimental designs, and interpretations of the results are not very compelling and sometimes contradictory. In addition, there is no in depth mechanistical studies showing how the CBL deficiency affects B cell maturation and tolerance. Moreover, statistics in some figures are missing; however, this is perhaps due to the limited access to human specimens. The following major concerns need to be addressed, which may help to improve the overall quality of the paper:

- 1) Authors described that CBL LOH patients had histories of repeated severe infections. However, Fig. 2A-B shows similar levels of serum Abs against various microbes in homozygosity CBLLOF patients and age-matched pediatric donors. Does this suggest that patients' long term antibody responses are normal and the unusual severe infections are not caused by the alteration in patient's B cell compartments? If so, this observation seems to be contradictory to the claim that B cell function is impaired in these patients.
- 2) Authors concluded that CBL LOH variant T cells retained normal functions, based on T cell proliferation upon TCR/CD28 stimulation, normal STAT5 phosphorylation after IL2 and IL27 stimulation, and INF-gamma and TNF production (Fig. 2 and related sFigs). However, antibody responses to microbe infections depend on Tfh cells. To claim that the LOH mutation does not affect T cell function in the context of pathogen infections, they should at least show that the differentiation and function of CBL LOH Tfh cells were not altered, given that CBL and CBLB proteins have been shown recently to negatively regulate the development and function of Tfh cells (DOI: 10.1016/j.immuni.2024.04.023)
- 3) Authors showed that transitional, immature, and memory B cells in young LOH patients were expanded and patients were hypergammaglobulinemia (Fig. 3A-C). However, plasmablasts were not increased relative to healthy controls. If the total number of plasmablasts, perhaps also PCs, are not increased, how can the excess amounts of antibodies be produced in these patients? Fig. 3D only shows the proportions of each cell subsets. How about absolute numbers?
- 4) The HSPC culture experiments (Fig. 4A-C) showed only a proportional increase of immature B cells in CBL mutant cell cultures relative to controls, similar to that found in the BM of CBL LOH patients. The authors should show the absolute numbers of each cell subsets because the difference could be caused by a reduction in the total number of PreB I cells.
- 5) As having been mentioned by authors that CD38 signaling is one of the triggers to induce apoptosis of immature B cells. Since CD38 is upregulated in CBL LOH immature B cells, they hypothesize that the CBL LOH mutation attenuates CD38 induced apoptosis of immature B cells, consequently impairing the immune tolerance of B cells. To test this hypothesis, they first generated CBL LOH HSPCs-derived B cells and showed that mutant B cells were less susceptible to CD38 induced apoptosis as compared to WT controls. This finding indicates that the CBL LOH mutation attenuates CD38 signaling in B cells. However, when they examined CD38 signaling in Y371C mutant REH cell lines, it was found that ERK signaling was elevated in WT cells and significantly enhanced in the mutant cells after CD38 stimulation (Fig. 4I). This observation seems to raise a paradoxical role of CD38-ERK axis in B cell tolerance, because in WT cells activation of CD38-ERK signaling is interpreted as apoptotic whereas in CBL mutant cells enhanced CD38-ERK signaling is anti-apoptotic. In this regard, they should look at right signaling pathways, rather than ERK, for the signaling insights that are responsible for CBL mutant B cell apoptosis and immune tolerance. Does CD38 activate PIK3CD? Since PIK3CD is involved in B cell apoptosis and PIK3CD GOF and CBL LOH mutations exert a similar effect on B cell development, it is surprising to me why they did not directly compare PIK3CD activities? Additionally, since BCR-induced apoptosis of immature B cells is the main mechanism to

delete autoreactive B cells, it will make more sense to look at BCR-induced apoptosis and signaling of immature B cells in this model.

6) To understand the influence of extrinsic factors that might influence B cell development, authors examined soluble factors in patient's serum samples and HSPC cultures. They found that sCD40L, but not APRIL and BAFF, was significantly upregulated as compared to controls (sFig 5A-B) (no statistics was provided). Is the increased sCD40L secreted by activated T cells? If so, it will suggest that the function of CBL LOH T cells is altered. Additionally, is it possible that the increased CD40L causes the expansion of CBL LOH immature B cells, since CD40L may stimulate B cell proliferation? It is also noted that authors have examined Ig secretion by WT and mutant HSPC-derived B cells after CD40L stimulation (Fig. 5A), however, B cell proliferation is not mentioned.

7) Authors have shown that mutant B cells are defective in antibody secretion in vitro after CD40L stimulation (Fig. 5A). However, there is no experiment showing the development of plasma cells. This is an important question to clarify because if the development of CBL LOH B cells to plasma cell is deficient, why patients are hypergammaglobulinemia (Fig. 3B)?

8) CBL LOH patients exhibited a breakdown of B cell immune tolerance, as they produced high titers of autoantibodies against multiple autoantigens. Increased autoreactive B cells were verified by flowcytometry using V9G4 anti-ideotype antibodies, as V9G4hi B cells were increased in LOH patients relative to healthy controls. Since the mutant B cells could have acquired an increased capability to bind non-specifically to the V9G4 antibody, it would be better to provide VH gene DNA sequence data to support their claim.

Reviewer #3 (Remarks to the Author):

This manuscript by Vatovec et al. investigates the role of the Casitas B lineage lymphoma proto-oncogene (CBL) in human lymphocyte development, maturation, and function, focusing on its distinct roles in T and B cells. The study examines patients with somatic homozygosity for loss-of-function (LOF) variants resulting in defective ubiquitination activity (UbLOF). The findings reveal differences between human and murine lymphocyte biology. In contrast to mouse models, where CBL has a significant role in T cell development and function, human CBL appears largely redundant in T cells. Patients with CBL UbLOF variants exhibited no major defects in T cell maturation or antigen responsiveness, highlighting species-specific variations in CBL function.

Conversely, human CBL is essential for B cell development and function. Patients with CBL UbLOF exhibited an increase in transitional B cell numbers during childhood and greater susceptibility to bacterial infections. The study demonstrates that CBL deficiency impairs B cell maturation through a cell-intrinsic mechanism involving reduced apoptosis and dysregulated B cell receptor (BCR) signaling. This dysregulation results in fewer memory B cells and disrupts adaptive immune memory. Additionally, CBL UbLOF alters bone marrow B cell development at the immature stage, promoting the survival and differentiation of autoreactive B cells. This impairment in B cell tolerance is associated with autoantibody production and an increased risk of autoimmune manifestations.

Overall, the study highlights the distinct role of CBL in human B cell biology and its redundancy in T cells. However, there are critical aspects that need to be addressed.

Specific comments:

1- The first two sections describing the patients' history and infectious disease events are overly descriptive and repetitive, making them difficult to follow. The focus on individual patient details, such as specific infectious episodes and their progression, detracts from the broader scientific implications. Additionally, much of this information, such as the occurrence of severe bacterial infections or associated conditions like JMML, has already been published in other reports.

2- Figure 3 presents data from mass cytometry and flow cytometry, but the differing approaches to patient and control comparisons make the figure difficult to interpret. In mass cytometry, the ages of both patients and controls should be clearly indicated in both datasets to facilitate understanding and to highlight any age-related trends more effectively. In contrast to mass cytometry, the flow cytometry data are shown for patients (P1–P5) without accounting for age, despite these patients being from different age groups. This inconsistency is confusing and obscures potential patterns, particularly as the pediatric group appears to show the highest elevation in B cell subsets in the mass cytometry analysis. When the flow cytometry data are presented without similar age-group stratification, it becomes unclear how these findings align with the trends observed in the mass cytometry data.

3- In Figure 4, the authors investigate the developmental block in B cell differentiation in patients and conclude that the transition from immature to mature B cells is impaired. To validate this finding, they perform a gene editing experiment in CD34+ hematopoietic progenitor cells, aiming to delete exon 8 of CBL. This deletion is intended to create a ubiquitination-deficient CBL protein while preserving the expression of the remaining portions of the protein. However, the data shown in Figure 3C cast doubt on the authors' interpretation. While the authors suggest that CBL exon 8 deficiency leads to a differentiation block, the results do not clearly support this conclusion. In fact, the differentiation of CBL-edited cells appears better than with the AAVS1 control. On the other hand, the data strongly suggest that CBL gene editing affects cell survival. This discrepancy makes it difficult to follow their interpretation.

Further issues are evident in Supplementary Figure 4, where the authors attempt to demonstrate the efficiency of the gene editing. The data suggest that sgRNA 1+3 results in complete CBL downregulation rather than exon editing, while sgRNA 1+2 is more effective in generating the exon 8 deletion. However, the absence of a loading control makes it impossible to confirm whether the rest of the CBL protein is expressed as expected or to evaluate the efficiency and specificity of the editing. Another limitation of the study is the reliance on data from monozygotic siblings (P1, P2, and P3) as key patient

representatives. Since these patients share identical genetic and environmental backgrounds, their data may not fully capture the broader variability among individuals with CBL deficiency. Including data from unrelated patients would provide a more comprehensive and generalizable assessment of the effects of CBL deficiency on B cell differentiation and survival.

4- The authors proceed to investigate apoptosis in CBL Ub-LOF patients and conclude that defective apoptosis is the primary reason for the accumulation of immature B cells. However, they rely on gene-edited CD34⁺ hematopoietic progenitor cells from in vitro differentiation experiments to support this conclusion, despite the issues identified with the gene editing experiment. To further support their hypothesis, the authors use the REH cell line as a model to investigate apoptosis. However, the rationale for selecting this cell line is unclear. REH cells represent progenitor B cells lacking cytoplasmic heavy chain expression, making them fundamentally distinct from the immature B cells that accumulate in CBL Ub-LOF patients. This discrepancy raises questions about the relevance of the REH cell model to the patient phenotype. The authors do not provide sufficient justification for its use or explain how results from REH cells are applicable to the context of CBL Ub-LOF patients. As a result, the connection between defective apoptosis and the observed accumulation of immature B cells remains speculative.

5- The authors investigate antibody secretion in CBL Ub-LOF patients and conclude that B cells from these patients are impaired in their ability to produce immunoglobulins (Ig). However, this interpretation raises several critical issues. First, the observation of hypergammaglobulinemia in CBL Ub-LOF patients (Figure 3B) contradicts the claim of impaired Ig secretion. Hypergammaglobulinemia suggests an overall increase in Ig levels, which is difficult to reconcile with reduced Ig secretion by patient B cells. Second, the authors report reduced IgG/A secretion by transitional B cells, which is biologically problematic. Transitional B cells are immature and have not undergone class switch recombination, making it unlikely that they would secrete IgG or IgA. This raises concerns about the purity of the sorted subsets and whether contaminating mature or memory B cells could be contributing to the observed results.

6- The data using 9G4 and the IgG reactivity against a wide range of human antigens point to increased autoreactivity, several aspects require further clarification. The authors associate the increase in 9G4 staining with autoreactive potential but do not address whether the elevated fluorescence intensity reflects increased BCR expression. Assessing IgM and IgD levels on B cells from patients compared to controls would help determine whether the increased staining reflects changes in BCR density or is solely due to the painting effect of circulating autoantibodies. While the shared autoreactive targets identified through protein microarray analysis suggest common mechanisms underlying autoreactivity, functional validation of these targets is needed to confirm their biological relevance. These additional investigations would help clarify the mechanisms by which CBL Ub-LOF leads to disrupted tolerance and autoreactivity, strengthening the conclusions drawn from this study.

7- The figure legends remain suboptimal and fail to provide sufficient detail to aid interpretation.

Version 1:

Reviewer comments:

Reviewer #1

(Remarks to the Author)

the authors have adequately addressed my concerns.

Reviewer #3

(Remarks to the Author)

The authors have made substantial efforts to address the reviewers' concerns, providing new experimental data and clarifying several aspects of the study. The additional analyses on CBL editing efficiency and BCR repertoire are valuable and strengthen the manuscript. However, several key points remain insufficiently supported or require clarification.

While the newly added data in Figure S4, including the improved CBL amplification gel, are satisfactory, the western blot showing CBL protein expression in gene-edited CD34⁺ HSPCs remains unconvincing. The vinculin loading control for the CBL sgRNA (1+3) sample displays a markedly stronger signal than the AAVS1 sgRNA (1+2) control, while the CBL band intensity appears comparable. This pattern is consistent with reduced CBL protein levels in the edited cells, contrary to the authors' statement that CBL expression is unaffected ("Figure R2").

Furthermore, the authors' explanation regarding class switching and antibody secretion by transitional B cells does not resolve the issue. The data presented suggest comparable IgA and IgG secretion by transitional and naïve B cells under the tested conditions. This is biologically unexpected. The earlier report (2006), which is cited as support, described class switching by transitional B cells after stimulation through CD40L + BCR engagement, or CD40L/BCR + IL-10, resulting in low-level Ig production compared with naïve B cells. However, the current study stimulated transitional cells with CD40L + IL-21, making the comparison inappropriate. Clarification is still required as to why transitional B cells in the present experiments show IgA and IgG secretion comparable to naïve B cells.

Finally, the REH cell experiments remain inappropriate for addressing the mechanistic question of impaired apoptosis in CBL-deficient B cells. REH cells are transformed pre-B cells lacking surface BCR expression and thus cannot model the immature B cells implicated in the patient phenotype. High CD38 expression alone does not justify their use as a surrogate,

and signaling dynamics in this leukemic context cannot be extrapolated to normal immature B cells. Consequently, these data do not substantiate the proposed mechanism linking CBL deficiency to defective apoptosis and B cell accumulation.

Decision Letter:

13th Oct 2025

Dear Jonathan,

Thank you for providing your rebuttal comments to the remaining concerns posed by referee #3 for your manuscript entitled "Somatic deficiency of human CBL in leukocytes impairs B cell but not T cell development and function" here at Nature Immunology. I had a look at your response and I think the remaining points (enumerated below) can be adequately addressed as proposed in your rebuttal. We are very interested in the possibility of publishing your study in Nature Immunology.

(1) measurement of CBL protein in gene-edited B cells - include the new experimental data, and would be good to scan the blot and report protein ratio to loading control band (CBL/GAPDH).

(2) more detailed explanation for in vitro activation of transitional B cells by T cells & IL-21 can induce Ig production (might include Supplementary Fig).

(3) Include Supplementary Figure looking at p-ERK activation downstream from CD38 in CBL-deficient B cells (knockdown using primary CD19+ B cells).

(4) Fine to include a limitations of the study to the Discussion section.

Please note that articles in Nature Immunology can have up to 8 main figures and 10 Extended Data Figures, hence you might wish to revise the Supplementary figures to Extended Data figures to increase their accessibility.

Also note, as a guideline, Articles allow up to 50 references in the main text. An additional 20 references can be included in the Online Methods. Only papers that have been published or accepted by a named publication or recognized preprint server should be in the numbered list.

We therefore invite you to revise your manuscript taking into account all reviewer and editor comments. Please highlight all changes in the manuscript text file in Microsoft Word format.

* If you have not done so already please begin to revise your manuscript so that it conforms to our Article format instructions at <http://www.nature.com/ni/authors/index.html>. Refer also to any guidelines provided in this letter.

* Please include a revised version of any required reporting checklist. It will be available to referees to aid in their evaluation of the manuscript goes back for peer review. They are available here:

Reporting summary:

When submitting the revised version of your manuscript, please pay close attention to our <https://www.nature.com/nature-portfolio/editorial-policies/image-integrity>>Digital Image Integrity Guidelines. and to the following points below:

Please note, Extended Data figures and tables are online-only (appearing in the online PDF and full-text HTML version of the paper), peer-reviewed display items that provide essential background to the Article but are not included in the printed version of the paper due to space constraints or being of interest only to a few specialists. A maximum of ten Extended Data display items (figures and tables) is typically permitted. When re-submitting your manuscript, please ensure that any supplementary figures and tables that are more critical to the manuscript's conclusions are converted to Extended data to increase these data's visibility.

Link Redacted

We hope to receive your revised manuscript within two weeks. If you cannot send it within this time, please let us know. We will be happy to consider your revision so long as nothing similar has been accepted for publication at Nature Immunology or published elsewhere.

Nature Immunology is committed to improving transparency in authorship. As part of our efforts in this direction, we are now requesting that all authors identified as 'corresponding author' on published papers create and link their Open Researcher and Contributor Identifier (ORCID) with their account on the Manuscript Tracking System (MTS), prior to acceptance. ORCID helps the scientific community achieve unambiguous attribution of all scholarly contributions. You can create and link your ORCID from the home page of the MTS by clicking on 'Modify my Springer Nature account'. For more information please visit www.springernature.com/orcid.

Kind regards,

Laurie

Laurie A. Dempsey, Ph.D.
Senior Editor
Nature Immunology
l.dempsey@us.nature.com
ORCID: 0000-0002-3304-796X

Referee expertise:

Referee #1:

Referee #2:

Referee #3:

Reviewers' Comments:

Reviewer #1 (Remarks to the Author):

the authors have adequately addressed my concerns.

Reviewer #3 (Remarks to the Author):

The authors have made substantial efforts to address the reviewers' concerns, providing new experimental data and clarifying several aspects of the study. The additional analyses on CBL editing efficiency and BCR repertoire are valuable and strengthen the manuscript. However, several key points remain insufficiently supported or require clarification.

While the newly added data in Figure S4, including the improved CBL amplification gel, are satisfactory, the western blot showing CBL protein expression in gene-edited CD34⁺ HSPCs remains unconvincing. The vinculin loading control for the CBL sgRNA (1+3) sample displays a markedly stronger signal than the AAVS1 sgRNA (1+2) control, while the CBL band intensity appears comparable. This pattern is consistent with reduced CBL protein levels in the edited cells, contrary to the authors' statement that CBL expression is unaffected ("Figure R2").

Furthermore, the authors' explanation regarding class switching and antibody secretion by transitional B cells does not resolve the issue. The data presented suggest comparable IgA and IgG secretion by transitional and naïve B cells under the

tested conditions. This is biologically unexpected. The earlier report (2006), which is cited as support, described class switching by transitional B cells after stimulation through CD40L + BCR engagement, or CD40L/BCR + IL-10, resulting in low-level Ig production compared with naïve B cells. However, the current study stimulated transitional cells with CD40L + IL-21, making the comparison inappropriate. Clarification is still required as to why transitional B cells in the present experiments show IgA and IgG secretion comparable to naïve B cells.

Finally, the REH cell experiments remain inappropriate for addressing the mechanistic question of impaired apoptosis in CBL-deficient B cells. REH cells are transformed pre-B cells lacking surface BCR expression and thus cannot model the immature B cells implicated in the patient phenotype. High CD38 expression alone does not justify their use as a surrogate, and signaling dynamics in this leukemic context cannot be extrapolated to normal immature B cells. Consequently, these data do not substantiate the proposed mechanism linking CBL deficiency to defective apoptosis and B cell accumulation.

Version 2:

Decision Letter:

Our ref: NI-A39326B

28th Oct 2025

Dear Jonathan,

Thank you for submitting your revised manuscript "Somatic deficiency of human CBL in leukocytes impairs B cell but not T cell development and function" (NI-A39326B). I have looked at your revised manuscript and the response to the previous round of external review. After discussion with my editorial colleagues, I am happy to announce that we'll be happy in principle to publish it in Nature Immunology, pending minor revisions to comply with our editorial and formatting guidelines.

We will now perform detailed checks on your paper and will send you a checklist detailing our editorial and formatting requirements in about a week. Please do not upload the final materials and make any revisions until you receive this additional information from us.

If you had not uploaded a Word file for the current version of the manuscript, we will need one before beginning the editing process; please email that to immunology@us.nature.com at your earliest convenience.

Thank you again for your interest in Nature Immunology Please do not hesitate to contact me if you have any questions.

Kind regards,

Laurie

Laurie A. Dempsey, Ph.D.
Senior Editor
Nature Immunology
l.dempsey@us.nature.com
ORCID: 0000-0002-3304-796X

We thank the Reviewers for the in-depth commentary and positive evaluation of our manuscript. We have performed extensive experiments to address the issues that were raised. Below we respond in detail to the suggestions.

Major additions of experiments and analyses include:

- BCR DNA sequencing of naïve, transitional and memory B cells from five CBL deficient individuals from three different kindreds demonstrating altered BCR repertoires and defects in somatic hypermutation

- Bulk RNA sequencing of CD10⁺CD20^{dim} B cell progenitors/precursors derived in vitro from CD34⁺ HSPCs, demonstrating inflammatory, pro-proliferative activation and overlapping pathway activation with PIK3CDGOF cells.

- Stimulation and Ig production assays of freshly isolated B cells from CBL-deficient patients that clarify that hypergammaglobulinemia is a B cell-extrinsically acquired phenotype

- Extensive validation of the results in the original version of the manuscript through further replicates, independent experiments, new analyses and statistical assessments.

Reviewer #1

(Remarks to the Author)

In their paper, Vatovec and colleagues provide a characterization of 10 patients homozygous for loss of function variants of CBL. They conclude that, unlike mice, CBL plays an important role in the development, tolerance and activation of B cells. In contrast, the defects in T cells are mild. This paper both provides important insights into CBL function and illustrates the limitations of mechanistic studies in mice.

The results presented in this paper represent a monumental, international effort. Indeed, there are many outstanding aspects to the primary human studies including the genetic studies and the extensive characterization of both bone marrow and peripheral blood samples. Indeed, it would be difficult to ask for additional experiments on primary human samples.

We thank Reviewer 1 for these positive comments and their appreciation of the data we have generated from these rare patients.

The primary human characterization is complemented by in vitro models of both B cell development and peripheral activation. While these studies provide important mechanistic insights, they could be improved. In particular, the effect of CBL LOF on relevant signaling pathways is under-explored. While they only assay ERK activation, they make frequent reference to a previous study of patients expressing a GOF p110delta mutant manifesting enhanced PI3K activation. However, they do not assay for PI3K activation in their systems. For example, ERK and PI3K play complementary but very distinct roles across the pre-BI and pre-BII cell stages (at least in mice). In their in vitro system of B cell development, they could sort these cell populations and do RNA-seq with paired-end sequencing. They could then examine ERK and PI3K signaling pathways relative to proliferative pathways and light chain recombination. Ideally, they would

have a GOF p110delta mutant control. However, I leave it to the authors on how they would like to assay for PI3K activation.

We appreciate the Reviewers' interest in a deeper mechanistic investigation of the observed phenotypes in isogenic models of CBL deficiency. To further investigate the impact of CBL deficiency on early B cell progenitors and precursors, we subjected control AAVS1 or CBL-edited CD34⁺ hematopoietic stem and progenitor cells (HSPCs) from healthy donor cord blood to undergo B cell development *in vitro*. In addition, we used an adenine base editor to knock-in a validated *PIK3CD* GOF allele, encoding the p110d subunit of PI3 kinase (p.C416R;(I)), which demonstrates enhanced activation of the PI3K pathway and thus served as a positive control (PIK3^{GOF}).

RNA-sequencing performed on CD10⁺CD20^{dim} B cell progenitors/precursors derived from CD34⁺ HSPCs showed significant activation of inflammatory pathways in CBL Ub^{LOF} edited cells, likely caused by secretion of proinflammatory cytokines by CBL Ub^{LOF} myeloid cells present in the co-culture system (Figure S4G-I). CBL Ub^{LOF} CD10⁺CD20^{dim} B cell progenitors/precursors showed increased G2-M checkpoint and mTORC1 signatures, which indicate altered cell proliferation and enhanced PI3K pathway activation.

Figure S4H-J: Bulk RNA-seq on AAVS1- and CBL-edited CD19⁺CD10⁺CD20^{low} CD34⁺ HSPC-derived B progenitor cells. (G) Gene set enrichment analysis with pathways for which significant enrichment (red) and depletion (blue) was detected. (H-I) Differential gene expression between control AAVS1- and CBL-edited B-lineage cells for leading edge genes in (H) Hallmark G2-M checkpoint and (I) Hallmark mTORC1 signaling gene sets. Colors reflect Z-transformed normalized read counts. NES: Normalized enrichment score.

Furthermore, we found that a significant number of shared genes was up- or downregulated in both Ub^{LOF} and PI3K^{GOF} CD34⁺ HSPC-derived B cell progenitors (Figures S4H). To determine whether the transcriptional changes in CBL Ub^{LOF} cells mirror those of PI3K^{GOF} cells, we generated PI3K^{GOF} gene sets by comparing AAVS1-edited and PI3K^{GOF} B cell samples. Gene set enrichment analysis revealed that genes upregulated or downregulated in PI3K^{GOF} cells were also significantly enriched or depleted, respectively, in CBL Ub^{LOF} cells (Figures S4K, Figure 4E).

Figures S4K and 4E: Transcriptional overlap between CBL-edited and PI3K^{GOF} CD34⁺-derived B cell progenitors. (S4K) Venn diagrams showing overlap between differentially expressed genes in PI3K^{GOF} and CBL-edited samples. Left: genes upregulated in both PI3K^{GOF} and CBL Ub^{LOF} samples as compared to AAVS1^{KO}. Right: genes downregulated in both comparisons. Overlap significance was calculated using a binomial test. (4E) Gene set enrichment analysis for PI3K^{GOF} gene signatures in CBL^{KO} samples. NES: normalized enrichment score.

Strikingly, more than half of the genes affected by CBL Ub^{LOF} were PI3K^{GOF} dependent, suggesting a substantial common axis of dysregulation due to these two genotypes. These findings indicate that CBL deficiency and PI3K^{GOF} mutations converge on common molecular pathways, supporting the hypothesis of enhanced PI3K signalling in the absence of CBL.

As the reviewer asked about effects in light chain rearrangement, we explored this in the RNA-sequencing data. Differential gene expression analysis of genes involved in V(D)J recombination showed reduced expression of *RAG1*, *LIG4* and *DNTT* in CBL Ub^{LOF} edited B-lineage cells compared to AAVS1 control CD34⁺ HSPCs (Figure R1A). We next inferred productive BCR repertoires from our bulk RNA-seq data using MiXCR (Figure R1B-C). CBL Ub^{LOF} B-lineage cells showed a reduction in clonal richness and Shannon diversity, whereas the Shannon equitability remained comparable to the AAVS1^{KO} B-lineage cells (Figure R1B-C).

Figure R1: Analysis of Ig light chain gene recombination in CBL-edited CD34⁺-derived B cell progenitors. (A) Volcano plot showing differential gene expression in CBL Ub^{LOF} vs. AAVS1-edited CD34⁺-derived B cell progenitors. Key genes involved in

V(D)J recombination are highlighted in red. Dashed line indicates significance cut-off of adjusted P value < 0.05. (B-C) Productive BCR repertoires inferred from bulk RNA-seq using MiXCR showing (B) Shannon diversity and (C) Shannon equitability of *IGK* and *IGL* chains.

These findings suggest that the reduced diversity of the BCR repertoire may reflect impaired or delayed VJ recombination, likely due to decreased *RAG* expression, rather than oligoclonal expansion. It is tempting to speculate that increased proliferative signals in CBL-deficient B cells might delay *RAG* activity, thereby affecting Ig light chain recombination. However, due to the paucity of primary bone marrow aspirates from the patients, we were unable to further test this hypothesis. While addressing this question in greater detail remains an important area for future investigation, we believe that it lies beyond the scope of the present study. As we can currently not confirm these data on primary samples from CBL-deficient patients, we would prefer to not include them in this publication and first explore this topic further to validate the findings. That being said, we include in the manuscript a whole new main figure on BCR sequencing data, showing clear defects in BCR rearrangement and somatic hypermutation (Figure 6, see response to other reviewers).

Furthermore, we have addressed the Reviewers concerns by monitoring ERK and PI3K pathways in our isogenic immature B cell line model REH, which is now included in the manuscript as Supplemental Figures S5D-I. We found that activation of the ERK pathway is upregulated at early (5-30 mins) and late (16 hours) timepoints in CBL Ub^{LOF} REH cells upon CD38 crosslinking compared to WT REH cells (Figure S5D,E). Conversely, the AKT pathway, downstream of PI3K exhibits increased activity only at later timepoints (Figure S5D-I). These data suggest that both the RAS-ERK and PI3K-AKT cascades are implicated by CBL-dependent CD38 signalling in immature B cells.

Suppl. Figure 6

Figure S6: (A-F) Western blots of wildtype and CBL Y371C knockin REH cells upon CD38 crosslinking with daratumumab for the indicated time periods (min/h). (B-C, E-F) Show quantifications of three biological replicates. Statistical significance was assessed with Mann Whitney tests. * $p < 0.05$.

Specific comments:

1. on page 11, they should show proliferation results from long-term cultures or just delete this statement

As suggested by Reviewer 1, we have removed this statement. However, we provide additional characterization of CD4⁺ T cell function (see below).

2. Did the authors stain for IgM and IgD, especially in the transitional stages? Was there an increase in a putative anergic B cell pool? This was a bit unclear in the manuscript.

We have performed flow cytometric analysis to assess expression levels of IgM and IgD on transitional and naïve B cells from 4 CBL-deficient patients and 10 healthy donors. As human B cells develop from a transitional into a naïve state, expression of surface IgM is significantly down-

regulated (2, 3). While we observed a reduction in IgM expression on CBL-deficient naïve B cells compared to CBL-deficient transitional B cells, the overall levels of IgM on CBL-deficient B cells remained 2-fold greater than on corresponding B cell subsets from healthy donors. Indeed, the level of IgM on CBL-deficient naïve B cells was comparable to levels on transitional B cells from healthy donors, indicating an arrest at the transitional g naïve stage of B cell development. In contrast to IgM, expression of IgD was reduced on transitional and naïve B cells from CBL-deficient patients compared to these cell subsets from healthy donors, being statistically significant for naïve B cells ($p=0.0027$, >2-fold difference; Figure S4B). Furthermore, while IgD increased as transitional B cells developed into naïve B cells in healthy donors, this upregulation was not observed for CBL-deficient B cells (Figure S4B). This data has been included in the revised manuscript as Figure S4B.

Figure S4B: Quantification of IgM and IgD surface expression on primary transitional and naïve B cells of healthy donors and the indicated CBL LOH patients. Statistical significance was assessed with Mann Whitney tests corrected for multiple testing.

These data are also consistent with data presented in Figure 3E, F and G of the original manuscript that showed aberrant expression of key surface receptors on CBL-deficient B cells ($n=5$ in original manuscript) that characterise human B cell development – i.e. sustained expression of CD5 and CD38, and reduced expression of CD19 and CD21 - indicating perturbed B cell development due to CBL-deficiency. We have been able to confirm this aberrant transitional B cell phenotype in an additional CBL-deficient patient, such that the data presented in Fig 3E in the revised manuscript is now derived from 6 patients.

Figure 3E: (E) CD5, CD9, CD21 and CD38 expression on transitional B cells of healthy donors (black) and CBL-LOH patients (red) as determined by flow-cytometry. The statistical significance of differences was assessed in multiple Mann-Whitney tests, with correction for multiple testing. * $p < 0.05$, ** $p < 0.005$, **** $p < 0.00005$.

Many anergic B cells in humans take the phenotype of peripheral blood naïve B cells that have an $IgD^+IgM^{lo/neg}$ phenotype (4, 5). As IgM expression was consistently ~2-fold higher and IgD 2-fold lower on CBL-deficient naïve B cells we conclude that there was not an increase in the putative anergic B cell pool in CBL-deficient patients. These data will be included in a revised manuscript.

3. They showed representative ERK blots but they need to provide quantitation across at least three experiments. This needs to be done for both 4I and 5E. It would be particularly useful for interpreting late time points in 5E.

We have quantified the respective replicates of these experiments and provide quantifications for stimulation of REH cells in Figures S6A-F, which as explained above, now also includes longer timepoints of Daratumumab crosslinking. Additionally, we have included quantifications of BJAB stimulations with anti-IgG/IgM in Figure S7D:

Figure S7D: (D) Quantification of three biological replicates of the western blot shown in Figure 5E. Statistical significance was assessed with Mann Whitney tests. * $p < 0.05$.

These data indicate that the response to the stimulation eventually reaches a plateau and the difference between WT and knock-in cells is only present at early timepoints.

4. The referencing is incomplete, XXXs in the methods, PMIDs in the discussion. The references are in a different font.

We have corrected these editorial issues wherever we could identify them. Thank you.

Reviewer #2

(Remarks to the Author)

In this paper, Vatovec and Bohlen, et al. reported their findings of defects in B cell development and function in a cohort of CBL LOH variants patients. Their data has revealed that the hematopoietic CBL mutation does not influence T cell development and function. Instead, it impairs the transitional, immature B cell development into mature and memory B cells. The differentiation of B cells to antibody secreting cells is also impaired in the absence of CBL. Additionally, mechanistic studies demonstrate that the mutation leads to an upregulation of CD38 expression on the mutant immature B cells and impaired CD38 induced apoptosis. Moreover, they have shown that these patients possess high titers of autoantibodies and autoreactive BCR-expressing B cells, indicating that the immune tolerance of B cells is broken.

This paper has reported an interesting and novel finding from a cohort of very interesting, rare immune disorder patients. Their data not only reveals an important, differential role of CBL in human B and T cell development and function but also presents another example that the same gene may play distinct roles in human and mice. However, I also feel that some hypotheses, experimental designs, and interpretations of the results are not very compelling and sometimes contradictory. In addition, there is no in depth mechanistical studies showing how the CBL deficiency affects B cell maturation and tolerance. Moreover, statistics in some figures are missing; however, this is perhaps due to the limited access to human specimens. The following major concerns need to be addressed, which may help to improve the overall quality of the paper:

1) Authors described that CBL LOH patients had histories of repeated severe infections. However, Fig. 2A-B shows similar levels of serum Abs against various microbes in homozygosity CBL LOF patients and age-matched pediatric donors. Does this suggest that patients' long term antibody responses are normal, and the unusual severe infections are not caused by the alteration in patient's B cell compartments? If so, this observation seems to be contradictory to the claim that B cell function is impaired in these patients.

We thank the Reviewer for raising this point, as similar comments were raised by Reviewer #3. It is potentially difficult to reconcile apparent intact levels of pathogen and vaccine specific IgG in the CBL-deficient patients with defects in B cell development and differentiation as observed both *ex vivo* using patients' primary B cells and *in vitro* using a model of HSPC / B cell development.

The challenge with relying on serological data to inform the magnitude and longevity of adaptive humoral immune responses is that the data is often a snapshot of a single time, with incomplete knowledge of history of antigen exposure. To address this more rigorously, we have further investigated SARS-CoV2-specific B cell responses in 4 CBL-deficient patients, with longitudinal samples being available for 3-5 patients for the different time points. We have extended our original finding to additional patients and time points, and confirmed that the frequencies of SARS-CoV2-specific B cells were reduced in CBL-deficient patients not only 6-12 mo post antigen encounter, but also after 2 years. In fact, while the proportion of Spike-binding B cells in healthy donors increased ~3-fold during the first 12 months following vaccination, these cells remained consistently low and unchanged at all time points tested in CBL-deficient patients. Importantly, the proportion of Spike-binding B cells that had undergone class switching to IgG in CBL-deficient patients was consistently and statistically less than Spike-specific B cells from healthy donors for all time points assessed.

Figure 5G+H: (I) Frequency of spike+ B cells in patients and healthy donors, as determined by flow cytometry with tetramer staining. Statistical significance was assessed by unpaired, two-sided t tests, * $p < 0.05$. (J) Frequency of IgG+, IgA+ and IgG-IgA- B cells among spike+ B cells in patients ($n = 3$) and healthy donors ($n = 6$ at 4-10wk and $n = 9$ at 6-12 mo). Error bars indicate standard error of the mean. Statistical significance was assessed by unpaired, two-sided t tests, * $p < 0.05$.

In contrast to these findings for Spike-binding B cells in peripheral blood, levels of anti-Spike specific IgG were detectable in CBL-deficient patients and were similar to or exceeded levels detected in healthy donors. Whilst this appears paradoxical in the setting of reductions in proportions of Spike-binding B cells, these data are consistent with the general hypergammaglobulinemia reported for CBL-deficient patients. Overall, these findings identify a non-redundant role for CBL in generating, expanding and maintaining Ag-specific memory B

cells. In contrast, the role of CBL in the generation, persistence and function of Ab-secreting plasma cells will require additional investigation.

An extension of this is that the presence of detectable levels of antigen-specific IgG does not necessarily mean that these antibodies will be adequately protective. Indeed, our new data derived from analysis of the BCR repertoires of transitional, naïve and memory B cells have revealed significantly reduced levels of somatic hypermutation in CBL-deficient IgM⁺ memory B cells, indicative of qualitative differences in the nature of the Ab expressed by memory B cells (and presumably secreted by plasma cells) from CBL-deficient patients and healthy donors (Figure 6). The data derived from determining humoral immune responses to SARS-CoV2 infection/vaccination in CBL-deficiency provide significant insight into the functionality of antigen-specific Ig generated in these patients.

2) Authors concluded that CBL LOH variant T cells retained normal functions, based on T cell proliferation upon TCR/CD28 stimulation, normal STAT5 phosphorylation after IL2 and IL27 stimulation, and INF-gamma and TNF production (Fig. 2 and related sFigs). However, antibody responses to microbe infections depend on Tfh cells. To claim that the LOH mutation does not affect T cell function in the context of pathogen infections, they should at least show that the differentiation and function of CBL LOH Tfh cells were not altered, given that CBL and CBLB proteins have been shown recently to negatively regulate the development and function of Tfh cells (DOI: 10.1016/j.immuni.2024.04.023)

We completely agree that it is important to assess the generation and function of Tfh-type cells in CBL-deficiency. We have now performed such experiments. First, we found that the proportions of CD4⁺CD45RA⁻CXCR5⁺ circulating Tfh-type (cTfh) cells in peripheral blood of CBL-deficient patients and healthy donors were comparable. Second, cTfh cells were sorted from PBMCs of 10 healthy donors and 4 CBL-deficient patients and then cultured with anti-CD2/CD3/CD28 mAbs for 5 days. Production of the canonical Tfh cytokine IL-21 was intact for CBL deficiency. Production of other B cell stimulatory cytokines – IL-10, IL-13 – by CBL-deficient cTfh cells was moderately, but not significantly reduced compared to healthy donors and stayed within the range of healthy donors. Production of other cytokines, such as IL-2, IL-4, IL-9, IL-22, IL-17A and TNF, was largely intact, and IFN-g slightly increased, for CBL-deficient cTfh cells. As both IL-10 and IL-13 can induce class switching to IgG in human B cells (6), it is possible that reduced production of these cytokines may represent a B cell extrinsic mechanism to explain impaired humoral immunity in CBL-deficiency. However, we think this is unlikely for at least two reasons. First, even though humans with inborn errors in IL-10/IL-10R signalling develop very early onset IBD, levels of total and Ag-specific Ig in these patients are unaffected (Glocker et al 2011, *Ann NY Acad Sci*). Similarly, the recent report of individuals with IBD secondary to anti-IL-10 neutralising autoAbs did not mention any differences in patient serology in the context of serum Ig levels (Griffin et al 2024, *N Engl J Med*). Second, while treatment of individuals with atopic dermatitis with Dupilimab – which blocks the function of IL-4 and IL-13 – effectively reduces levels of serum IgE, it has no effect on Ag-specific Ab responses (Blauvelt et al 2019, *J Am Acad Dermatol*). Thus, while there were moderate defects in production of some cytokines by CBL-deficient cTfh cells, these are unlikely to explain the altered ability of CBL-deficient patients to mount and maintain effective long-lived humoral immune responses. These data regarding cytokine production by cTfh cells have been included in the revised version of the manuscript and appear in Figure S3D-E.

Figure S3D, E: Intracellular and extracellular cytokine production by patients T_h cells stimulated with CD2/CD3/CD28 under T_h0 conditions for 5 days. (D) Intracellular staining of the indicated cytokines. Mann-Whitney testing with correction for multiple testing did not reveal significant ($p < 0.05$) differences between healthy donors and CBL-LOH patients. (E) Extracellular detection of cytokines by ELISA. Mann-Whitney testing with correction for multiple testing did not reveal significant ($p < 0.05$) differences between healthy donors and CBL-LOH patients.

3) Authors showed that transitional, immature, and memory B cells in young LOH patients were expanded and patients were hypergammaglobulinemia (Fig. 3A-C). However, plasmablasts were not increased relative to healthy controls. If the total number of plasmablasts, perhaps also PCs, are not increased, how can the excess amounts of antibodies be produced in these patients? Fig. 3D only shows the proportions of each cell subsets. How about absolute numbers?

The discrepancy noted by Reviewer 2 (and Reviewer 3) relating to hypergammaglobulinemia *in vivo* and intrinsically impaired B cell function *in vitro* is certainly confounding. And we thank the Reviewers for raising this. We hypothesize that this is specifically due to chronic inflammation. This is based on our recent findings published in the *J Clin Invest* which initially reported this cohort of CBL deficient individuals (7). Specifically, these patients have a transcriptional and cellular signature of chronic inflammation, and we propose that this inflammatory state underlies the hypergammaglobulinemia.

To assess whether this pro-inflammatory environment in CBL deficient patients affects B cells, we compared mRNA expression profiles of total B cells isolated from PBMCs of CBL patients P1-3 to healthy children of similar age to that of corresponding B cells that were then cultured for 24 hours following isolation. The analysis revealed a strong and significant upregulation of inflammatory pathways in directly isolated B cells, but not in B cells that were cultured for 24 hours. These data indicate that B cells of CBL-deficient patients acquired an inflammatory gene expression signature *in vivo* that was dependent on the *in vivo* inflammatory milieu and lost following *in vitro* culture. Thus, this feature is B cell extrinsic.

Figure S7E: Pathway enrichment analysis of bulk RNA-sequencing of healthy donor and CBL LOH patients' primary naïve B cells directly after sorting from cryopreserved PBMCs or after 24 hours of non-stimulated culture. NES: normalized enrichment ratio. p values were adjusted for multiple testing.

Next, we isolated total B cells from freshly collected blood samples, cultured them for 24 hours and then measured Ig production. We found significant increases in levels of secreted IgG1, IgG3, and IgM by the patients B cells (other Igs were not affected) compared to corresponding B cells from healthy donors. This is in contrast to the Ig secretion data in the manuscript (Figure 5A), which had been measured from B cells isolated from cryopreserved PBMCs of the patients:

Figure S7F: Immunoglobulin production by sorted primary B cell subsets of healthy donors and CBL-LOH patients from fresh blood samples without stimulation. Supernatants were collected after 24 hours and Igs were measured by ELISA. Statistical significance was assessed using Mann Whitney tests and correction for multiple testing. * $p < 0.05$

Finally, we speculated that the pro-inflammatory nature of the patients' B cells could be transferred to B cells from healthy donors by exposure to the inflammatory milieu mediated by CBL-deficient monocytes (7). We therefore isolated monocytes from fresh PBMCs of CBL-deficient patients and healthy donors, cultured them in RPMI with 10% FCS for 24 hours and collected the supernatants.

We then cultured total B cells from healthy donors in these supernatants for another 24 hours and measured Ig production.

Figure 5E: Primary monocytes of healthy donors and CBL LOH patients were sorted from fresh blood samples. After 24 hours of non-stimulated culture, supernatants were collected and B cells from healthy donors were stimulated with these supernatants for 24 hours. Ig production was assessed by ELISA. Statistical significance was assessed with multiple Mann Whitney tests adjusted for multiple testing. * $p < 0.05$.

Under these conditions, supernatants harvested from co-cultures and CBL-deficient monocytes induced substantially more IgG1 and IgM production from healthy donor total B cells compared to supernatants prepared from co-cultures of healthy donor monocytes and B cells. In summary, our data indicate that chronic inflammation in these patients *in vivo* extrinsically boosts Ig production and may cause hypergammaglobulinemia.

This notion of an inflammatory milieu promoting polyclonal IgG production *in vivo*, resulting in hypergammaglobulinemia, is supported by several other published reports; First, inflammation due to ongoing mycobacterial infection in individuals with inborn errors of IFN- γ signalling (i.e. loss of function mutations in *IL12RB1*, *IFNGR1*, *STAT1*) is associated with increased levels of serum IgG (up to 5-fold greater than the upper limit of the reference range for healthy donors; see CS Ma et al 2015 (8)). Second, transgenic expression of IL-6 is sufficient to drive hypergammaglobulinemia in mouse models (9). This is consistent with IL-6 being one of the major cytokines overproduced by CBL Ub^{LOF} patients' monocytes (7) and being a key survival factor for plasma cells.

4) The HSPC culture experiments (Fig. 4A-C) showed only a proportional increase of immature B cells in CBL mutant cell cultures relative to controls, similar to that found in the BM of CBL LOH patients. The authors should show the absolute numbers of each cell subsets because the difference could be caused by a reduction in the total number of PreB I cells.

The volume of sample collected from each bone marrow aspirate can differ from individual to individual, as well as from the anatomical site of collection. Furthermore, the absolute volume of bone marrow (average 25 mL/kg in an adult) can fluctuate with age, infection and medications (Hassan et al, *J Royal Soc Med*, 2004). Thus, it is very difficult – if not impossible - to provide absolute numbers of B cell subsets present in human BM, especially when only a small amount of fluid is collected. For this reason, we report proportions of B cell subsets within BM samples. In addition, we do not have additional bone marrow aspirates from the patients to provide *in vivo* cell counts.

We have refrained from providing absolute cell counts for each B cell subset in our *in vitro* differentiation system, as the CD34⁺ cells are co-cultured on MS5 stroma cells, which makes it difficult to obtain accurate absolute counts of developing B cell subsets. Importantly, we have shown that cells from other leukocyte lineages (including dendritic cells and other myeloid cells) are concomitantly generated in these cultures, further limiting the ability to accurately determine absolute numbers of B cell subsets.

5) As having been mentioned by authors that CD38 signaling is one of the triggers to induce apoptosis of immature B cells. Since CD38 is upregulated in CBL LOH immature B cells, they hypothesize that the CBL LOH mutation attenuates CD38 induced apoptosis of immature B cells, consequently impairing the immune tolerance of B cells. To test this hypothesis, they first generated CBL LOH HSPCs-derived B cells and showed that mutant B cells were less susceptible to CD38 induced apoptosis as compared to WT controls. This finding indicates that the CBL LOH mutation attenuates CD38 signaling in B cells. However, when they examined CD38 signaling in Y371C mutant REH cell lines, it was found that ERK signaling was elevated in WT cells and significantly enhanced in the mutant cells after CD38 stimulation (Fig. 4I). This observation seems to raise a paradoxical role of CD38-ERK axis in B cell tolerance, because in WT cells activation of CD38-ERK signaling is interpreted as apoptotic whereas in CBL mutant cells enhanced CD38-ERK signaling is anti-apoptotic. In this regard, they should look at right signaling pathways, rather than ERK, for the signaling insights that are responsible for CBL mutant B cell apoptosis and immune tolerance. Does CD38 activate PIK3CD? Since *PIK3CD* is involved in B cell apoptosis and PIK3CD GOF and CBL LOH mutations exert a similar effect on B cell development, it is surprising to me why they did not directly compare PIK3CD activities? Additionally, since BCR-induced apoptosis of immature B cells is the main mechanism to delete autoreactive B cells, it will make more sense to look at BCR-induced apoptosis and signaling of immature B cells in this model.

We think that the Reviewer may have misinterpreted the result section “Impaired apoptosis in CBL-deficient immature B cells”. The ERK pathway provides a pro-proliferative, pro-survival signal (10, 11). We suggest that elevated ERK signalling in CBL mutant REH cells upon CD38 stimulation leads to a preferential ‘life’ instead of ‘death’ decision by the cells. Indeed, ERK pathway activation is very subtle in the control WT cells. This all is consistent with the role of CBL as a repressor of RAS-ERK signalling.

As mentioned above in response to Reviewer 1’s comments, we have deepened the mechanistic insight into this process with additional experiments assessing PI3K pathway activation:

--- Start of copied response ---

We appreciate the Reviewers’ interest in a deeper mechanistic investigation of the observed phenotypes in isogenic models of CBL deficiency. To further investigate the impact of CBL deficiency on early B cell progenitors and precursors, we subjected control AAVS1 or CBL-edited CD34⁺ hematopoietic stem and progenitor cells (HSPCs) from healthy donor cord blood to undergo B cell development *in vitro*. In addition, we used an adenine base editor to knock-in a validated *PIK3CD* GOF allele, encoding the p110d subunit of PI3 kinase (p.C416R;(1)), which demonstrates enhanced activation of the PI3K pathway and thus served as a positive control (PIK3^{GOF}).

RNA-sequencing performed on CD10⁺CD20^{dim} B cell progenitors/precursors derived from CD34⁺ HSPCs showed significant activation of inflammatory pathways in CBL Ub^{LOF} edited cells, likely caused by secretion of proinflammatory cytokines by CBL Ub^{LOF} myeloid cells present in the co-culture system (Figure S4G-I). CBL Ub^{LOF} CD10⁺CD20^{dim} B cell progenitors/precursors showed increased G2-M checkpoint and mTORC1 signatures, which indicate altered cell proliferation and enhanced PI3K pathway activation.

Figure S4H-J: Bulk RNA-seq on AAVS1- and CBL-edited CD19⁺CD10⁺CD20^{low} CD34⁺ HSPC-derived B progenitor cells. (G) Gene set enrichment analysis with pathways for which significant enrichment (red) and depletion (blue) was detected. (H-I) Differential gene expression between control AAVS1- and CBL-edited B-lineage cells for leading edge genes in (H) Hallmark G2-M checkpoint and (I) Hallmark mTORC1 signaling gene sets. Colors reflect Z-transformed normalized read counts. NES: Normalized enrichment score.

Furthermore, we found that a significant number of shared genes was up- or downregulated in both Ub^{LOF} and PI3K^{GOF} CD34⁺ HSPC-derived B cell progenitors (Figures S4H). To determine whether the transcriptional changes in CBL Ub^{LOF} cells mirror those of PI3K^{GOF} cells, we generated PI3K^{GOF} gene sets by comparing AAVS1-edited and PI3K^{GOF} B cell samples. Gene set enrichment analysis revealed that genes upregulated or downregulated in PI3K^{GOF} cells were also significantly enriched or depleted, respectively, in CBL Ub^{LOF} cells (Figures S4K, Figure 4E).

Figures S4K and 4E: Transcriptional overlap between CBL-edited and PI3K^{GOF} CD34⁺-derived B cell progenitors. (S4K) Venn diagrams showing overlap between differentially expressed genes in PI3K^{GOF} and CBL-edited samples. Left: genes upregulated in

both PI3K^{GOF} and CBL Ub^{LOF} samples as compared to AAVS1^{KO}. Right: genes downregulated in both comparisons. Overlap significance was calculated using a binomial test. (4E) Gene set enrichment analysis for PI3K^{GOF} gene signatures in CBL^{KO} samples. NES: normalized enrichment score.

Strikingly, more than half of the genes affected by CBL Ub^{LOF} were PI3K^{GOF} dependent, suggesting a substantial common axis of dysregulation due to these two genotypes. These findings indicate that CBL deficiency and PI3K^{GOF} mutations converge on common molecular pathways, supporting the hypothesis of enhanced PI3K signalling in the absence of CBL.

As the reviewer asked about effects in light chain rearrangement, we explored this in the RNA-sequencing data. Differential gene expression analysis of genes involved in V(D)J recombination showed reduced expression of *RAG1*, *LIG4* and *DNTT* in CBL Ub^{LOF} edited B-lineage cells compared to AAVS1 control CD34⁺ HSPCs (Figure R1A). We next inferred productive BCR repertoires from our bulk RNA-seq data using MiXCR (Figure R1B-C). CBL Ub^{LOF} B-lineage cells showed a reduction in clonal richness and Shannon diversity, whereas the Shannon equitability remained comparable to the AAVS1^{KO} B-lineage cells (Figure R1B-C).

Figure R1: Analysis of Ig light chain gene recombination in CBL-edited CD34⁺-derived B cell progenitors. (A) Volcano plot showing differential gene expression in CBL Ub^{LOF} vs. AAVS1-edited CD34⁺-derived B cell progenitors. Key genes involved in V(D)J recombination are highlighted in red. Dashed line indicates significance cut-off of adjusted P value < 0.05. (B-C) Productive BCR repertoires inferred from bulk RNA-seq using MiXCR showing (B) Shannon diversity and (C) Shannon equitability of *IGK* and *IGL* chains.

These findings suggest that the reduced diversity of the BCR repertoire may reflect impaired or delayed VJ recombination, likely due to decreased *RAG* expression, rather than oligoclonal expansion. It is tempting to speculate that increased proliferative signals in CBL-deficient B cells might delay *RAG* activity, thereby affecting Ig light chain recombination. However, due to the paucity of primary bone marrow aspirates from the patients, we were unable to further test this hypothesis. While addressing this question in greater detail remains an important area for future investigation, we believe that it lies beyond the scope of the present study. As we can currently not confirm these data on primary samples from CBL-deficient patients, we would prefer to not include them in this publication and first explore this topic further to validate the findings. That being said, we include in the manuscript a whole new main figure on BCR sequencing data,

showing clear defects in BCR rearrangement and somatic hypermutation (Figure 6, see response to other reviewers).

Furthermore, we have addressed the Reviewers concerns by monitoring ERK and PI3K pathways in our isogenic immature B cell line model REH, which is now included in the manuscript as Supplemental Figures S5D-I. We found that activation of the ERK pathway is upregulated at early (5-30 mins) and late (16 hours) timepoints in CBL Ub^{LOF} REH cells upon CD38 crosslinking compared to WT REH cells (Figure S5D,E). Conversely, the AKT pathway, downstream of PI3K exhibits increased activity only at later timepoints (Figure S5D-I). These data suggest that both the RAS-ERK and PI3K-AKT cascades are implicated by CBL-dependent CD38 signalling in immature B cells.

Suppl. Figure 6

Figure S6: (A-F) Western blots of wildtype and CBL Y371C knockin REH cells upon CD38 crosslinking with daratumumab for the indicated time periods (min/h). (B-C, E-F) Show quantifications of three biological replicates. Statistical significance was assessed with Mann Whitney tests. * $p < 0.05$.

--- End of copied response ---

In addition to the above listed experiments, we observed no significant difference in the apoptosis of BJAB cells upon BCR crosslinking:

Figure S5I: CBL Y371C knock-in BJAB cells are not resistant to BCR-induced apoptosis.

6) To understand the influence of extrinsic factors that might influence B cell development, authors examined soluble factors in patient's serum samples and HSPC cultures. They found that sCD40L, but not APRIL and BAFF, was significantly upregulated as compared to controls (sFig 5A-B) (no statistics was provided). Is the increased sCD40L secreted by activated T cells? If so, it will suggest that the function of CBL LOH T cells is altered. Additionally, is it possible that the increased CD40L causes the expansion of CBL LOH immature B cells, since CD40L may stimulate B cell proliferation? It is also noted that authors have examined Ig secretion by WT and mutant HSPC-derived B cells after CD40L stimulation (Fig. 5A), however, B cell proliferation is not mentioned.

While we observed that levels of sCD40L are elevated in patients' plasma and PBMC supernatants, we have not currently investigated the source of this cytokine as we believe it to be unlikely to underly B cell dysregulation present in the patients. It is more likely that increased sCD40L contributes to the chronic autoinflammation present in CBL-deficient patients due to its action on myeloid cells (monocytes, macrophages). This is reminiscent of inflammation, autoAbs and autoimmune conditions in:

- individuals with *CD40LG* duplication (12)
- *CD40lg* transgenic mice (13)

To investigate this further, we induced B cell development from healthy donor-derived CD34⁺ HSPCs in the presence or absence of moderate (5ng/mL) or high (100 ng/mL) concentrations of sCD40L (see Figure S5C-F). The generation of B-lineage progenitor cells from CD34⁺ HSPCs ($n = 2$) was not impacted irrespective of the presence or absence of two doses of sCD40L (Figure S5C-D). Similarly, progression of CD19⁺ progenitors through the early stages of B cell development was unaffected by exogenous sCD40L (Figure S5E-F), suggesting that the increased levels of sCD40L are unlikely to contribute to the perturbed trajectory of B cell development observed directly *ex vivo* in patient BM, or in *in vitro* models of B cell development from CBL-deficient CD34⁺ HSPCs.

Figure S5C-F: Impact of sCD40L levels on B cell differentiation *in vitro* using CD34⁺ HSPCs (C-D) B cell output at day 21 of co-culture of CD34⁺ HSPCs from two healthy donors. The MS5 co-culture was supplemented with IL-7 (20 ng/mL) and the indicated doses of sCD40L. Mean \pm s.d. Dots represent technical replicates. **(E-F)** Quantification of B cell subsets based on CD10 and CD20 marker expression. Cells were treated as in **(C-D)**. Mean \pm s.d.

7) Authors have shown that mutant B cells are defective in antibody secretion *in vitro* after CD40L stimulation (Fig. 5A). However, there is no experiment showing the development of plasma cells. This is an important question to clarify because if the development of CBL LOH B cells to plasma cell is deficient, why patients are hypergammaglobulinemia (Fig. 3B)?

We agree with Reviewer 2 that it would be important to assess plasmablast generation *in vitro*. While measuring Ig production is certainly a readout of B-cell differentiation into Ab-secreting cells, it does not quantify this at the cellular level. To extend our findings, we quantified the differentiation of transitional and naive B cells into plasmablast-type cells (ie CD38^{hi}CD27^{hi}CD20^{lo} phenotype) *in vitro* in response to stimulation with CD40L/IL-21.

Figure R2: Formation of plasmablast cells *in vitro* by healthy donor and P8 primary B cells.

Interestingly, the proportions of plasmablast cells generated from CBL-deficient B cells was comparable to those derived from CBL-sufficient B cells (Figure R2). While this seems to contrast reduced Ig production *in vitro* by CBL-deficient B cells, these data resemble our earlier study which assessed *in vitro* differentiation of transitional and naïve B cells isolated from individuals harboring *PIK3CD* GOF variants (14). Here, *PIK3CD* GOF B cells exhibited comparable differentiation into plasmablast-type cells *in vitro* but produced significantly lower amounts of Ig compared to these B cell subsets from healthy donors. This was associated with poor induction of a plasmablast-specific transcriptional signature in *PIK3CD* GOF B cells. These plasmablast differentiation experiments are carried out by the Australian team, and we have so far only been able to obtain data from one patient, we would therefore prefer to refrain from including these data in the current manuscript but are willing to include them upon request from the reviewers or editor.

Regarding the comment toward hypergammaglobulinemia, we refer to our response above (Reviewer 2, Query 2):

--- Start of copied response ---

The discrepancy noted by Reviewer 2 (and Reviewer 3) relating to hypergammaglobulinemia *in vivo* and intrinsically impaired B cell function *in vitro* is certainly confounding. And we thank the Reviewers for raising this. We hypothesize that this is specifically due to chronic inflammation. This is based on our recent findings published in the *J Clin Invest* which initially reported this cohort of CBL deficient individuals (7). Specifically, these patients have a transcriptional and cellular signature of chronic inflammation, and we propose that this inflammatory state underlies the hypergammaglobulinemia.

To assess whether this pro-inflammatory environment in CBL deficient patients affects B cells, we compared mRNA expression profiles of total B cells isolated from PBMCs of CBL patients P1-3 to healthy children of similar age to that of corresponding B cells that were then cultured for 24 hours following isolation. The analysis revealed a strong and significant upregulation of inflammatory pathways in directly isolated B cells, but not in B cells that were cultured for 24 hours. These data indicate that B cells of CBL-deficient patients acquired an inflammatory gene expression signature *in vivo* that was dependent on the *in vivo* inflammatory milieu and lost following *in vitro* culture. Thus, this feature is B cell extrinsic.

Figure S7E: Pathway enrichment analysis of bulk RNA-sequencing of healthy donor and CBL LOH patients' primary naïve B cells directly after sorting from cryopreserved PBMCs or after 24 hours of non-stimulated culture. NES: normalized enrichment ratio. p values were adjusted for multiple testing.

Next, we isolated total B cells from freshly collected blood samples, cultured them for 24 hours and then measured Ig production. We found significant increases in levels of secreted IgG1, IgG3, and IgM by the patients B cells (other Igs were not affected) compared to corresponding B cells from healthy donors. This is in contrast to the Ig secretion data in the manuscript (Figure 5A), which had been measured from B cells isolated from cryopreserved PBMCs of the patients:

Figure S7F: Immunoglobulin production by sorted primary B cell subsets of healthy donors and CBL-LOH patients from fresh blood samples without stimulation. Supernatants were collected after 24 hours and Igs were measured by ELISA. Statistical significance was assessed using Mann Whitney tests and correction for multiple testing. * $p < 0.05$

Finally, we speculated that the pro-inflammatory nature of the patients' B cells could be transferred to B cells from healthy donors by exposure to the inflammatory milieu mediated by CBL-deficient monocytes (7). We therefore isolated monocytes from fresh PBMCs of CBL-deficient patients and healthy donors, cultured them in RPMI with 10% FCS for 24 hours and collected the supernatants.

We then cultured total B cells from healthy donors in these supernatants for another 24 hours and measured Ig production.

Figure 5E: Primary monocytes of healthy donors and CBL LOH patients were sorted from fresh blood samples. After 24 hours of non-stimulated culture, supernatants were collected and B cells from healthy donors were stimulated with these supernatants for 24 hours. Ig production was assessed by ELISA. Statistical significance was assessed with multiple Mann Whitney tests adjusted for multiple testing. * $p < 0.05$.

Under these conditions, supernatants harvested from co-cultures and CBL-deficient monocytes induced substantially more IgG1 and IgM production from healthy donor total B cells compared to supernatants prepared from co-cultures of healthy donor monocytes and B cells. In summary, our data indicate that chronic inflammation in these patients *in vivo* extrinsically boosts Ig production and may cause hypergammaglobulinemia.

This notion of an inflammatory milieu promoting polyclonal IgG production *in vivo*, resulting in hypergammaglobulinemia, is supported by several other published reports; First, inflammation due to ongoing mycobacterial infection in individuals with inborn errors of IFN- γ signalling (i.e. loss of function mutations in *IL12RB1*, *IFNGR1*, *STAT1*) is associated with increased levels of serum IgG (up to 5-fold greater than the upper limit of the reference range for healthy donors; see CS Ma et al 2015 (8)). Second, transgenic expression of IL-6 is sufficient to drive hypergammaglobulinemia in mouse models (9). This is consistent with IL-6 being one of the major cytokines overproduced by CBL Ub^{LOF} patients' monocytes (7) and being a key survival factor for plasma cells.

--- End of copied response ---

8) CBL LOH patients exhibited a breakdown of B cell immune tolerance, as they produced high titers of autoantibodies against multiple autoantigens. Increased autoreactive B cells were verified by flowcytometry using V9G4 anti-idiotypic antibodies, as V9G4hi B cells were increased in LOH patients relative to healthy controls. Since the mutant B cells could have acquired an increased capability to bind non-specifically to the V9G4 antibody, it would be better to provide VH gene DNA sequence data to support their claim.

We have now performed BCR repertoire analysis on transitional, naïve and memory B cells isolated from 5 CBL-deficient patients from 3 different kindreds. This revealed clear differences between the CBL-deficient patients and healthy donors (Figure 6). First, there was an enriched usage of the Ig *VH4-34* gene element in transitional, naïve and IgM⁺ memory B cells from the patient's compared to healthy donors. This data confirms our observations at the cellular level of:

1. increased binding of the anti-idiotypic 9G4 mAb to CBL-deficient B cells,

- increased reactivity of serum from CBL-deficient patients to B cells from healthy donors, as revealed by reactivity with the 9G4 mAb, and
- detection of autoAbs against a wide range of self-Ags in serum from the patients.

Figure 6

Figure 6: B cell receptor repertoire of CBL-deficient patients reveals a defect in Ig V gene usage and somatic hypermutation. (A, B) Usage of top 3 (A) IGHV VH4-34, VH4-59 and VH3-23 gene elements and (B) Ig JH4 and JH6 elements in transitional, naïve and memory B cells isolated from healthy donors and the indicated CBL-deficient patients. (C, D) CDR3 lengths in (C) transitional, naïve and memory cells isolated from healthy donors and CBL-deficient patients, or (D) CD10⁺CD20^{dim} HSPC-derived B cell progenitors edited at the AAVS1 or CBL loci. (E) Frequency of Ig somatic hypermutations in memory B cells defined by expression of distinct class switched Ig isotypes. (F) Frequency of clones with different levels of SHM within IgM⁺ memory B cells. (G) CDR replacement:silent ratios in IgM, IgG and IgA memory B cells. Statistical significance was assessed by Wilcoxon with Bonferroni correction for multiple testing (if needed) (B, C, D, G) or Dunn's test for multiple comparisons (E, F), *p < 0.05.

Second, we detected reduced usage of *IGHJ6* genes and increased usage of *IGHJ4* genes by CBL-deficient transitional and naïve B cells relative to healthy donors. Interestingly, CDR3 lengths of Ig expressed by transitional and naïve B cells were significantly shorter for CBL-deficient patients compared to healthy donors. These shorter CDR3 lengths likely results from decreased usage of *IGHJ6* as this gene element contributes the highest number of amino acids to Ig CDR3 regions(15). Lastly, we confirmed that while the shift away from *IGHJ6* usage typically occurs as healthy donor naïve B cells differentiate in memory B cells, this occurred at a much earlier stage of B-cell development in CBL-deficient patients. Thus, these differences in the molecular architecture of Ig genes expressed by CBL-deficient B cells likely contribute to autoreactivity exhibited by these patients and their B cells.

Our analysis of BCR repertoires also revealed significantly reduced levels of somatic hypermutation in CBL-deficient IgM⁺ memory B cells, which suggest impaired selection and/or affinity maturation of memory B cells from CBL-deficient patients relative to healthy donors. This would result in qualitative differences between CBL-deficient and CBL-sufficient memory B cells. These data detailing the features of the BCR repertoire of CBL-deficient B cell subsets relative to that of B cells from healthy donors, as well as levels of somatic hypermutation, are included in the revised manuscript as Figure 6 and the following results section:

“BCR repertoire reveals numerous perturbations in IG gene usage and somatic hypermutation due to CBL deficiency

To further extend our understanding of the impact of CBL-deficiency on B cell development and differentiation, we performed BCR repertoire analysis on transitional, naïve and memory B cells isolated from 5 CBL-deficient patients from 3 different kindreds. This revealed unequivocal differences between the CBL-deficient patients and healthy donors. First, there was an increased usage of the Ig VH4-34 gene element in transitional, naïve and IgM⁺ memory B cells from CBL-deficient patients compared to healthy donors (Figure 6A). Second, we detected significantly reduced usage of *IGHJ6* and increased usage of *IGHJ4* genes by CBL-deficient transitional and naïve B cells relative to healthy donors (Figure 6B). Interestingly, CDR3 lengths of Ig expressed by transitional and naïve B cells were shorter for CBL-deficient patients compared to healthy donors (Figure 6C). This likely results from decreased usage of *IGHJ6* (Figure 6B), as this gene element contributes the highest number of amino acids to Ig CDR3 regions (15). Interestingly, while the shift away from *IGHJ6* usage typically occurs as healthy donor naïve B cells differentiate in memory B cells, this occurred at a much earlier stage of B-cell development in CBL-deficient patients (Figure 6B). To solidify these findings, we assessed BCR rearrangements from the bulk RNA-sequencing data of CD10⁺ CD20^{dim} HSPC-derived B cell progenitors. This analysis also revealed significantly reduced CDR3 lengths in the CBL UbLOF CD10⁺CD20^{dim} B cell progenitors compared to AAVS1KO cells (Figure 6D). This strongly suggests that aberrations to the BCR repertoire of CBL-LOH B cells is caused by a cell-intrinsic process during BCR locus rearrangement triggered by CBL deficiency. Furthermore, our analysis of BCR repertoires revealed significantly reduced levels of somatic hypermutation in CBL-deficient IgM⁺ memory B cells (Figure 6E). Indeed, when mutation load was quantified in terms of percentiles, the majority of clones isolated from IgM⁺ memory B cells from healthy donors exhibited a mutation rate of 2.5 to >7.5%, while most clones from CBL-deficient IgM⁺ memory B cells had accumulated somatic mutations at a rate of <2.5% (Figure 6F). Lastly, mutational targeting and selection, as determined by replacement:silent (R:S) ratios, was also significantly reduced in CBL-deficient versus CBL-

sufficient IgM+ memory B cells (Figure 6G). Combined, these findings are indicative of impaired selection and/or affinity maturation of memory B cells from CBL-deficient patients relative to healthy donors, resulting in qualitative differences between CBL-deficient and CBL-sufficient memory B cells.“

Reviewer #3

(Remarks to the Author)

This manuscript by Vatovec et al. investigates the role of the Casitas B lineage lymphoma proto-oncogene (CBL) in human lymphocyte development, maturation, and function, focusing on its distinct roles in T and B cells. The study examines patients with somatic homozygosity for loss-of-function (LOF) variants resulting in defective ubiquitination activity (UbLOF). The findings reveal differences between human and murine lymphocyte biology. In contrast to mouse models, where CBL has a significant role in T cell development and function, human CBL appears largely redundant in T cells. Patients with CBL UbLOF variants exhibited no major defects in T cell maturation or antigen responsiveness, highlighting species-specific variations in CBL function.

Conversely, human CBL is essential for B cell development and function. Patients with CBL UbLOF exhibited an increase in transitional B cell numbers during childhood and greater susceptibility to bacterial infections. The study demonstrates that CBL deficiency impairs B cell maturation through a cell-intrinsic mechanism involving reduced apoptosis and dysregulated B cell receptor (BCR) signaling. This dysregulation results in fewer memory B cells and disrupts adaptive immune memory. Additionally, CBL UbLOF alters bone marrow B cell development at the immature stage, promoting the survival and differentiation of autoreactive B cells. This impairment in B cell tolerance is associated with autoantibody production and an increased risk of autoimmune manifestations.

Overall, the study highlights the distinct role of CBL in human B cell biology and its redundancy in T cells. However, there are critical aspects that need to be addressed.

Specific comments:

1- The first two sections describing the patients' history and infectious disease events are overly descriptive and repetitive, making them difficult to follow. The focus on individual patient details, such as specific infectious episodes and their progression, detracts from the broader scientific implications. Additionally, much of this information, such as the occurrence of severe bacterial infections or associated conditions like JMML, has already been published in other reports.

To avoid repetition, we have moved the detailed patient description section to the supplemental material. Additionally, we have included an additional CBL LOH patient from India who succumbed severe pneumonia and inflammation, increasing the size of the cohort to 11 patients:

Patient P11:

P11 was a 6-year-old Indian boy with recurrent and severe infections beginning at 3 years of age, characterized by persistent fever, pneumonia with bilateral lung consolidations, multiple

skin and soft tissue infections (facial and peri-orbital abscesses, tinea corporis), and massive splenomegaly with hepatomegaly. Despite extensive evaluation, no pathogen could be consistently isolated from his pulmonary disease, but recurrent staphylococcal skin abscesses and fungal lesions were identified. Hematologic investigations revealed evolving cytopenias with leukoerythroblastosis, raising initial concern for juvenile myelomonocytic leukemia, although the clinical course was dominated by infection and inflammatory features, including markedly elevated ESR and hypergammaglobulinemia (IgG 2400 mg/dL, IgA 275 mg/dL). The child deteriorated with progressive hypersplenism and pulmonary complications, culminating in fatal infection-related decompensation at age 6, after which a homozygous private variant in CBL (p.Cys416Arg) was identified on genetic testing.

2- Figure 3 presents data from mass cytometry and flow cytometry, but the differing approaches to patient and control comparisons make the figure difficult to interpret. In mass cytometry, the ages of both patients and controls should be clearly indicated in both datasets to facilitate understanding and to highlight any age-related trends more effectively. In contrast to mass cytometry, the flow cytometry data are shown for patients (P1–P5) without accounting for age, despite these patients being from different age groups. This inconsistency is confusing and obscures potential patterns, particularly as the pediatric group appears to show the highest elevation in B cell subsets in the mass cytometry analysis. When the flow cytometry data are presented without similar age-group stratification, it becomes unclear how these findings align with the trends observed in the mass cytometry data.

We have now clarified and more clearly present the immunophenotyping data from mass cytometry and flow cytometry experiments in the revised version. In the figure and text, we now clearly refer to each methods used respectively. While the changes in frequencies of transitional B cells and B cells generally are clearly age-dependent, the changes in marker expression (e.g. Figure 3E, F) are present in both paediatric and adult CBL LOH patients, as evident by results from adult patients P4 and P6.

3- In Figure 4, the authors investigate the developmental block in B cell differentiation in patients and conclude that the transition from immature to mature B cells is impaired. To validate this finding, they perform a gene editing experiment in CD34+ hematopoietic progenitor cells, aiming to delete exon 8 of CBL. This deletion is intended to create a ubiquitination-deficient CBL protein while preserving the expression of the remaining portions of the protein. However, the data shown in Figure 3C cast doubt on the authors' interpretation. While the authors suggest that CBL exon 8 deficiency leads to a differentiation block, the results do not clearly support this conclusion. In fact, the differentiation of CBL-edited cells appears better than with the AAVS1 control. On the other hand, the data strongly suggest that CBL gene editing affects cell survival. This discrepancy makes it difficult to follow their interpretation. Further issues are evident in Supplementary Figure 4, where the authors attempt to demonstrate the efficiency of the gene editing. The data suggest that sgRNA 1+3 results in complete CBL downregulation rather than exon editing, while sgRNA 1+2 is more effective in generating the exon 8 deletion. However, the absence of a loading control makes it impossible to confirm whether the rest of the CBL protein is expressed as expected or to evaluate the efficiency and specificity of the editing.

We respectfully disagree with the Reviewer about the uncertainty of the editing efficiency at the CBL locus in our CD34+ HSPC B cell differentiation system.

To prove successful editing at the *CBL* locus, we PCR amplified genomic DNA from control or *CBL*-edited cells 72 hours after nucleofection. Mock transfected cells showed amplification of the complete locus (922 bp), whereas *CBL*-edited cells yielded a smaller band at ~500 bp, corresponding to the ~400 bp in-frame deletion of exon 8 by our duo-sgRNA approach (see **revised Figure S4D**). To better clarify this point, we have revised **Figure S4** and included a gel with more comparable *CBL* amplification, which shows similar editing efficiency with all three guide pairs used. Quantification of editing by next generation sequencing (NGS) showed that 70-80% of amplified DNA fragments carried a complete deletion of exon 8 (**revised Figure S4D-F**) further confirming that most cells were successfully edited. We would also emphasize that these results are consistent with our prior experience in editing other loci and assessing editing efficiencies in primary human CD34⁺ HSPCs (16-18).

Revised Figure S4D-G: Modelling *CBL* ΔExon 8 variant in primary human CD34⁺ HSPCs. (D) Agarose gel electrophoresis of PCR products for the *AAVS1* or *CBL* loci performed on genomic DNA isolated from CD34⁺ HSPCs 72 hours after nucleofection. Representative of $n > 5$ biological replicates. (E) Editing efficiency at the *CBL* locus with sgRNA pair 1+2 by NGS, showing ~80% editing efficiency of ~400bp deletion containing exon 8. (F) Quantification of exon 8 deletions at the *CBL* locus determined by NGS for all three guide pairs. $n =$ two biological replicates. (G) Editing efficiencies on day 8 and day 21 post editing in differentiation cultures as determined by NGS.

In addition, western blotting of edited CD34⁺ HSPCs edited at the *CBL* locus confirmed that the deletion of exon 8 did not result in reduced *CBL* protein levels (Figure R2).

Figure R2: CBL protein expression in gene edited primary human CD34+ HSPCs. Western Blot of AAVS1- or *CBL*-edited CD34+ HSPCs performed 7 days post nucleofection.

Our *in vitro* differentiation data show that *CBL* Ub^{LOF} results in an increased proportion of CD10⁺CD20⁺ cells, as correctly highlighted by Reviewer 3. As we also observe an accumulation of CD10^{hi}CD20^{hi} cells in BM of 2 *CBL*-deficient patients, we interpret this increase in immature B cells because of reduced apoptosis of autoreactive B cells and/or reduced capacity of the cells to fully mature. Our *in vitro* culture system is limited in its capacity to recapitulate all stages of human B cell development and relies on immortalized mouse feeder cells (MS-5) and a limited set of cytokines as a replacement for the niche provided by the endogenous human bone marrow niche. This limits our ability to test subsequent stages of B cell development. However, our data of annexin V staining on bulk CD19⁺ cells generated from gene-edited HSCs *in vitro* (see **Figure 4H**) suggest no difference in cell death between control and *CBL*-edited cells in the absence of stimulation.

Another limitation of the study is the reliance on data from monozygotic siblings (P1, P2, and P3) as key patient representatives. Since these patients share identical genetic and environmental backgrounds, their data may not fully capture the broader variability among individuals with *CBL* deficiency. Including data from unrelated patients would provide a more comprehensive and generalizable assessment of the effects of *CBL* deficiency on B cell differentiation and survival.

We agree with the reviewer that the analysis of monozygotic triplets could be a limitation of our study. However, we would like to highlight that while the study of these 3 related individuals certainly laid the foundation for this project, almost all aspects of our study have been performed using material available from additional unrelated patients, with highly reproducible and consistent data generated. This includes for example:

- allelic ratios (Fig 1D, E, *n*=6)
- T cell and NK cell development and differentiation *in vivo* (Fig 2, Supp Fig 3, *n*=7)
- serum Ig levels (*n*=10, Fig 3), B cell numbers (*n*=8, Fig 3), determination of B-cell subsets (*n*=7, Fig 3) and extended phenotyping (*n*=5, Fig 3, 4 and Fig S4), *in vitro* culture of isolated B cell subsets (*n*=5, Fig 5), assessment of autoreactivity using 9G4 staining (*n*=5, Fig 6), autoAb arrays

($n=8$, Fig 7), BCR repertoire ($n=5$, Fig 6) and quantification of SARS-CoV2-specific B cell ($n=4$) and IgG ($n=5$) responses.

Currently, the data sets that have been derived from only these 3 related patients are analysis of BM samples. Unfortunately, extending analysis of B cell development using BM samples from additional patients is not feasible as there are no medical indications for patients to undergo invasive procedures such as BM aspiration/biopsy. Despite this, we have developed invaluable models to complement and orthogonally confirm the findings from analysis of primary patient material, and to explore cellular defects at a detailed molecular and mechanistic level. This includes recapitulating B cell development from HPSCs *in vitro* and gene editing in CD34⁺ cells to inactivate CBL, and establishing various experimental knock-in/knock-out B cell lines.

We would also take this opportunity to emphasise the following points:

- relying on data obtained from detailed analysis of monozygotic triplets is akin to many publications utilising mouse models, which is also highly constrained genetically and environmentally. However, we are unable to recall any study that tested multiple and distinct strains of inbred mice that have been engineered to lack the same specific gene in order to exclude a role for background genotype on a phenotype.
- CBL-deficiency is a rare disease, to date, only around 100 affected individuals have been reported (7, 19, 20). Thus, being able to perform a large number of experiments on many of these patients is a significant achievement. This should be recognised as a positive, rather than being considered as a limitation and as a reason to be dubious of the generality of the findings for all CBL-deficient patients. As a comparison, our data would be the equivalent of undertaking extensive studies in >5 genetically distinct strains of *Cbl*-deficient mice.

4- The authors proceed to investigate apoptosis in CBL Ub-LOF patients and conclude that defective apoptosis is the primary reason for the accumulation of immature B cells. However, they rely on gene-edited CD34⁺ hematopoietic progenitor cells from *in vitro* differentiation experiments to support this conclusion, despite the issues identified with the gene editing experiment. To further support their hypothesis, the authors use the REH cell line as a model to investigate apoptosis. However, the rationale for selecting this cell line is unclear. REH cells represent progenitor B cells lacking cytoplasmic heavy chain expression, making them fundamentally distinct from the immature B cells that accumulate in CBL Ub-LOF patients. This discrepancy raises questions about the relevance of the REH cell model to the patient phenotype. The authors do not provide sufficient justification for its use or explain how results from REH cells are applicable to the context of CBL Ub-LOF patients. As a result, the connection between defective apoptosis and the observed accumulation of immature B cells remains speculative.

We have selected the REH cell line as a model for these experiments because of their intrinsically high expression of CD38, which phenocopies B cells from CBL-deficient patients and is a property of both pre- and immature B cells. Furthermore, BJAB cells do not express high levels of CD38 and do not respond with apoptosis to CD38 crosslinking, rendering this cell line unsuitable to model the impact of CBL deficiency on CD38 signalling. We have now added the surface expression levels of CD38 of HEK293T, BJAB and REH cells to the manuscript:

Figure S5H: Surface staining of CD38 expressed on HEK293T, REH and BJAB cells as determined by flow cytometry.

The Reviewer is correct in pointing out that REH cells do not express surface BCR and may therefore be classified as more similar to a pre-B cell than an immature B cell. Nonetheless, we believe REH B cells are a useful and relevant model to assess CD38 signalling and activity, independently of the BCR. Given their transformed status, they should anyway not be considered as a physiological model, for which our CD34⁺ HSPC differentiation is much more appropriate.

5- The authors investigate antibody secretion in CBL Ub-LOF patients and conclude that B cells from these patients are impaired in their ability to produce immunoglobulins (Ig). However, this interpretation raises several critical issues. First, the observation of hypergammaglobulinemia in CBL Ub-LOF patients (Figure 3B) contradicts the claim of impaired Ig secretion. Hypergammaglobulinemia suggests an overall increase in Ig levels, which is difficult to reconcile with reduced Ig secretion by patient B cells.

Here, we refer to our previous discussion and comments on the hypergammaglobulinemia in CBL-deficient patients, as outlined above.

It is also important to recognise that measurement of levels of polyclonal Ig cannot be relied on as a correlate of intact humoral immune responses. Clear examples of this include patients with inborn errors of immunity due to recessive mutations in *DOCK8* or dominant mutations in *STAT3*, both causing hyper IgE syndrome, or even some cases of activated PI3K delta syndrome (APDS) due to activating variants in *PIK3CD* (encoding p110d) or loss of function mutations in *PIK3R1* (encoding p85). Many of these patients have normal or even increased levels of total serum IgM or IgG, as well as IgE in cases of hyper-IgE syndrome – however, levels of Ig against specific antigens are all reduced, indicating that the pathways responsible for inducing basal levels of serum Ig can be distinct from those required for inducing robust long-lived protective antibodies against pathogens or vaccines (see Leung et al, *J Allergy Clin Immunol* 1988; Sheerin et al *J Allergy Clin Immunol* 1991; Avery et al, *J Exp Med* 2010; Jabara et al *Nat Immunol* 2012; Stentzel et al *Clin Inf Dis* 2017; Avery et al *J Exp Med* 2018; Renner et al *Clin Immunol* 2021; Nguyen et al *J Exp Med* 2023). Thus, we plan to extend the data presented in the original manuscript relating to reduced frequencies of SARS-CoV2 spike-binding B cells and determine levels of total and neutralising IgG against SARS-CoV2 in CBL-deficient patients over time post vaccination. We believe these data, together with all the additional data to come from the above proposed experiments will resolve this enigma of impaired humoral immunity despite hypergammaglobulinemia.

Second, the authors report reduced IgG/A secretion by transitional B cells, which is biologically problematic. Transitional B cells are immature and have not undergone class switch recombination, making it unlikely that they would secrete IgG or IgA. This raises concerns about the purity of the sorted subsets and whether contaminating mature or memory B cells could be contributing to the observed results.

In terms of this comment, we respectfully disagree with Reviewer 3. First, there is abundant evidence that transitional B cells *are* capable of being activated *in vitro* and undergoing class switching to produce different Ig isotypes. We first reported this in 2006 (see Table I in (2)). This paper clearly demonstrated that transitional B cells **DO** produce detectable levels of IgM, IgG and IgA in response to stimulation through CD40L + BCR engagement, or CD40L/BCR and IL-10. We did conclude that “*CD24^{hi}CD38^{hi} [i.e. transitional] B cells produce low amounts of Ig*”, but **this was in comparison to naïve B cells, which produced 5-20-fold higher levels of Ig in vitro**. When we followed up this initial study and examined the capacity of human transitional B cells to produce Ig *in vitro*, we included additional stimuli such as the TLR agonist CpG and IL-21 (see Table 3 in (21)). This study confirmed our original findings that both CD21^{lo} and CD21⁺ subsets of human transitional B cells secrete detectable amounts of IgM, IgG and IgA *in vitro* (i.e. undergo class switching) in response to CD40L/IL-21, CD40L/CpG, or CD40L/CpG/IL-21. Similar to our initial study from 2006, the magnitude of the responses of transitional B cells was modestly reduced compared to naïve B cells. This is not just a phenomenon restricted to human B cells. Indeed, Wesemann and Alt reported that immature B cells isolated from mice preferentially undergo class switching to IgE (22). For these reasons, we do not agree that Reviewer 3 is correct to suggest that human transitional B cells “*have not undergone class switch recombination, making it unlikely that they would secrete IgG or IgA*”. We also routinely confirm the purity of sorted B cell subsets, and these are >98%.

6- The data using 9G4 and the IgG reactivity against a wide range of human antigens point to increased autoreactivity, several aspects require further clarification. The authors associate the increase in 9G4 staining with autoreactive potential but do not address whether the elevated fluorescence intensity reflects increased BCR expression. Assessing IgM and IgD levels on B cells from patients compared to controls would help determine whether the increased staining reflects changes in BCR density or is solely due to the painting effect of circulating autoantibodies.

As noted above in response to Reviewer 1, we performed flow cytometric analysis to assess expression levels of IgM and IgD on transitional and naïve B cells from 4 CBL-deficient patients and 10 healthy donors. As human B cells develop from a transitional into a naïve state, expression of surface IgM is significantly down-regulated (2, 3). While we observed a reduction in IgM expression on CBL-deficient naïve B cells compared to CBL-deficient transitional B cells, the overall levels of IgM on CBL-deficient B cells remained 2-fold greater than on corresponding B cell subsets from healthy donors. Indeed, the level of IgM on CBL-deficient naïve B cells was comparable to levels on transitional B cells from healthy donors, indicating an arrest at the transitional g naïve stage of B cell development. In contrast to IgM, expression of IgD was reduced on transitional and naïve B cells from CBL-deficient patients compared to these cell subsets from healthy donors, being statistically significant for naïve B cells ($p=0.0027$, >2 fold difference; Fig S4B). Furthermore, while IgD increased as transitional B cells developed into naïve B cells in healthy donors, this upregulation was not observed for for CBL-deficient B cells (Fig S4B).

Figure S4B: Quantification of IgM and IgD surface expression on primary transitional and naïve B cells of healthy donors and the indicated CBL LOH patients. Statistical significance was assessed with Mann Whitney tests corrected for multiple testing.

While this increased in BCR levels on CBL-deficient B cells may make a modest contribution to increased binding of the 9G4 mAb, it is highly unlikely to account for the 6-10-fold increase in gMFI for 9G4 binding to CBL-deficient transitional and naïve B cells relative to healthy donor B cells (Figure 6A in our original manuscript). Thus, we believe our data reflect increased autoreactivity of intrinsically expressed BCR on CBL-deficient B cells. This of course is consistent with our new data showing greater usage of the well-described autoreactive Ig V gene VH4-34 by CBL-deficient transitional and naïve B cells.

While the shared autoreactive targets identified through protein microarray analysis suggest common mechanisms underlying autoreactivity, functional validation of these targets is needed to confirm their biological relevance. These additional investigations would help clarify the mechanisms by which CBL Ub-LOF leads to disrupted tolerance and autoreactivity, strengthening the conclusions drawn from this study.

We are not completely clear on how to interpret this query. “*Functional validation of the hits from our protein microarray to confirm biological relevance*” may refer to an experimentally independent validation of the presence of such autoantibodies by, for example, ELISA. Following the reviewers suggestion, we validated the presence of increased titers of anti-TXLNB IgG antibodies in patients P1, P2 and P3 through bead-based ELISA (Figure S7I):

Figure S7I: Validation of presence of anti-TXLNB autoantibodies in three CBL-deficient patients by bead-based ELISA.

Alternatively, this query may refer to functional validation of the autoantibodies and potential clinical relevance. For example, assessing formation of immune complexes or neutralization of self-antigen in relevant cellular assays. If this is the intended meaning of the comment, we would respectfully contest the reviewer's assessment. Firstly, we would like to mention that these are intracellular antigens that are unlikely to have a biological or clinical implication in the absence of cell death and inflammation. Consistently, the patients in this cohort do not have clinical autoimmune diseases. Second, we think the deep investigation of cellular and clinical relevance of the autoantibodies detected here goes beyond the scope of this current study. We present data from several independent experimental methods that clearly demonstrates a break in the tolerance of the patients B cells. The downstream consequences of this break of tolerance will be explored in a follow up investigation.

7- The figure legends remain suboptimal and fail to provide sufficient detail to aid interpretation.

We have expanded the figure legends with more technical details. Full methodological description are found in the methods section, as the length of figure legends is quite restricted.

Reviewer References:

1. M. C. Crank *et al.*, Mutations in PIK3CD can cause hyper IgM syndrome (HIGM) associated with increased cancer susceptibility. *J Clin Immunol* **34**, 272-276 (2014).
2. A. K. Cuss *et al.*, Expansion of functionally immature transitional B cells is associated with human-immunodeficient states characterized by impaired humoral immunity. *J Immunol* **176**, 1506-1516 (2006).
3. G. P. Sims *et al.*, Identification and characterization of circulating human transitional B cells. *Blood* **105**, 4390-4398 (2005).
4. J. A. Duty *et al.*, Functional energy in a subpopulation of naive B cells from healthy humans that express autoreactive immunoglobulin receptors. *J Exp Med* **206**, 139-151 (2009).

5. T. D. Quach *et al.*, Anergic responses characterize a large fraction of human autoreactive naive B cells expressing low levels of surface IgM. *J Immunol* **186**, 4640-4648 (2011).
6. L. Moens, S. G. Tangye, Cytokine-Mediated Regulation of Plasma Cell Generation: IL-21 Takes Center Stage. *Front Immunol* **5**, 65 (2014).
7. J. Bohlen *et al.*, Autoinflammation in patients with leukocytic CBL loss of heterozygosity is caused by constitutive ERK-mediated monocyte activation. *J Clin Invest* **134**, (2024).
8. C. S. Ma *et al.*, Monogenic mutations differentially affect the quantity and quality of T follicular helper cells in patients with human primary immunodeficiencies. *J Allergy Clin Immunol* **136**, 993-1006 e1001 (2015).
9. S. Suematsu *et al.*, IgG1 plasmacytosis in interleukin 6 transgenic mice. *Proc Natl Acad Sci U S A* **86**, 7547-7551 (1989).
10. K. Balmanno, S. J. Cook, Tumour cell survival signalling by the ERK1/2 pathway. *Cell Death Differ* **16**, 368-377 (2009).
11. A. Bonni *et al.*, Cell survival promoted by the Ras-MAPK signaling pathway by transcription-dependent and -independent mechanisms. *Science* **286**, 1358-1362 (1999).
12. C. Le Coz *et al.*, CD40LG duplication-associated autoimmune disease is silenced by nonrandom X-chromosome inactivation. *J Allergy Clin Immunol* **141**, 2308-2311 e2307 (2018).
13. C. H. Clegg *et al.*, Thymus dysfunction and chronic inflammatory disease in gp39 transgenic mice. *Int Immunol* **9**, 1111-1122 (1997).
14. D. T. Avery *et al.*, Germline-activating mutations in PIK3CD compromise B cell development and function. *J Exp Med* **215**, 2073-2095 (2018).
15. H. Wang *et al.*, Antibody heavy chain CDR3 length-dependent usage of human IGHJ4 and IGHJ6 germline genes. *Antib Ther* **4**, 101-108 (2021).
16. J. D. Martin-Rufino *et al.*, Transcription factor networks disproportionately enrich for heritability of blood cell phenotypes. *Science* **388**, 52-59 (2025).
17. R. A. Voit *et al.*, A genetic disorder reveals a hematopoietic stem cell regulatory network co-opted in leukemia. *Nat Immunol* **24**, 69-83 (2023).
18. J. Zhao *et al.*, Human hematopoietic stem cell vulnerability to ferroptosis. *Cell* **186**, 732-747 e716 (2023).
19. A. Hecht *et al.*, Molecular and phenotypic diversity of CBL-mutated juvenile myelomonocytic leukemia. *Haematologica* **107**, 178-186 (2022).
20. T. Yoshida *et al.*, Clinical and molecular features of CBL-mutated juvenile myelomonocytic leukemia. *Haematologica* **108**, 3115-3119 (2023).
21. S. Suryani *et al.*, Differential expression of CD21 identifies developmentally and functionally distinct subsets of human transitional B cells. *Blood* **115**, 519-529 (2010).
22. D. R. Wesemann *et al.*, Immature B cells preferentially switch to IgE with increased direct Smu to Sepsilon recombination. *J Exp Med* **208**, 2733-2746 (2011).

Dear Dr. Dempsey,

Thank you for providing us with the comments from Reviewers 1 and 3 following submission of our revised manuscript “*Somatic deficiency of human CBL in leukocytes impairs B cell but not T cell development and function*” by Vatovec et al. We are pleased that our response/rebuttal satisfactorily addressed all the issues raised by Reviewer 1, and appreciate that we have the opportunity to address the remaining concerns raised by Reviewer 3. Importantly, the technical concerns raised about single experiments or samples do not impact the general conclusions of the manuscript and can be fully addressed, as we outline below. The major conclusions of our work are not impacted by these issues.

Here are our detailed responses:

Reviewer #3 comment: “While the newly added data in Figure S4, including the improved CBL amplification gel, are satisfactory, the western blot showing CBL protein expression in gene-edited CD34⁺ HSPCs remains unconvincing. The vinculin loading control for the CBL sgRNA (1+3) sample displays a markedly stronger signal than the AAVS1 sgRNA (1+2) control, while the CBL band intensity appears comparable. This pattern is consistent with reduced CBL protein levels in the edited cells, contrary to the authors’ statement that CBL expression is unaffected (“Figure R2”).”

Author response: We thank the Reviewer for this observation. We agree that sample loading in the previously presented western blot was not optimal and this can make it difficult to draw firm conclusions regarding the relative amount of CBL protein in CD34⁺ HSPCs edited with CBL sgRNA (1+3) compared with the AAVS1 control. To address this concern, we have performed additional western blots and quantified CBL protein expression relative to the loading control. These additional analyses show that CBL protein levels are largely comparable across all three pairs of sgRNA used (ED Figure 4F-G). This finding is consistent with the genomic DNA sequencing data presented in the manuscript (see ED Figure 4D-G of the revised manuscript), demonstrating that the majority of cells carry the intended in-frame exon 8 deletion without evidence of frameshifts or other undesired insertions or deletion events. We have included these new data in the manuscript in extended data figure 4F-G.

ED Figure 4F-G: CBL protein expression in gene-edited primary human CD34⁺ HSPCs. (F) Western Blot of AAVS1- or CBL-edited CD34⁺ HSPCs performed 7 days after nucleofection. A total of 30 μ g protein per sample was loaded onto 4-15% TGX gels. Proteins were transferred onto nitrocellulose membranes, incubated with primary antibodies overnight, and subsequently incubated with HRP-conjugated secondary antibodies. Signal detection was performed using chemiluminescence. (G) Quantification of CBL protein expression relative to loading control for three western blot replicates.

Reviewer #3 comment: *“Furthermore, the authors’ explanation regarding class switching and antibody secretion by transitional B cells does not resolve the issue. The data presented suggest comparable IgA and IgG secretion by transitional and naïve B cells under the tested conditions. This is biologically unexpected. The earlier report (2006), which is cited as support, described class switching by transitional B cells after stimulation through CD40L + BCR engagement, or CD40L/BCR + IL-10, resulting in low-level Ig production compared with naïve B cells. However, the current study stimulated transitional cells with CD40L + IL-21, making the comparison inappropriate. Clarification is still required as to why transitional B cells in the present experiments show IgA and IgG secretion comparable to naïve B cells.”*

Author response: To expand on our initial response to this point raised by Reviewer #3, and to provide greater explanation and clarification to this issue, we have now included additional supporting data in the revised manuscript to show class switching and Ig secretion by transitional B cells under myriad *in vitro* culture conditions.

We optimised several *in vitro* culture systems to induce activation of distinct subsets of human B cells. Basically, this involved isolating and stimulating transitional, naïve and memory B cells isolated from healthy donors with agonists that engage CD40, Toll-like receptors (TLR7, TLR9), the BCR or cytokine receptors. This optimisation was based on numerous previous studies from the Tangye lab¹⁻⁹ and other investigators¹⁰⁻¹⁵ who also reported methods to induce activation, proliferation, differentiation, cytokine production and Ig secretion by human B-cell subsets using combinations of CD40L, CpG or R848, anti-Ig, or IL-2, IL-10, IL-15 or IL-21.

Our experiments established that when transitional or naïve B cells are stimulated via CD40 (CD40L), TLR9 (CpG) or the BCR (BCR crosslinking) alone, or the combination of CD40/BCR, very little Ig (IgM, IgG, IgA) is produced (**ED Figure 7A**). In contrast, co-stimulation with CD40L/CpG or CD40L/CpG/BCR induces robust IgM secretion by **naïve B cells**. Notably, **transitional B cells** also produce IgM, **but only in response to the combination of CD40/TLR9/BCR agonists (ED Figure 7A)**. **IgM levels detected following stimulation of naïve B cells with CD40L/CpG or CD40L/CpG/BCR are markedly higher than those observed for transitional B cells subjected to the same stimulatory conditions (30-fold and 5-fold, respectively; see ED Figure 7A-B and Figure R1)**. Under these same conditions, only low amounts (5 – 30 ng/mL) of switched Ig (IgG, IgA) are induced by transitional or naïve B cells, with substantial interindividual variability.

Another culture condition we routinely use is the combination of CD40L and IL-21, which represents a mimic of T cell-dependent B cell stimulation¹⁶⁻²². CD40L/IL-21 induces robust IgM secretion, as well as Ig class switching, **yielding IgG and IgA production by both transitional and naïve B cells (ED Figure 7B)**. Naïve B cells generally secrete more IgM than transitional B cells, while IgG and IgA levels are comparable between the transitional and naïve B cell subsets (**ED Figure 7B**). Importantly, **levels of Ig induced by CD40L/IL-21 stimulation are >10-fold higher than those obtained with CD40L/CpG or CD40L/CpG/BCR, indicating that the latter conditions provide weaker activation signals than those achieved in the presence of cytokines (compare ED Figure 7A-B)**.

ED Figure 7A-B: Transitional B cells ($n=4$ independent experiments) and naïve B cells ($n=12$ independent experiments) were sort-purified from PBMCs of healthy donors and then cultured *in vitro*. Levels of IgM, IgG and IgA were measured in supernatants from transitional and naïve B cells after 7 days following stimulation with: (A) CD40L alone or in combination with CpG and/or BCR crosslinking, or (B) CD40L and IL-21.

Figure R1: Figure Panel 5A from the submitted manuscript with thick red lines illustrating the differences in IgM secretion by transitional versus naïve B cells in our cultures.

These data presented in **ED Figure 7A and B** are consistent with those in **Figure 5A** of our revised manuscript: ie naïve B cells produce more IgM than transitional B cells in response to T cell-dependent (CD40L/IL-21) or CD40L/CpG/BCR stimuli (also refer to **Figure R1** for illustration of that difference). However, under T cell-dependent stimulatory conditions, both transitional and naïve B cells produce comparable levels of IgG and IgA. As minimal Ig class switching and secretion is observed even for naïve B cells when cultured in the absence of IL-21, it is clear that human B cells require a cytokine-specific signal to induce significant IgG and IgA production^{1-3,5,17,23-34}. Given the potency of IL-21 in inducing proliferation, differentiation, Ig class switching and plasma cell generation in human B cells, it is perhaps not surprising that transitional and naïve B cells produce comparable amounts of IgG and IgA in response to co-stimulation with CD40L plus IL-21.

Taken together, there is no discrepancy in our data regarding Ig production by transitional and naïve B cells. The data presented here, and in our manuscript, reinforce that:

- **In response to weaker signals provided through the BCR, CD40 or TLR9, induction of IgM secretion in vitro by naïve B cells exceeds that induced by transitional B cells;**

- **In response to stronger signals provided by IL-21, transitional and naïve B cells will undergo class switching to produce IgG and IgA at similar levels in vitro.**

These observations address and explain Reviewer 3's concern that "*The data presented suggest comparable IgA and IgG secretion by transitional and naïve B cells under the tested conditions*". These results reflect the superior activation induced by CD40/IL-21 over BCR/CD40/TLR9 for induction of secretion of switched Ig isotypes.

Reviewer 3 also states that "*the current study stimulated transitional cells with CD40L + IL-21*". This is not completely correct – as detailed above and in **Figure 5A** in our original manuscript (see also **Figure R1**) transitional, naïve and memory B cell subsets purified from healthy donors and CBL-deficient patients were cultured *in vitro* **with CD40L/IL-21 OR CD40L/CpG/BCR**.

Overall, we would like to emphasise that a key conclusion of our study – as presented in **Figure 5A** of the revised manuscript - is that **irrespective of the B cell subsets being examined (transitional, naïve, memory), CBL deficiency causes significant reduction in production of IgM, IgG and IgA induced in response to diverse stimuli.**

In summary, and consistent with many peer-reviewed publications on regulation of Ig secretion by human B lymphocytes^{1,3-7,10-16,18,19,23,24,26-28,30,34}, the data we present in the current manuscript are:

- 1) **transitional B cells produce less IgM than naïve B cells** upon e.g. CD40L/CpG/BCR stimulation. Importantly, this validates the separation and purification of transitional versus naïve B cells in the current manuscript;
- 2) **transitional and naïve B cells produce relatively low, but comparable amounts of IgA and IgG upon CD40L+IL21 stimulation, driven by IL21 induced class switching in vitro (reproducing previously-published data)^{3,5,7,16,18,19};**
- 3) **Ig secretion by CBL-deficient B cells is broadly impaired in all subsets and models tested** (transitional, naïve, memory; plasma cells; two types of isogenic BJAB cells).

We have now included the results shown in **ED Figure 7A-B** in the manuscript, although they largely recapitulate the findings shown in **Figure 5A** and prior studies.

Reviewer #3 comment: "*Finally, the REH cell experiments remain inappropriate for addressing the mechanistic question of impaired apoptosis in CBL-deficient B cells. REH cells are transformed pre-B cells lacking surface BCR expression and thus cannot model the immature B cells implicated in the patient phenotype. High CD38 expression alone does not justify their use as a surrogate, and signaling dynamics in this leukemic context cannot be extrapolated to normal immature B cells. Consequently, these data do not substantiate the proposed mechanism linking CBL deficiency to defective apoptosis and B cell accumulation.*"

Author response: We would like to emphasize again that REH cells were used solely as a model cellular assay to investigate CD38 signalling. These cells express a functional CD38 pathway, including surface expression of CD38. Following receipt of the original reviews of our manuscript, we had discussed with Dr. Dempsey whether it would be necessary to perform analogous experiments in another cell line (MN60). The editorial team indicated to us that this would not be required to sufficiently address this concern.

We acknowledge that REH cells represent a model with limited direct applicability to primary immature B cells. To strengthen the relevance of our findings, we replicated the most pertinent observation in our more physiological model: recapitulating B cell development from HSPCs. In this system, we similarly observed reduced apoptosis upon CD38 stimulation with anti-CD38 mAb treatment (see **Figure 4H**). This is consistent with results obtained in REH cells (see **Figure 4I**), suggesting this is a valid, reproducible result across independent model systems.

We have included additional data in the revised manuscript that demonstrate that elevated pERK signalling downstream of CD38 is observed in CBL-deficient HSPC-derived CD19⁺ progenitors (**ED Figure 6G-H**), consistent with our finding of elevated ERK signalling in REH cells (see **Figure 4J** and **ED Figure 6**). These data further support the consistency between the two cellular models and strengthen the confidence in our findings.

ED Figure 6G-H: ERK phosphorylation upon CD38 crosslinking in gene-edited HSPC-derived CD19⁺ B progenitors. Total CD19⁺ (**G**) or CD19⁺CD20^{high} (**H**) B progenitors were sorted from co-cultures edited at the AAVS1 or CBL locus. Cells were stimulated for 15 minutes with Daratumumab, followed by fixation, permeabilization and intracellular staining for phosphorylated ERK (pERK). *n* = 2 biological replicates.

We have more explicitly indicated the limitations of this model in a dedicated section in the discussion:

“LIMITATIONS OF THIS STUDY

REH cells are a leukemic cell line that lacks BCR expression, they therefore potentially have limited applicability to model transitional B cells, which typically express surface IgM.”

We have also removed the description of REH cells as an “immature” B cell line.

References cited:

1. Suryani S, Fulcher DA, Santner-Nanan B, et al. Differential expression of CD21 identifies developmentally and functionally distinct subsets of human transitional B cells. *Blood*. 2010;115(3):519-529.
2. Cuss AK, Avery DT, Cannons JL, et al. Expansion of functionally immature transitional B cells is associated with human-immunodeficient states characterized by impaired humoral immunity. *J Immunol*. 2006;176(3):1506-1516.
3. Avery DT, Bryant VL, Ma CS, de Waal Malefyt R, Tangye SG. IL-21-induced isotype switching to IgG and IgA by human naive B cells is differentially regulated by IL-4. *J Immunol*. 2008;181(3):1767-1779.
4. Avery DT, Deenick EK, Ma CS, et al. B cell-intrinsic signaling through IL-21 receptor and STAT3 is required for establishing long-lived antibody responses in humans. *J Exp Med*. 2010;207(1):155-171.
5. Bryant VL, Ma CS, Avery DT, et al. Cytokine-mediated regulation of human B cell differentiation into Ig-secreting cells: predominant role of IL-21 produced by CXCR5+ T follicular helper cells. *J Immunol*. 2007;179(12):8180-8190.
6. Good KL, Avery DT, Tangye SG. Resting human memory B cells are intrinsically programmed for enhanced survival and responsiveness to diverse stimuli compared to naive B cells. *J Immunol*. 2009;182(2):890-901.
7. Recher M, Berglund LJ, Avery DT, et al. IL-21 is the primary common gamma chain-binding cytokine required for human B-cell differentiation in vivo. *Blood*. 2011;118(26):6824-6835.
8. Tangye SG, Avery DT, Deenick EK, Hodgkin PD. Intrinsic differences in the proliferation of naive and memory human B cells as a mechanism for enhanced secondary immune responses. *J Immunol*. 2003;170(2):686-694.
9. Good KL, Tangye SG. Decreased expression of Kruppel-like factors in memory B cells induces the rapid response typical of secondary antibody responses. *Proc Natl Acad Sci U S A*. 2007;104(33):13420-13425.
10. Bernasconi NL, Onai N, Lanzavecchia A. A role for Toll-like receptors in acquired immunity: up-regulation of TLR9 by BCR triggering in naive B cells and constitutive expression in memory B cells. *Blood*. 2003;101(11):4500-4504.
11. Bernasconi NL, Traggiai E, Lanzavecchia A. Maintenance of serological memory by polyclonal activation of human memory B cells. *Science*. 2002;298(5601):2199-2202.
12. Poeck H, Wagner M, Battiany J, et al. Plasmacytoid dendritic cells, antigen, and CpG-C license human B cells for plasma cell differentiation and immunoglobulin production in the absence of T-cell help. *Blood*. 2004;103(8):3058-3064.
13. Huggins J, Pellegrin T, Felgar RE, et al. CpG DNA activation and plasma-cell differentiation of CD27- naive human B cells. *Blood*. 2007;109(4):1611-1619.
14. Jiang W, Lederman MM, Harding CV, Rodriguez B, Mohner RJ, Sieg SF. TLR9 stimulation drives naive B cells to proliferate and to attain enhanced antigen presenting function. *Eur J Immunol*. 2007;37(8):2205-2213.
15. Malaspina A, Moir S, DiPoto AC, et al. CpG oligonucleotides enhance proliferative and effector responses of B Cells in HIV-infected individuals. *J Immunol*. 2008;181(2):1199-1206.
16. Ettinger R, Sims GP, Fairhurst AM, et al. IL-21 induces differentiation of human naive and memory B cells into antibody-secreting plasma cells. *J Immunol*. 2005;175(12):7867-7879.
17. Ettinger R, Kuchen S, Lipsky PE. The role of IL-21 in regulating B-cell function in health and disease. *Immunol Rev*. 2008;223:60-86.
18. Pene J, Gauchat JF, Lecart S, et al. Cutting edge: IL-21 is a switch factor for the production of IgG1 and IgG3 by human B cells. *J Immunol*. 2004;172(9):5154-5157.
19. Kuchen S, Robbins R, Sims GP, et al. Essential role of IL-21 in B cell activation, expansion, and plasma cell generation during CD4+ T cell-B cell collaboration. *J Immunol*. 2007;179(9):5886-5896.
20. Diehl SA, Schmidlin H, Nagasawa M, et al. STAT3-mediated up-regulation of BLIMP1 is coordinated with BCL6 down-regulation to control human plasma cell differentiation. *J Immunol*. 2008;180(7):4805-4815.

21. Moens L, Tangye SG. Cytokine-Mediated Regulation of Plasma Cell Generation: IL-21 Takes Center Stage. *Front Immunol.* 2014;5:65.
22. Tangye SG, Ma CS. Regulation of the germinal center and humoral immunity by interleukin-21. *J Exp Med.* 2020;217(1).
23. Briere F, Servet-Delprat C, Bridon JM, Saint-Remy JM, Banchereau J. Human interleukin 10 induces naive surface immunoglobulin D⁺ (sIgD⁺) B cells to secrete IgG1 and IgG3. *J Exp Med.* 1994;179(2):757-762.
24. Defrance T, Vanbervliet B, Briere F, Durand I, Rousset F, Banchereau J. Interleukin 10 and transforming growth factor beta cooperate to induce anti-CD40-activated naive human B cells to secrete immunoglobulin A. *J Exp Med.* 1992;175(3):671-682.
25. Malisan F, Briere F, Bridon JM, et al. Interleukin-10 induces immunoglobulin G isotype switch recombination in human CD40-activated naive B lymphocytes. *J Exp Med.* 1996;183(3):937-947.
26. Pene J, Rousset F, Briere F, et al. IgE production by normal human B cells induced by alloreactive T cell clones is mediated by IL-4 and suppressed by IFN-gamma. *J Immunol.* 1988;141(4):1218-1224.
27. Gascan H, Gauchat JF, Aversa G, Van Vlasselaer P, de Vries JE. Anti-CD40 monoclonal antibodies or CD4⁺ T cell clones and IL-4 induce IgG4 and IgE switching in purified human B cells via different signaling pathways. *J Immunol.* 1991;147(1):8-13.
28. Gascan H, Gauchat JF, Roncarolo MG, Yssel H, Spits H, de Vries JE. Human B cell clones can be induced to proliferate and to switch to IgE and IgG4 synthesis by interleukin 4 and a signal provided by activated CD4⁺ T cell clones. *J Exp Med.* 1991;173(3):747-750.
29. Aversa G, Punnonen J, de Vries JE. The 26-kD transmembrane form of tumor necrosis factor alpha on activated CD4⁺ T cell clones provides a costimulatory signal for human B cell activation. *J Exp Med.* 1993;177(6):1575-1585.
30. Punnonen J, Aversa G, Cocks BG, et al. Interleukin 13 induces interleukin 4-independent IgG4 and IgE synthesis and CD23 expression by human B cells. *Proc Natl Acad Sci U S A.* 1993;90(8):3730-3734.
31. Punnonen J, Aversa G, de Vries JE. Human pre-B cells differentiate into Ig-secreting plasma cells in the presence of interleukin-4 and activated CD4⁺ T cells or their membranes. *Blood.* 1993;82(9):2781-2789.
32. Punnonen J, Aversa GG, Vandekerckhove B, Roncarolo MG, de Vries JE. Induction of isotype switching and Ig production by CD5⁺ and CD10⁺ human fetal B cells. *J Immunol.* 1992;148(11):3398-3404.
33. Punnonen J, de Vries JE. IL-13 induces proliferation, Ig isotype switching, and Ig synthesis by immature human fetal B cells. *J Immunol.* 1994;152(3):1094-1102.
34. van Vlasselaer P, Punnonen J, de Vries JE. Transforming growth factor-beta directs IgA switching in human B cells. *J Immunol.* 1992;148(7):2062-2067.